# Differential Privacy Under Class Imbalance: Methods and Empirical Insights

Lucas Rosenblatt [1]    Yuliia Lut [2]    Ethan Turok [2]    Marco Avella-Medina [3]    Rachel Cummings [3]

## Abstract

Imbalanced learning occurs in classification settings where the distribution of class-labels is highly skewed in the training data, such as when predicting rare diseases or in fraud detection. This class imbalance presents a significant algorithmic challenge, which can be further exacerbated when privacy-preserving techniques such as differential privacy are applied to protect sensitive training data. In this paper, we formalize approach-specific privacy challenges faced by standard imbalanced learning remedies and develop algorithmic adaptations with proven DP guarantees. We consider DP variants of *pre-processing* methods that privately augment the original dataset to reduce the class imbalance, alongside DP variants of *in-processing* techniques, which adjust the learning algorithm to account for the imbalance. For each method, we either adapt an existing imbalanced learning technique to the private setting or demonstrate its incompatibility with differential privacy. Finally, we empirically evaluate these privacy-preserving imbalanced learning methods under various data and distributional settings. We find that private synthetic data methods perform well as a data pre-processing step, while class-weighted ERMs are an alternative in higher-dimensional settings where private synthetic data suffers from the curse of dimensionality.

## 1. Introduction

The problem of *imbalanced learning* typically refers to classification tasks where one of the label-classes is substantially underrepresented in the training data. This occurs commonly in real-world applications, such as detecting fraudulent transactions (Makki et al., 2019), medical diagnostics for rare diseases (Singh et al., 2020; Yuan et al., 2018), or predicting natural disasters (Johnson & Khoshgoftaar, 2019). Applying standard machine learning algorithms without adjustment can lead to poor predictions on rare events because these methods are designed for training data that are approximately balanced, or assume that false positives and false negatives have equal misclassification costs.

In the machine learning community, the problem of imbalanced learning has been widely studied in non-private settings (He & Garcia, 2009; Sun et al., 2009; Chawla et al., 2004). This issue can be tackled with two main approaches. The first approach is to use *pre-processing* techniques to balance the training dataset, such as oversampling (Chawla et al., 2002) or other data augmentation based methods. The second approach is to use *in-processing* techniques to modify the machine learning model itself to account for the imbalance, such as bagging (Breiman, 1996) or loss re-weighting (Karakoulas & Shawe-Taylor, 1998; Rigollet & Tong, 2011; Scott, 2012; Tong, 2013; Xu et al., 2020b).

In applications when data are both sensitive and imbalanced – for instance, in the detection of rare diseases (Ficek et al., 2021) – we need machine learning tools that preserve privacy while maintaining high accuracy. Differential privacy (abbreviated DP) has emerged as a powerful technical definition in machine learning and theoretical computer science to address privacy concerns using formal algorithmic tools. Many traditional machine learning algorithms have DP implementations (Gong et al., 2020; Chaudhuri et al., 2011; Abadi et al., 2016; Fletcher & Islam, 2017). However, extending such approaches to private data augmentation methods for imbalanced settings encounters two main challenges. Firstly, it has been shown that private classifiers can amplify minority group loss, magnifying bias and unfairness (Tran et al., 2021; Pujol et al., 2020; Rosenblatt et al., 2024b). Secondly, pre-processing techniques such as oversampling run the risk of increasing sensitivity of the learning task, thus increasing privacy loss. One must therefore be careful when designing privacy-preserving techniques for imbalanced learning that improve performance with respect to the minority class without over-inflating the privacy budget.

---

[*]Equal contribution  [1]New York University  [2]Work completed while Y.L. and E.T. were at Columbia University.  [3]Columbia University. Correspondence to: Lucas Rosenblatt <lr2872@nyu.edu>.

*Proceedings of the 42nd International Conference on Machine Learning*, Vancouver, Canada. PMLR 267, 2025. Copyright 2025 by the author(s).

Table 1: Summary of methods we consider alongside short takeaways. The (**Type**) indicates the practical outlook for each method: **P** (Positive), **N** (Negative), or **M** (Mixed).

| Method (Type) | Key Observations |
| --- | --- |
| Oversampling (Pre-processing, **N**) | Sensitivity rises with repeated samples, leading to excessive privacy loss. |
| SMOTE (Pre-processing, **N**) | Sensitivity scales with dimension, rendering privacy loss impractical. |
| Private Synthetic Data (Pre-processing, **P**) | Post-processed synthetic data preserves DP and performs well for imbalanced tasks. |
| Bagging (Non-Private) (In-processing, **N**) | Lacks meaningful privacy guarantees; too small $\epsilon$ or $\delta$ values are infeasible. |
| Bagging (Private) (In-processing, **N**) | Splitting the privacy budget across weak learners impacts both privacy and utility. |
| Weighted ERM (Pre-processing, **P**) | Theory shows how to incorporate class weights into the objective while maintaining privacy. |
| Weighted DP-SGD (Pre-processing, **M**) | Class weight adjustments for DP-SGD are straightforward, but mixed performance. |

## 1.1. Our Contributions

In this work, we explore both *pre-processing* and *in-processing* methods for private imbalanced binary classification (and note that many of our methods have natural extensions to the multi-class settings). In Section 3, we account for the privacy degradation caused by the well-known data-augmentation technique SMOTE (Chawla et al., 2002), showing that its privacy loss scales as $\Theta(\epsilon 2^d k)$ for $d$-dimensional data and $k$ new data points (unacceptably large for practical settings). This motivates an alternative: using black-box DP synthetic data techniques for augmenting minority data, which is trivially private via post-processing (Proposition 8) and empirically effective.

Shifting to *in-processing* in Section 4, we evaluate model bagging (Breiman, 1996); though prior work claimed an 'intrinsic' DP guarantee for bagging (Liu et al., 2020), we demonstrate that the resulting privacy parameters are not meaningful (Proposition 4). As a positive in-processing result, we adapt the canonical *private* empirical risk minimization (ERM) algorithm (Chaudhuri et al., 2011) to a *class-weighted* variant (Algorithm 1, privacy given in Theorem 5). DP-SGD also trivially allows for class weights (Algorithm 4, privacy given in Proposition 6). However, we find that even a strong neural model trained with class-weighted DP-SGD performs poorly, suggesting neural models may not be ideal for small to medium-sized privacy-preserving imbalanced classification tasks on tabular data. Instead, our extensive experiments on datasets from `imblearn` (Lemaitre et al., 2017) suggest that training a strong non-private model (like XGBoost) on DP class-balanced synthetic data yields the best performance, followed by DP weighted ERM.

## 1.2. Related Work

**Imbalanced learning and privacy.** The problem of imbalanced data often arises in machine learning when the size of one data class is considerably smaller than the other data class. Prior work on imbalanced learning without privacy constraints is extensive (Chawla et al., 2004; He & Garcia, 2009; Sun et al., 2009; Galar et al., 2011; Krawczyk, 2016; López et al., 2013; Branco et al., 2016), alongside work studying adjustments to common learning losses for imbalanced classification (Scott, 2012; Menon et al., 2013). The challenge of handling imbalanced data in machine learning becomes much harder when privacy constraints are added, as accuracy for the minority class can be low even for non-private classification (Lau & Passerat-Palmbach, 2021). Additionally, prior work shows that differentially private algorithms can disproportionally affect minority groups by amplifying the lost of accuracy of a minority class (Bagdasaryan et al., 2019; Jaiswal & Provost, 2020) as well as magnify bias and unfairness (Xu et al., 2020a; Pujol et al., 2020; Farrand et al., 2020; Tran et al., 2021). Work by (Jordon et al., 2019) studies bagging under differential privacy with the assumption of a publicly available data sample; we do not make any such assumptions, and thus can operate in the most general settings.

**Private synthetic data generation.** There has been much progress in recent years on methods for differentially private data synthesis and generation (McKenna et al., 2019; Vietri et al., 2020; Rosenblatt et al., 2020; Aydore et al., 2021; Zhang et al., 2021; Cai et al., 2021; Boedihardjo et al., 2022). Some of the best-performing methods follow the *Select-Measure-Project* paradigm (Tao et al., 2021; McKenna et al., 2022); we discuss further in Section 3.3.

**Private subsampling and deep learning.** It is known that randomly *subsampling* the input database before running a private mechanism can improve the privacy guarantees (known as *amplification by subsampling*) (Bassily et al., 2014; Bun et al., 2015; Wang et al., 2016), although *data-dependent* subsampling can have negative privacy impacts (Bun et al., 2022). Relatedly, the advent of differentially private gradient descent via gradient clipping and moments accounting (Abadi et al., 2016; Sun et al., 2022) has led to privatized versions of many standard deep learning models that exhibit strong empirical performance (Gong et al., 2020; Yousefpour et al., 2021), although private generative models can intensify the imbalance in the data or offer a lower quality synthetic data (Cheng et al., 2021; Ganev et al., 2022). Recent work has expressed skepticism over the performance of these methods, hypothesizing instead that their strength can be partially explained by the effect of unreported hyper-parameter tuning in a "dishonestly" private manner (Papernot & Steinke, 2021; Redberg et al., 2024). Thus, we solely run our models with default hyper-parameter settings.

**Personalized differential privacy.** Our observation that minority samples can incur larger privacy loss echoes the literature on personalized (a.k.a. individualized) DP (PDP) (Kifer & Machanavajjhala, 2011; Jorgensen et al., 2015), which relaxes the global definition by assigning user-specific budgets. In contrast, all results in this paper respect the standard *global* DP guarantee; incorporating PDP into class-imbalance learning pipelines remains an open direction.

## 2. Preliminaries

**Imbalanced Learning.** Let $D = (X, y)$ denote a dataset, where $X$ is a set of $d$-dimensional instances from a known range $[-R, R]^d$ and $y$ is a vector of binary labels. Each $(x_i, y_i) \in [-R, R]^d \times \{0, 1\}$ is a single labeled training example.[1] We partition $X$ into $X^0$ and $X^1$, respectively denoting the sets of entries of $X$ that are labeled with 0 and 1, where these sets are of size $|X^0| = n_0$ and $|X^1| = n_1$. To model the *imbalanced* setting, we assume without loss of generality that $n_1 \ll n_0$.

We define $r = \frac{n_0}{n_1} > 1$ to be the *imbalance ratio* between the positive and negative label classes in the sample. The goal of imbalanced learning is to develop a binary classifier that accurately learns from the imbalanced dataset $D$. In other words, we seek to learn a function $f : [-R, R]^d \to \{0, 1\}$ by minimizing a given loss function $\mathcal{L}$ weighted by class imbalance in the training label distribution.

**Differential Privacy.** DP limits the effect of any individual's data on a computation and ensures that little can be inferred about the individual from an appropriately calibrated randomized output. Intuitively, it bounds the maximum amount that a single data entry can affect analysis performed on the database. Two databases $D, D'$ are *neighboring* if they differ in at most one entry. In this work, we present results for the *bounded* variant of neighboring datasets, i.e., neighboring datasets are the same size, $|D| = |D'|$, and are identical except for a single entry. All of our results can be extended to the *unbounded* variant, i.e., where $D'$ can be constructed through addition/removal, so $|D| = |D'| \pm 1$ (Kifer & Machanavajjhala, 2011).

**Definition 1** (Differential Privacy (Dwork et al., 2006))**.** An algorithm $\mathcal{M} : \mathcal{D} \to \mathbb{R}$ is $(\epsilon, \delta)$-*differentially private* if for every pair of neighboring databases $D, D' \in \mathcal{D}$, and for every subset of possible outputs $\mathcal{S} \subseteq \mathbb{R}$,

$$\Pr[\mathcal{M}(D) \in \mathcal{S}] \le \exp(\epsilon) \Pr[\mathcal{M}(D') \in \mathcal{S}] + \delta.$$

When $\delta = 0$, $\mathcal{M}$ may be called $\epsilon$-differentially private.

---

[1] We assume that $X$ lies in a bounded range because this is necessary for differentially private regression (see, e.g., (Chaudhuri et al., 2011)). If a bound on the data points is not known *a priori*, then one can use domain knowledge or using other private methods such as Propose-Test-Release (Dwork & Lei, 2009).

A well-known technique for achieving $(\epsilon, 0)$-DP is with the Laplace mechanism (given in Appendix A). Alternatively, one can use the the Gaussian mechanism to achieve $(\epsilon, \delta)$-DP (Dwork et al., 2006). In both cases, *noise* is added to the output of a real valued function $f$ that depends on the *sensitivity* of $f$. For the Gaussian mechanism, we consider the $\ell_2$ sensitivity of the function (i.e. $\Delta_2 f = \max_{\text{neighbors } D, D'} \|f(D) - f(D')\|_2$, where $\|\cdot\|_2$ denotes the Euclidean norm), and add noise sampled from $\mathcal{N}\left(\mu = 0, \sigma^2 = (\Delta_2 f)^2 \cdot \frac{2\log(\frac{1.25}{\delta})}{\epsilon^2}\right)$. The Gaussian mechanism further requires that $\epsilon < 1$ for the privacy guarantees to hold. In settings where data points can be unbounded, *clipping* can be applied to project each $X_i$ in the range $[-R, R]$; doing so reduces the sensitivity of the function, and hence the scale of noise that must be added. Additionally, DP has a number of helpful properties. It *composes* (Theorem 7) i.e. the privacy parameter degrades gracefully as additional computations are performed on the same database, and is robust to *post-processing* (Theorem 8), meaning that any further analysis on the output of a differentially private algorithm cannot diminish the privacy guarantees (formal statements available in Appendix A).

## 3. Pre-processing Methods for Private Imbalanced Learning

In this section, we consider applying pre-processing methods for data augmentation to address class imbalance: given a level of class imbalance in the training data, augment or replace the dataset to increase support for the minority class. After applying a pre-processing method, we can then privately learn a classifier on the augmented dataset. The first two methods we consider – *oversampling* in Section 3.1 and *SMOTE* in Section 3.2 – are non-private pre-processing methods; we show that both of these methods *substantially* increase the sensitivity of the downstream private learning mechanism. This increase in sensitivity is due to the fact that these methods generate synthetic minority samples that are highly dependent on the original data, so changing one input point in the original database may lead to *many* points being changed in the augmented database. This motivates our consideration of *private synthetic data generation* for data pre-processing in Section 3.3. In the case of private synthetic data, we instead perform our privacy intervention upstream, learning a DP parameterization of the distribution of our data, from which we can draw arbitrary samples for downstream, non-private model training.

### 3.1. Oversampling

A common technique for dealing with class imbalance in data is to apply an *oversampling* algorithm that first generates $N$ additional synthetic samples from the minority class, before performing learning on the augmented dataset. The

learning algorithm then takes as input the original dataset $D = (X, y)$, concatenated with the $N$ new minority class (positive label) samples. While $N$ can be chosen freely by the analyst, a common parameter regime is to choose $N = n_0 - n_1$ to equalize the size of the two classes. A simple oversampling method is to replicate each minority point in $X_1$ either $\lceil N/n_1 \rceil$ or $\lfloor N/n_1 \rfloor$ times to ensure $N$ total new points; we refer to this as *deterministic oversampling*.[2]

As formalized in Proposition 2, deterministic oversampling increases sensitivity of any downstream DP learning algorithm by a multiplicative factor of $\lceil N/n_1 \rceil + 1$. This is because the maximum of $\lceil N/n_1 \rceil$ additional samples generated from each minority point, along with the minority point themselves, will all be used in the downstream algorithm.

**Proposition 2.** *Let $D = (X, y)$ be a dataset with $n_1$ minority instances, and let $\mathcal{M}$ be an arbitrary $(\epsilon, \delta)$-DP algorithm. Instantiating $\mathcal{M}$ on the dataset $D$ concatenated with the output of oversampling to generate $N$ additional minority samples is $(\epsilon(\lceil \frac{N}{n_1} \rceil + 1), \delta(\lceil \frac{N}{n_1} \rceil + 1))$-differentially private.*

### 3.2. SMOTE

The Synthetic Minority Oversampling TEchnique (SMOTE (Chawla et al., 2002), Algorithm 2, deferred to Appendix B.1) is a more advanced oversampling technique and has become a benchmark for imbalanced learning (see, e.g., (Fernández et al., 2018)). For $N$ iterations, the algorithm: (1) selects an instance from the minority class, (2) finds the $k$ nearest neighbors of this point under $\ell_2$ distance and samples one uniformly at random, and (3) generates a new minority instance as a random convex combination of the original instance and its selected nearest neighbor. Unfortunately, Theorem 3 shows that applying SMOTE as a pre-processing step before any differentially private algorithm substantially increases the sensitivity of the downstream computation: the increase in effective epsilon is exponential in $d$ and linear in $N$. This dramatic increase in the $\epsilon$ factor, if unaccounted for, leads to an overall $\epsilon'$-DP guarantee for extremely large $\epsilon'$ values that provide meaningless privacy guarantees.

**Theorem 3.** *Let $D = (X, y)$ be a $d$-dimensional dataset, with $n_1$ minority instances, and let $\mathcal{M}$ be an arbitrary $\epsilon$-DP algorithm. Then instantiating $\mathcal{M}$ on $D$ concatenated with the output of SMOTE$(X, N, k)$ is both $(\epsilon(2^{0.4042d} \lceil \frac{N}{n_1} \rceil + 1), 0)$-DP and $(\epsilon', \delta)$-DP, for any $\gamma \geq 0$ and for,*

$$\epsilon' = \epsilon(1+\gamma)2^{0.4042d} \left\lceil \frac{N}{n_1} \right\rceil \frac{1}{k}, \; \delta = e^{k2^{0.4042d} \lceil \frac{N}{n_1} \rceil \left(\epsilon - \frac{\gamma^2}{k(2+\gamma)}\right)}.$$

A full proof of Theorem 3 is deferred to Appendix B.1. The-

---

[2]One could also randomly and independently sample a point to replicate from the minority class $N$ times, also exacerbating downstream sensitivity. For simplicity of presentation we stick with deterministic oversampling.

Table 2: SMOTE requires a dramatic adjustment to the privacy parameter:(*left*) the adjusted values of input privacy parameter $\epsilon'$ to the DP algorithm for varying desired privacy budgets $\epsilon$, and (*right*) the resulting privacy budgets $\epsilon$ if $\epsilon'$ is unadjusted, under practical assumptions: $\delta = 1/n^2$ with $n = 10000$, $d = 25$, $k = 5$, $\gamma = 0$, $\lceil N/n_1 \rceil = 1$.

| Input $\epsilon'$ required for desired $\epsilon$. | | | Resulting $\epsilon$ from unadjusted $\epsilon'$. | | |
|---|---|---|---|---|---|
| $\epsilon = 1$ | $\epsilon = 5$ | $\epsilon = 10$ | $\epsilon' = 1$ | $\epsilon' = 5$ | $\epsilon' = 10$ |
| $\epsilon' = 0.00469$ | $0.02346$ | $0.04692$ | $\epsilon = 213.21$ | $1066.06$ | $2132.1$ |

orem 3 should be viewed as a negative result (i.e., SMOTE makes ensuring downstream privacy very difficult). Additionally, we note that with only the $(\epsilon, 0)$-DP result, one might wonder whether the large increase in epsilon can be avoided by allowing a positive $\delta$. Thus, we include and highlight the $(\epsilon, \delta)$ result, which shows that this is not the case; even when a strictly positive failure probability $\delta > 0$ is allowed, the explosion in $\epsilon$ is still present (albeit reduced by a $1/k$ factor). Intuitively, we frame the result as follows: introducing new, minority class examples based on linear interpolations of existing minority class examples leads to *significantly* higher privacy sensitivity. See Table 2 for examples of how large practical $\epsilon$ values can become, after adjusting for the sensitivity of SMOTE preprocessed data.

We further note that this negative result has implications for more advanced class-imbalanced methods that embed the SMOTE algorithm (SMOTEBoost (Chawla et al., 2003), SMOTEBagging (Wang & Yao, 2009), etc.). For good measure, we empirically demonstrate the poor private performance of SMOTE (with the proper sensitivity adjustment) in Appendix B.1.

### 3.3. Private Synthetic Data

We have shown that *non-private* data augmentation techniques for imbalanced learning, like oversampling and SMOTE, explode downstream privacy parameters by amplifying sensitivity. *Private* data pre-processing avoids this limitation. Specifically, we propose leveraging existing private synthetic data algorithms to produce a private balanced dataset that is usable for learning. State-of-the-art methods for producing synthetic data with a DP guarantee follow the same general approach: first, they select DP measurements to evaluate on the data (*Select*), then compute these measurements on the sensitive data (*Measure*), and finally fit a new distribution to those measurements (*Project*) (Liu et al., 2021). New samples can then be drawn from the private distributional model to combat data imbalance – one simple and general approach is to draw enough new samples of the minority class to balance both classes. A formal version of this procedure is given in Algorithm 3 (deferred to Appendix B.2 for space); note that *any* DP synthetic data

generation method could be substituted in to *Stage 1* of Algorithm 3. Furthermore, arbitrarily many samples can be drawn from the privately fitted distributional model due to post-processing (Theorem 8).

Algorithm 3 defaults to performing *conditional* sampling to up-sample the minority class for parametric models (i.e. condition a new generated sample on a fixed positive or negative feature label), as this is sample efficient. For non-parametric models, one can take a more general rejection sampling approach. In Appendix E, we empirically compare two state-of-the-art DP synthetic data methods: the *PrivBayes* (Zhang et al., 2017) and *Generative Networks with the Exponential Mechanism* (GEM) (Liu et al., 2021) algorithms. Both GEM and PrivBayes are parametric models and thus permit conditional class sampling. We then give our empirical results in Section 5 with GEM for clarity, as we found it outperformed PrivBayes across the board.

## 4. In-processing Methods for Private Imbalanced Learning

In-processing methods account for class imbalance by adjusting the learning process. They broadly fall into two main categories: *ensemble-based* classifiers and *cost-sensitive* classifiers. Our first in-processing method we consider in Section 4.1 is bagging, which is an ensemble-based classifier over splits of the training data. We show that although bagging non-private learners does provide some inherent privacy, the resulting DP parameters are *not* meaningful in practice. Cost-sensitive classification assumes a greater cost to misclassifying minority class examples in the training data (Chawla et al., 2004); the primary approach to accommodate asymmetric misclassification costs are weighting strategies during model training. In Section 4.2, we revisit canonical results from (Chaudhuri et al., 2011) on differentially private empirical risk minimization (ERM) and show how to introduce sample weights. Finally, in Section 4.3, we show that the widely-used differentially private stochastic gradient descent (DP-SGD) methods for deep learning can easily accommodate class-based weighting.

### 4.1. Bagging and Private Bagging

Bagging is used widely in practice in imbalanced learning settings, as it has been shown to foster more diversity in model parameters and may help mitigate overfitting to the majority class by elevating minority class importance in the bootstrapped training subsets. This empirical strength, robustness, and improved bias-variance tradeoff of bagging techniques in imbalanced learning is well known (Ueda & Nakano, 1996; Moniz et al., 2017; Haixiang et al., 2017).

The standard bagging procedure (Breiman, 1996) is as follows: create $m$ subsamples $\{D_1, ..., D_m\}$ of a training dataset $D$ by randomly subsampling $k$ examples from $D$ (with or without replacement) to constitute each $D_i$. Then train a base model on each subsample $D_i$ using a base weak learner. To generate a prediction $\hat{y}_i$ for a given sample $X_i$, take the majority vote of predictions from each weak learner.

Since the bagging procedure is randomized, recent work has suggested that it is *intrinsically* differentially private, based on the randomness in sampling and in the predictions of the weak learners, which would imply that bagging is a potential in-processing method for handling imbalanced data. Specifically, Liu et al. (2020) showed that for a dataset of size $n$, bagging with parameters $(m, k)$ satisfies $(\epsilon, \delta)$-DP for $\epsilon = m \cdot k \cdot \ln(\frac{n+1}{n})$ and $\delta = 1 - (\frac{n-1}{n})^{m \cdot k}$.

However, we highlight a significant issue with this approach, simply by inverting these expressions, and solving for $m$ and $k$ given commonly desired settings of $\epsilon$ and $\delta$, namely that $\delta$ is polynomially small in $n$.[3] Proposition 4 states that this re-parameterization reveals a major issue: we cannot set $\delta$ to be very small without setting $\epsilon$ to be exceedingly small as well; see Appendix C.1 for a simple proof.

**Proposition 4.** *For a bagging classifier composed of non-differentially private learners to achieve $\delta = n^{-c}$, then it must also be that $\epsilon \leq \frac{1}{n}$, for all $c > 1$.*

Such a small $\epsilon$ value, paired with a constant-sensitivity function, would not allow the private output to sufficiently vary across different databases, even if they differ in many data-points, meaning that the private output cannot provide meaningful accuracy. Therefore, non-private classifiers *cannot* be used in bagging procedures to simultaneously provide meaningful privacy and accuracy guarantees.

One approach to improving private bagging would be to use *private* classifiers as the weak learners; in that setting, the privacy would follow easily via composition over all the private classifiers used. Given a dataset $D$ and a bagging procedure that trains $m$ $(\epsilon, \delta)$-DP regression models, then by advanced composition (Dwork et al., 2014), for any $\delta' > 0$, this version of private bagging would satisfy $(\epsilon', m\delta + \delta')$-DP for $\epsilon' = \sqrt{2m \ln(1/\delta')} \cdot \epsilon + m\epsilon(e^\epsilon - 1)$. As we show empirically in Appendix C.1 (Figure 3), this can still result in poor empirical performance in reasonable settings. One explanation is that since many private weak learners are needed, the privacy budget is "spread too thinly" over all the classifiers. That is, to satisfy a desired $\epsilon'$ privacy budget, the per-learner privacy parameter $\epsilon$ has to be small, thus significantly reducing performance.

Tighter composition analyses exist based on moments accountants (Abadi et al., 2016; Wang et al., 2019), where the dataset is also subsampled for each computation. These methods are most effective when only a small fraction of

---

[3]Many even prefer a stronger requirement, which is that $\delta$ is cryptographically small, or *negligible*, in $n$.

the dataset are included in each subsample; to contrast, many bagging procedures rely on much larger sub-samples disbursed among fewer learners (Sun et al., 2015). Additionally, in Appendix C.1, we explore a range of private bagging configurations – including disjoint splits (exploiting parallel composition), stratified sampling to preserve minority-class representation (under different privacy assumptions), and alternative voting schemes – yet find these bagging variants similarly yield limited performance under class imbalance. Overall, these empirical results are negative for bagging: one potential reason is that since so few minority class examples existed in the dataset, subsampling further reduces the number of minority examples available to each weak learner. We provide further insights in Appendix C.1.

### 4.2. Weighted Approaches

Cost-sensitive classification assumes a greater *cost* to misclassifying minority class examples and is a well-studied and practically effective method for combating class imbalance (Zhou & Liu, 2005). *Weighting* strategies during model training are the primary approach used to accommodate misclassification costs (Chawla et al., 2004). In Section 4.2.1 we show how to adapt the private ERM given in (Chaudhuri et al., 2011) under a bounded weighting scheme. Later in Section 4.3 we show that DP-SGD can be modified to accommodate weights naturally. We also evaluate weighting strategies with additive noise mechanisms under distributional assumptions where the Bayes optimal classifier is known (e.g., a Gaussian mixture). This follows the setting of (Yang & Xu, 2020), but within a differential privacy framework. We present this as a "warm-up" problem and refer interested readers to Appendix C.2 for details.

#### 4.2.1. WEIGHTED PRIVATE ERM

Standard Empirical Risk Minimization (ERM) trains a model by minimizing an average loss function over a dataset, i.e., optimizing parameters of some model class to reduce the gap between predicted and true data values (Vapnik, 1991; Devroye et al., 2013). Many cost-sensitive approaches to class imbalance rely on sample-weighted objective minimization in the ERM framework, where the minority class samples are up-weighted in the loss function relative to the sample majority (Tang et al., 2008). We show in Theorem 5 that the *differentially private* empirical risk minimization (ERM) procedure of (Chaudhuri et al., 2011) can be adapted to accommodate such minority sample weights, which we outline in Algorithm 1. Weighting samples in the objective function allows us to tune the impact of the minority class on the final model parameters.

We instantiate Algorithm 1 with the weight function $\mathcal{W}(D)$ as the inverted class frequency for each sample in our experiments in Section 5. More formally, for a dataset

$\mathcal{D} = \{(x_i, y_i)\}_{i=1}^n$, where $y_i \in \{0, \dots, k\}$ represents the class label of each sample, we compute the class frequencies for class $k$ as $\hat{\pi}_k = \frac{1}{n} \sum_{i=1}^n \mathbb{I}[y_i = k]$. The inverted class frequency vector $\hat{\pi}^{-1} = (1/\hat{\pi}_0, \dots, 1/\hat{\pi}_k)$ gives the sample weights $w_i = \frac{1}{\pi_{y_i}} \cdot \|\hat{\pi}^{-1}\|_1^{-1} \in [0, 1]$, where each sample is weighted according to the inverse frequency of its class in the dataset. We choose this weighting scheme to align with our "warm-up" results in Appendix C.2 along with prior work (Chawla et al., 2004; Galar et al., 2011).

---

**Algorithm 1** Weighted ERM w/ Objective Perturbation

1 **Inputs:** Data $\mathcal{D} = \{x_i, y_i\}$ with $y_i \in \{0, \dots, k\}$, parameters $\epsilon$, $\lambda$, $c$, loss $\ell(\mathbf{y}_i, \mathbf{x}_i^T \boldsymbol{\beta})$, weight function $\mathcal{W} : \mathcal{D} \to [0, 1]^n$

2 **Output:** Approximate minimizer $\boldsymbol{\beta}_{priv}$.

3 Let $\mathbf{w} = \mathcal{W}(\mathcal{D})$, $\epsilon'' = \frac{4cd}{n(\lambda + \Delta)}$, $\epsilon' = \epsilon - \epsilon''$.

4 If $\epsilon' > 0$ then $\Delta = 0$ else $\Delta = \frac{c}{n(e^{\epsilon/4} - 1)} - \lambda$, $\epsilon' = \frac{\epsilon}{3}$

(ensuring $\epsilon' \geq 0$ as in (Chaudhuri et al., 2011)).

5 Draw vector $\mathbf{b}$ according to PDF $\nu(\mathbf{b}) \propto e^{-\frac{\epsilon' \|\mathbf{b}\|}{3}}$.

6 Compute $\boldsymbol{\beta}_{priv} = \mathrm{argmin}_{\boldsymbol{\beta}} \{\frac{1}{n} \sum_{i=1}^n w_i \cdot \ell(\mathbf{y}_i, \mathbf{x}_i^T \boldsymbol{\beta}) + \frac{1}{n} \mathbf{b}^T \boldsymbol{\beta} + \frac{1}{2} \Delta \|\boldsymbol{\beta}\|^2\}$. =0

---

Theorem 5 states that Algorithm 1 is still DP. To accommodate changes in class balance between neighboring datasets, our privacy analysis handles scenarios where neighboring datasets differ on a minority class label, thereby slightly altering class proportions and associated sample weights.

**Sketch of Theorem 5.** Specifically, we show that the total change in weights, if the minority class size and label proportions changed by one, is bounded as $\sum_{i=1}^n |w_i - w_i'| \leq 2 - \frac{1}{n}$. Consequently, we must adjust the sensitivity of the perturbation vector $\mathbf{b}$. A previous bound of 2 was shown *without weights* in (Chaudhuri et al., 2011), and under our weighting scheme we show that it's necessary to adjust that bound to $\|\mathbf{b} - \mathbf{b}'\|_2 \leq 3$, and to adjust the algorithm accordingly. Here we offer an intuitive sketch of the key idea. Recall that objective-perturbation ensures privacy by adding a random linear term $\mathbf{b}^\top \boldsymbol{\beta}$ whose scale matches the *sensitivity* of the weighted loss. When one minority label flips between neighboring datasets, the class proportions, and hence the sample weights, change by at most $1/n$. This perturbs the weighted empirical risk by $\mathcal{O}(1/n)$ in $\ell_2$ norm, and thus we show that we can retain privacy by inflating the noise radius from 2 (standard ERM) to 3. Importantly, the convexity and smoothness of the loss still guarantee a unique minimizer, and the optimization landscape remains well-behaved. A full proof, with all the necessary adjustments and details introduced by the weighting, is deferred Appendix C.3.1.

**Theorem 5.** *Algorithm 1 instantiated with a loss function $\ell(y, \eta)$ that is convex and twice differentiable with respect to $\eta$, with $|\frac{\partial}{\partial \eta} \ell(y, \eta)| \leq 1$ and $|\frac{\partial^2}{\partial \eta^2} \ell(y, \eta)| \leq c$ for all $y$, is $\epsilon$-differentially private.*

Although our theoretical (and empirical in Section 5) results focus on a logistic regression ERM algorithm, our results directly apply to the kernel method and SVM given in (Chaudhuri et al., 2011). Surprisingly, no adaptation of private ERM under sample weights was previously known; (Giddens et al., 2023) had recently considered the problem for more complicated weighting schemes, but under some undesirable assumptions (more on this in Appendix C.3).

### 4.3. Weighted DP-SGD

Competitive approaches to many private classification problems are given with deep learning models, often tuned using a variant of the differentially private stochastic gradient descent (DP-SGD) algorithm (Bassily et al., 2014; Abadi et al., 2016) (canonical version given in Algorithm 4, but with *weighted* cross-entropy loss). DP-SGD follows an iterative process of sampling mini-batches of the data, computing gradients on the sampled points, clipping the gradients to have a bounded $\ell_2$-norm to reduce sensitivity, adding noise that scales with $\epsilon$ and the clipping parameter to preserve privacy, and finally updating the model using the resulting clipped noisy gradients.

For cost-sensitive gradient updates under class imbalance, it is straightforward to show that weights can be incorporated into a standard binary classification loss $\mathcal{L}(y, \hat{y}; \mathbf{w})$ (e.g. cross-entropy) while maintaining privacy. Proposition 6 formalizes this claim, with the simple proof deferred to Appendix D. It's worth noting, however, that incorporating weights may alter the per-sample gradients in ways that complicate naive sensitivity arguments. In Appendix D.1, we provide a refined analysis (Lemma 19) showing that the overall gradient sensitivity in the weighted setting can be bounded by $2C/B$, ensuring the same privacy guarantees where here $B$ denotes the DP-SGD mini-batch size.

**Proposition 6.** *Algorithm 4, a standard DP-SGD procedure with weighted cross-entropy loss given by* $\mathcal{L}(y, \hat{y}; \mathbf{w}) = -\frac{1}{n} \sum_{i=1}^{n} w_i [y_i \log(\hat{y}_i) + (1 - y_i) \log(1 - \hat{y}_i)]$, *is $(\epsilon, \delta)$-differentially private.*

## 5. Experiments

We evaluate methods under varying privacy and class imbalance conditions on real datasets from the `imblearn` (Lemaitre et al., 2017) repository, with imbalance ratios $r \in [8.6, 130]$ and sizes $n \in [336, 11183]$ (full summary in Table 11, deferred to Appendix E). For private classifiers, we evaluate **(1)** GEM as a *pre-processing* step to generate balanced synthetic data for a non-private XGBoost model (*GEM + NonPriv. XGBoost*, Algorithm 3, Section 3.3), **(2)** a private *in-processing* ERM logistic regression model without class weights (*Priv. LogReg*, (Chaudhuri et al., 2011), Section 4.2), **(3)** a private *in-processing* ERM logistic regression model *with* sample

weights (*Priv. Weighted LogReg*, Algorithm 1, Section 4.2 i.e. private ERM *under class weighting*), and **(4)** an *in-processing* DP-SGD-trained FTTransformer model (Huang et al., 2020) with sample-weighted cross-entropy loss (*Priv. Weighted FTT*, Algorithm 4, Section 4.3). Note that we defer some additional visualizations that help build intuition for the effect of private noise on each classifier's decision boundary (using synthetic data) to Appendix D.2, along with complete results and details (Section E).

### 5.1. Philosophy: Pipelines, not Isolated Models

Throughout the empirical component of this work we consider the overall learning *pipeline*, that couples a privacy mechanism with the downstream model best suited to that mechanism. DP mechanisms are never deployed in a vacuum: their performance depends strongly on how they are paired with downstream architectures, optimization routines, and hyperparameter regimes. In particular, our philosophy is motivated by three observations: **(1.)** Synthetic-data generators decouple privacy from prediction, so their natural partner is a strong non-private tabular learner (e.g. boosted trees). **(2.)** Weighted DP-SGD is expressly tuned for deep networks; forcing it onto a linear model would artificially depress its performance. **(3.)** Prior DP benchmarks (Jayaraman & Evans, 2019; Suriyakumar et al., 2021) likewise compare methods in their most effective configurations. Accordingly, the atomic unit of comparison is the pipeline: ⟨ `DP mechanism`, `intermediate data`, `final predictor` ⟩. Table 1 should be interpreted as comparing *best-effort* pipelines under a common $(\epsilon, \delta)$ budget, rather than as an architecture-controlled ablation.

### 5.2. Evaluations on Real Data

We next empirically evaluate the performance of our methods for private binary classification under class imbalanced data using eight datasets from the Imbalanced-learn (Lemaitre et al., 2017) repository. These datasets represent a variety of settings, with imbalance ratios $r \in [8.6, 130]$ and sizes $n \in [336, 11183]$; see Table 11 for complete details.

In Figure 1, we show how performance varies with privacy level; our performance metrics include general metrics like AUC, F1, and Precision, as well as metrics that are more tailored to imbalanced classification, such as Recall, Worst Class Accuracy, etc. The macro-average accuracy (Macro-Avg-ACC) helps evaluate performance across both classes without bias toward the majority class, while the geometric mean (G-Mean) balances sensitivity and specificity. Higher is better for all metrics. Figure 1 presents results on the *mammography* dataset, which was representative of general trends for all datasets. Complete plots are presented in Figures 6 to 13 in Appendix E.

Table 3: Average performance rankings of the DP imbalanced learning approaches, across all $\epsilon$ settings and datasets. Average ranks are in $[1, 4]$ and in descending order, so lower is better. We adopt the Olympic medal convention: gold , silver and bronze cells signify first, second and third best performance, respectively.

| Model | AUC | F1 | Bal-ACC | Precision |
|---|---|---|---|---|
| GEM + XGBoost | 1.45 | 1.45 | 1.48 | 1.45 |
| Priv. LogReg | 2.77 | 2.62 | 2.89 | 2.62 |
| Priv. Weighted LogReg | 3.19 | 2.65 | 2.59 | 2.65 |
| Priv. Weighted FTT | 2.89 | 3.58 | 3.34 | 3.58 |
| | **Recall** | **Worst-ACC** | **Macro-ACC** | **G-Mean** |
| GEM + XGBoost | 2.26 | 1.45 | 1.48 | 1.45 |
| Priv. LogReg | 2.20 | 2.89 | 2.89 | 2.86 |
| Priv. Weighted LogReg | 2.11 | 2.59 | 2.59 | 2.59 |
| Priv. Weighted FTT | 3.70 | 3.37 | 3.34 | 3.40 |

**Varying Privacy Budget** We observed that for all datasets, the *GEM+XGBoost* method improved with increased privacy budget. Figure 1 presents results on the *mammography* dataset, which is representative of general trends. Higher dimensionality increased the difficulty across the board (e.g., there was a larger difference between non-private performance and private performance with, for example, the *car_eval_4* dataset (Figure 10)), but we did not find a meaningful trend or interaction between imbalance ratio and dimensionality. Absolute dataset size correlated with the classification performance, as expected. Complete plots are presented in Figures 6-13 in Section E.

Additionally, in our experiments, we found that more minority examples led to more stable improved performance from the *GEM+XGBoost* model. For example, the *mammography* (Figure 1) and *abalone* (Figure 13) datasets, both of which have the highest number of minority class examples, also exhibited the best performance for the *GEM+XGBoost* synthesizer at low levels of epsilon, and the most stable performance overall across varied privacy parameters.

In Table 3, we present average rankings across all datasets and epsilon values for the four privacy-preserving imbalanced learning approaches we explore; here, lower is better, and highest average performance in each row is highlighted according to the Olympic medal convention (gold, silver, bronze). *GEM + XGBoost* performs best, ranking highest across 7 of the 8 metrics on average. As expected, *Priv. Weighted LogReg* performs worse than its unweighted counterpart on overall metrics. Overall metrics are well known to be poor indicators in imbalanced learning, as many of them weight negative and positive class performance equally (He & Garcia, 2009). However, on the metrics more appropriate for imbalanced classification, *Priv. Weighted LogReg* outperforms the unweighted variant in 3 out of 4 metrics, and has the best average Recall among all private models. In stark contrast, *Priv. Weighted FTT* consistently under-performed.

Figure 1: Performance for *mammography* dataset for overall performance metrics (AUC, F1, Balanced Accuracy, Precision) and metrics appropriate for imbalanced classification settings (Recall, Worst Class Accuracy, Macro Average Accuracy, Geometric Mean) under varying $\epsilon \in \{0.05, 0.1, 0.5, 1.0, 5.0\}$ and over 10 random initializations.

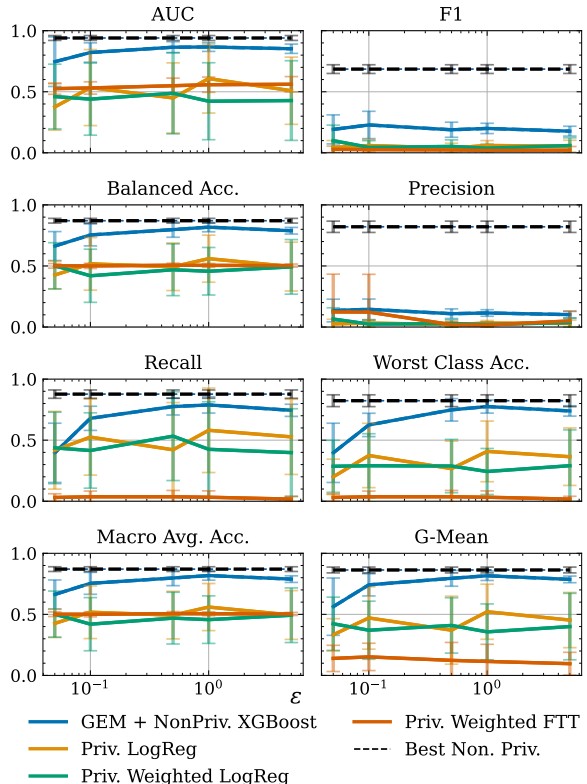

**Empirical Takeaways.** Private variants of neural models (*Priv. Weighted FTT*, for example) may be inappropriate in general for relatively low-data regimes under class imbalance due to minority example sparsity, especially when weighted ERM based methods like *Priv. Weighted LogReg* perform well and are less expensive to train. Moreover, pre-processing with private synthetic data (*GEM + Non-Priv. XGBoost*) displayed the most robust performance across varying privacy levels and imbalanced datasets in our experiments, consistently ranking highest across nearly all metrics. Unfortunately, DP data synthesis is limited to low-dimensional datasets (Rosenblatt et al., 2022). *Priv. Weighted LogReg* performed best in terms of Recall, and performed second best on average in terms of the other imbalanced classification metrics. Our empirical results lead us to recommend these two methods, pre-processing with private synthetic data or weighted ERM based methods, depending on chosen metric or data context.

**Explaining GEM's Strong Performance** One plausible explanation for GEM's strong performance is rooted in well-

studied properties of differentially private synthetic data generation (Tao et al., 2021; McKenna et al., 2022; Qian et al., 2024). In particular, DP synthetic data algorithms rely heavily on $k$-way counting queries, which are precisely the kinds of aggregate statistics that differential privacy is designed to protect and measure accurately. By focusing on low-dimensional correlations and co-occurrence relationships – which may be sufficient for many classification tasks (Hollmann et al., 2025) – GEM can capture and preserve essential structure in the data with high fidelity. Moreover, class-conditional sampling allows GEM to up-sample minority classes similarly to SMOTE, mitigating imbalance *at its root*. While a formal theory to explain these phenomena would likely require distributional assumptions and an extensive theoretical framework, our experiments suggest these capabilities help explain GEM's performance.

## 6. Conclusion and Future Work

Private binary classification under class imbalance is especially challenging. We showed that standard non-private methods like SMOTE and bagging become ill-suited for DP, largely because they inflate the downstream privacy loss. By contrast, private weighted ERM and DP private synthetic data generation approaches can preserve privacy and yield good performance, although DP synthetic data methods suffer from the curse of dimensionality, which is a significant limitation in most practical settings (McKenna et al., 2019; Rosenblatt et al., 2022).

While our study focused on binary classification (a common setting in fraud detection, rare-disease diagnosis, spam filtering, etc.), many of our ideas naturally extend to multiple classes. **Pre-processing.** For oversampling or SMOTE, Proposition 2 and Theorem 3 depend on the factor $\lceil N/n_1 \rceil$, i.e. the *maximum* number of synthetic samples per real minority point. In the multiclass setting we could simply replace this by $\max_{k \in [c]} \lceil N_k/n_k \rceil$, analyze each class independently, and take the worst-case bound. **In-processing.** Weighted ERM admits multinomial logistic regression with categorical cross-entropy; the convexity assumptions required by Theorem 5 continue to hold. Likewise, DP-SGD uses the same gradient-clipping routine with a softmax loss, so the sensitivity bound of Lemma 19 carries over verbatim. **Synthetic data.** GEM and other Select–Measure–Project synthesizers already model the joint distribution of all attributes; conditioning on a categorical target variable merely changes the sampling query. With these natural extensions in mind, we leave a rigorous end-to-end evaluation of multiclass performance to future work.

In addition, there are ample open questions about the interplay between differential privacy and imbalanced learning. New imbalanced-learning-specific loss functions (Cui et al., 2019; Cao et al., 2019; Li et al., 2021) might further improve private classification. Hybrid methods that leverage DP synthetic data in lower-dimensional embeddings before training a weighted ERM model also warrant exploration. We hope our initial findings prompt broader theoretical and empirical advances in private imbalanced classification.

## Impact Statement

Learning algorithms are deployed more and more in sensitive domains, such as healthcare and finance, necessitating algorithms that ensure data privacy, *particularly* in the presence of class imbalance (for example, when fairness or equity are a concern). Our work highlights the challenges of applying standard imbalanced learning techniques in differentially private settings, identifying both algorithmic tools that work and those that do not. We believe that combining DP techniques with imbalanced classification strategies can lead to the safeguarding of individual-level information while maintaining strong predictive performance in high stakes settings. Nonetheless, deploying these approaches in practice would require careful consideration of the domain constraints, and likely some collaboration with policymakers and other stakeholders to ensure responsible use.

## Acknowledgements

Y.L. was supported in part by NSF grant CNS-2138834 (CAREER). M.A.M. was supported in part by NSF grant DMS-2310973. R.C. was supported in part by NSF grants CNS-2138834 (CAREER) and IIS-2147361. L.R. was supported by the NSF under GRFP Grant No. DGE-2234660.

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

# A. Deferred Preliminaries

In this section, we provide some standard preliminaries for DP mechanisms that were deferred for length. One well-known technique for achieving $(\epsilon, 0)$-DP is by adding Laplace noise. The *Laplace distribution* with scale $b$ is the distribution with probability density function: $h(x|b) = \frac{1}{2b} \exp(-\frac{|x|}{b})$. The Laplace Mechanism of (Dwork et al., 2006) takes in a real-valued function $f$, a database $D$, and a privacy parameter $\epsilon$, and produces the (random) output: $f(D) + \text{Lap}(\Delta f / \epsilon)$. For the Laplace mechanism, the sensitivity of a real-valued function $f$ is defined as: $\Delta f = \max_{\text{neighbors } D, D'} |f(D) - f(D')|$.

Additionally, here are formal statements of two key theorems in differential privacy. Composition (Theorem 7) states that applying two differentially private algorithms sequentially results in combined privacy guarantees of $(\epsilon_1 + \epsilon_2, \delta_1 + \delta_2)$. Post-processing (Theorem 8) notes that any post-processing of the output of a differentially private mechanism preserves its original privacy guarantees.

**Theorem 7** (Basic Composition (Dwork et al., 2006)). *Let $\mathcal{M}_1$ be an algorithm that is $(\epsilon_1, \delta_1)$-DP, and let $\mathcal{M}_2$ be an algorithm that is $(\epsilon_2, \delta_2)$-DP. Then their composition $(\mathcal{M}_1, \mathcal{M}_2)$ is $(\epsilon_1 + \epsilon_2, \delta_1 + \delta_2)$-DP.*

**Theorem 8** (Post-processing (Dwork et al., 2006)). *Let $\mathcal{M} : \mathcal{D} \to \mathcal{R}$ be an algorithm that is $\epsilon$-differentially private, and let $f : \mathcal{R} \to \mathcal{R}'$ be an arbitrary function. Then $f \circ \mathcal{M} : \mathcal{D} \to R'$ is $\epsilon$-differentially private.*

## B. Pre-processing Methods and Analysis

---

**Algorithm 2** SMOTE($X_1$, $N$, $k$) (Chawla et al., 2002)

---

1 **Input:** minority class instances $X_1 = \{x_1, \ldots x_n\}$, dataset dimension $d$, number of points to be generated $N$, number of nearest neighbors $k$.
2 **Output:** $N$ synthetic minority class samples
   **for** $i = 1, \ldots, n$ **do**
3    Compute $k$ nearest $\ell_2$ neighbors of $x_i$ from $X_1$: $(x_i^1, \ldots, x_i^k)$
   **end for**
   **for** $t = 1, \ldots, N$ **do**
4    $i = t \mod n$, where $0 \mod n$ is interpreted as $n$
5    Randomly choose $x_i'$, one of the $k$ nearest neighbors of $x_i$
     **for** $j = 1, \ldots d$ **do**
6      Sample $u_j$ uniformly from $[0, 1]$
7      $z_{t,j} = (1 - u_j)x_{i,j}' + u_j \cdot x_{i,j}$
     **end for**
8    **return** $(z_t, 1)$
   **end for**=0

---

### B.1. SMOTE

Before offering the complete proof of the SMOTE result (Theorem 3), we provide a brief sketch to help guide the reader. We first define the quantity $Y = |SMOTE(X, N, k) \oplus SMOTE(X', N, k)|$ which gives the symmetric difference between SMOTE applied to two neighboring datasets $X$, $X'$, where $\oplus$ denotes symmetric difference. $Y$ can be fully described as a sum of Bernoulli random variables with parameters that depend on $k$, $N$, $n_1$, and the maximum number of times one point from $\mathbb{R}^d$ can appear among $k$-nearest neighbors of other points from $\mathbb{R}^d$. SMOTE only takes in the minority class data, and does not use majority class data at all in generating new synthetic data. Thus, without loss of generality, Theorem 3 only considers the modification of a minority class example that has a positive label; if the minority class was actually the negative label, this could be dealt with in the analysis simply by renaming.

Now, to bound the maximum number of times one point can appear among $k$-nearest neighbors of other points, denoted $l(d, k)$, we require Lemma 9. This lemma lower bounds $l(d, k)$ via a geometric argument that relies on the notion of a *kissing number* $K(d)$, defined as the greatest number of equal non-overlapping spheres in $\mathbb{R}^d$ that can touch another sphere of the same size (Musin, 2008; Jenssen et al., 2018).

**Lemma 9.** *Let $l(d, k)$ be the maximum number of times one point from $\mathbb{R}^d$ can appear among the $k$-nearest neighbors of $n_1$ other points from $\mathbb{R}^d$. Then, $l(d, k) = \min\{k \cdot K(d), n_1\}$.*

The exact value of the kissing number $K(d)$ for general $d$ is an open problem, but is known to be asymptotically bounded by $k2^{0.2075d(1+o(1))} \le l(d, k) \le k2^{0.4042d}$ (Wyner, 1965; Musin, 2008; Kabatiansky & Levenshtein, 1978). Returning to $Y$, we then apply a one-sided Chernoff bound constraining the probability that $Y$ is much greater than its mean. Plugging in Lemma 9 and appropriate parameters yields Theorem 3.

Now, for the formal proof, following a restatement of Theorem 3.

**Theorem 3.** *Let $D = (X, y)$ be a $d$-dimensional dataset, with $n_1$ minority instances, and let $\mathcal{M}$ be an arbitrary $\epsilon$-DP algorithm. Then instantiating $\mathcal{M}$ on $D$ concatenated with the output of SMOTE($X, N, k$) is both $(\epsilon(2^{0.4042d}\lceil \frac{N}{n_1}\rceil + 1), 0)$-DP and $(\epsilon', \delta)$-DP, for any $\gamma \ge 0$ and for,*

$$\epsilon' = \epsilon(1 + \gamma)2^{0.4042d}\left\lceil \frac{N}{n_1}\right\rceil \frac{1}{k}, \; \delta = e^{k2^{0.4042d}\lceil \frac{N}{n_1}\rceil\left(\epsilon - \frac{\gamma^2}{k(2+\gamma)}\right)}.$$

*Proof.* Let $D = (X, y)$ and $D' = (X', y')$ be two neighboring datasets such that $D' = D \cup \{(x, 1)\}$, and let $M$ be an arbitrary $\epsilon$-DP algorithm. For the remainder of the proof, fix SMOTE parameters $N \in \mathbb{N}$ and $k \in \mathbb{Z}^+$. Define $l(d, k)$ to be the maximum number of times one point from $\mathbb{R}^d$ can appear among the $k$-nearest neighbors of an arbitrary set of other points in $\mathbb{R}^d$. To simplify notation, we may denote $l(d, k)$ as simply $l$ when $d$ and $k$ are clear from context.

To compare the outputs of SMOTE$(X, N, k)$ and SMOTE$(X', N, k)$, we fix the internal randomness of SMOTE between these two runs, which includes randomly choosing a nearest neighbor and randomly sampling $u$ inside the for-loop. That is, an output point will be different only if the new point $x$ in $X'$ replaces the selected nearest neighbor $x'_i$ that was chosen under $X$. For each iteration where $x$ replaces a previous nearest neighbor, there is a $1/k$ probability that $x$ is the selected nearest neighbor.

Define the random variable $Y = |SMOTE(X, N, k) \oplus SMOTE(X', N, k)|$, where $\oplus$ denotes a symmetric difference. Then $Y$ can be described as the sum of independent Bernoulli random variables that are 1 if $x$ is the selected $k$-nearest neighbor. Each trial has success probability $1/k$, and there are $l \lceil \frac{N}{n_1} \rceil$ total trials, corresponding to the $l$ datapoints that can be neighbors to $x$ and the $\lceil \frac{N}{n_1} \rceil$ iterations through the database. For simplicity of presentation, we drop the ceiling notation for the remainder of the proof, but it is implied if $N$ is not divisible by $n_1$. Thus $Y \sim Binomial\left(\frac{l \cdot N}{n_1}, 1/k\right)$ and $\mathbb{E}[Y] = \frac{l \cdot N}{n_1 \cdot k}$.

Note that using the upper bound $Y \leq \frac{lN}{n_1}$, we can obtain an immediate DP guarantee for $M$ applied to the output of SMOTE using group privacy. Specifically, since $Y \leq \frac{lN}{n_1}$, we know that changing one entry of $X$ would change up to $\frac{lN}{n_1}$ entries of the output of SMOTE, which is equivalent to changing $\frac{lN}{n_1} + 1$ entries of the input to $M$ (since the input to $M$ is the original database $X$ concatenated with the output of SMOTE). Thus by the group privacy property of DP, these $\frac{lN}{n_1} + 1$ entries that depend on $x$ would jointly receive a $(\epsilon(\frac{lN}{n_1} + 1), 0)$-DP guarantee.

However, since $Y$ is a random variable, one can instead use a high probability bound on $Y$ as it may lead to an improved $\epsilon$ bound. There is some chance that $Y$ will fail to satisfy this bound, and this failure probability will later be incorporated into the $\delta$ parameter of DP. Using a one-sided Chernoff bound, we bound the probability that $Y$ is significantly greater than its mean:

$$\Pr\left[Y \geq (1 + \gamma)\frac{lN}{n_1 k}\right] \leq e^{-\frac{\gamma^2}{2+\gamma}\frac{lN}{n_1 k}}, \tag{1}$$

for any $\gamma \geq 0$.

For ease of notation, let $N_l = l \cdot N/n_1$, and let $T(D) = (X, y) \cup SMOTE(X, N, k)$. Then for an arbitrary set of outputs $S \subset Range(M)$, we can obtain the following bounds on the output of $M \circ T$ on $D$ and $D'$:

$$\Pr[M(T(D)) \in S] = \sum_{j=1}^{N_l} \Pr[M(T(D)) \in S | Y = j] \cdot \Pr[Y = j]$$

$$\leq \sum_{j=1}^{N_l} e^{\epsilon \cdot j} \Pr[M(T(D')) \in S | Y = j] \cdot \Pr[Y = j]$$

$$= \sum_{j=1}^{\frac{(1+\gamma)N_l}{k}} e^{\epsilon \cdot j} \Pr[M(T(D')) \in S | Y = j] \cdot \Pr[Y = j] + \sum_{j=\frac{(1+\gamma)N_l}{k}+1}^{N_l} e^{\epsilon \cdot j} \Pr[M(T(D')) \in S | Y = j] \cdot \Pr[Y = j]$$

$$\leq e^{\epsilon \cdot (1+\gamma)N_l/k} \sum_{j=1}^{(1+\gamma)N_l/k} \Pr[M(T(D')) \in S | Y = j] \Pr[Y = j] + e^{\epsilon \cdot N_l} \Pr[Y \geq (1+\gamma)\frac{N_l}{k}]$$

$$\leq e^{\epsilon \cdot (1+\gamma)N_l/k} \Pr[M(T(D')) \in S] + e^{\epsilon \cdot N_l - \frac{\gamma^2}{2+\gamma}\frac{N_l}{k}}.$$

The first equality is due to the law of total probability, the second step is due to the group privacy property of DP, and the third step separates the sum into small and large $j$ values based on the parameter $\gamma$. In the fourth and fifth steps, we bound each coefficient $e^{\epsilon j}$ by the largest value of $j$ in the respective sum. For small $j$ values we then the apply the law of total probability; for large $j$, we upper bound each term $\Pr[M(T(D')) \in S | Y = j]$ by 1, so the remaining sum is simply the probability that $Y$ is greater than the smallest "large" $j$ value, which is then bounded by the one-sided Chernoff bound of Equation (1).

Therefore, the composition of first applying SMOTE and then applying $M$ to the original dataset along with the output of SMOTE is $\left(\epsilon(1+\gamma)\frac{lN}{n_1 k}, e^{\epsilon\frac{lN}{n_1} - \frac{\gamma^2}{2+\gamma}\frac{lN}{kn_1}}\right)$-differentially private.

Next, we prove Lemma 9, which gives a lower bound for a parameter $l(d, k)$ that describes the maximum number of times

one point from $\mathbb{R}^d$ can appear among $k$-nearest neighbors of other points from $\mathbb{R}^d$. The proof is a geometric argument that relies on the notion of *kissing number* $K(d)$, which is the greatest number of equal sized non-overlapping spheres in $\mathbb{R}^d$ that can touch another sphere of the same size (Musin, 2008; Jenssen et al., 2018).

**Lemma 9.** *Let $l(d, k)$ be the maximum number of times one point from $\mathbb{R}^d$ can appear among the $k$-nearest neighbors of $n_1$ other points from $\mathbb{R}^d$. Then, $l(d, k) = \min\{k \cdot K(d), n_1\}$.*

*Proof.* Trivially $l(d, k) \leq n_1$ so in the following, we will consider the case where $n_1$ is sufficiently large. Also w.l.o.g. we will consider $K(d)$ kissing number spheres of radius $r = 1$.

Consider constructing a point set $S$ around the origin $O = (0, ..., 0) \in \mathbb{R}^d$ such that $S$ contains points whose 1-nearest neighbor is $O$. We next define the points $x_i \in \mathbb{R}^d$, s.t. $S = \{x_1, ..., x_{K(d)}\}$ where each $x_i$ is the center point in each of the $K(d)$ kissing point spheres around the unit sphere centered at $O$. By construction, each $||x_i||_2 = 2$ and $||x_i - x_j||_2 \geq 2$ for every other $x_j \in S$.

We have so far a set $S$ with cardinality $|S| = K(d)$, which contains the $K(d)$ centroids whose 1-nearest neighbor is $O$. Recall that we allow ourselves to break ties in distance arbitrarily. Ties are often broken probabilistically in $k$-nearest implementations, but we are considering the "worst-case" scenario for our analysis of $S$.

We now demonstrate that we cannot locally increase $S$. That is, $K(d)$ is the maximum number of points who can share a 1-nearest neighbor with $O$. We show this by contradiction.

Consider adding a new point $x_{K(d)+1}$ into the set $S$ of 1-nearest neighbors with $O$. How can $x_{K(d)+1}$ be a valid 1-nearest neighbor of $O$? If $||x_{K(d)+1}||_2 > 2$, $O$ is certainly not its 1-nearest neighbor; instead, for some $x_i \in S$, $||x_{K(d)+1} - x_i||_2 < ||x_{K(d)+1} - O||_2$ by construction. If $||x_{K(d)+1}||_2 \leq 2$ then it would either: **(1)** Have $O$ as its 1-nearest neighbor, implying that $||x_{K(d)+1} - O||_2 \leq ||x_{K(d)+1} - x_i||_2$ for all $x_i \in S$, which then implies that $||x_j - x_{K(d)+1}||_2 \leq ||x_j - O||_2$ for at least one point $x_j \in S$, thus either shrinking $|S|$ or leaving it the same size, or, **(2)** Have a fixed $x_j \in S$ as its 1-nearest neighbor, implying that $||x_{K(d)+1} - x_j||_2 \geq ||x_{K(d)+1} - x_i||_2$ for all $x_i \in S$ and $O$, but then implying that $||x_j - x_{K(d)+1}||_2 \leq ||x_a - O||_2$ for some other $x_a \in S$ by the triangle inequality, again shrinking $|S|$. Thus, a new point $x_{K(d)+1}$ cannot be added to $S$ when $k = 1$, and $l(d, 1) = K(d)$.

Next we generalize this result from 1-nearest neighbors to $k$ nearest neighbors, demonstrating that $l(d, k) = kK(d)$. We will do this by duplicating points in $S$ from the 1-nearest neighbor construction to create a set $S'$, and then again show by contradiction that this set $S'$ cannot locally increase in size.

For $k$-nearest neighbor, we construct a set of points $S'$ as follows, where each $x_i^j$ for $j \in \{1, ..., k\}$ is an exact replica of $x_i$ from $S$. Thus, $S' = \{x_1^1, x_2^1, ..., x_{K(d)}^1\} \cup ... \cup \{x_1^k, x_2^k, ..., x_{K(d)}^k\}$. Note that $|S'| = kK(d)$, where we have $k$ duplicates of the set $S$ from the 1-nearest neighbor example.

For each point $x_i^j \in S'$, there are $k - 1$ points $\{x_i^1, ..., x_i^k\} \neq x_i^j$ for which $||x_i^j - x_i^c||_2 = 0$. As before, the distance from the origin to each point $||x_i^j - O|| = 2$ by construction, and for each $x_i^j$ and the $kK(d) - k + 1$ points $x_a^b$ that are not duplicates of $x_i^j$, $||x_i^j - x_a^b|| \geq 2$. Thus for $S'$ of size $kK(d)$, then $O$ is a $k$-nearest neighbor of every point in $S'$, using worst-case tie breaking.

We next show that the number of points with $O$ as a nearest neighbor cannot increased by adding a new point $x_{K(d)+1}^{j+1}$ to $S'$. The argument is analogous to the argument for 1-nearest neighbor. If $||x_{K(d)+1}^{j+1}||_2 > 2$, then $O$ is certainly not its $k$-nearest neighbor; instead, for some size $k$ set $\{x_i^1, ..., x_i^k\} \subset S'$, $||x_{K(d)+1}^{j+1} - x_i^j||_2 < ||x_{K(d)+1}^{j+1} - O||_2$ by construction. If $||x_{K(d)+1}^{j+1}||_2 \leq 2$ then it would either:

- Have $O$ in its set of $k$-nearest neighbors, implying that $||x_{K(d)+1}^{j+1} - O||_2 \leq ||x_{K(d)+1}^{j+1} - x_i^j||_2$ for at least one $x_i^j \in S'$, which then implies that $||x_i^a - x_{K(d)+1}^{j+1}||_2 \leq ||x_i^a - O||_2$ for at least one point $x_i^a \in S$, thus either shrinking $|S'|$ or leaving it the same size.

- Have an entire set $\{x_i^1, ..., x_i^k\} \in S'$ as its $k$-nearest neighbors, again shrinking $|S|$ by the triangle inequality as in the 1-nearest neighbor argument.

Thus, we have shown that $|S'|$ cannot be locally improved, and that $l(d, k) = kK(d)$. □

The exact value of the kissing number $K$ in general $d$ dimensions is an open problem, but is known to be lower bounded by $K \geq 2^{0.2075d(1+o(1))}$ (Wyner, 1965; Musin, 2008) and upper bounded by $K \leq 2^{0.4042d}$ (Kabatiansky & Levenshtein, 1978). Thus when $n_1$ is not too small, $k2^{0.2075d(1+o(1))} \leq l(d,k) \leq k2^{0.4042d}$. We note that even though the exact value of the kissing number is unknown, its bounds are asymptotically tight, with exponential dependence on $d$.

Plugging in the maximum value of $k2^{0.4042d}$ for $l(d,k)$ into the differentially privacy bounds derived above recovers the guarantees of the theorem. $\qquad\square$

**Empirical Results: SMOTE**   Here we present simple empirical results illustrating that SMOTE as a pre-processing step before differentially private learning results in extremely poor performance. Figure 2 presents the performance of SMOTE as a pre-processing method before DP logistic regression with three different $\epsilon$ values, compared with non-private logistic regression and DP logistic regression without SMOTE applied. The evaluation is performed on the *mammography* dataset (see Section 5) with a variety of imbalance ratios created by subsampling.

As predicted, downstream performance degrades significantly after SMOTE-induced $\epsilon$ adjustments as described in Table 2. Note how proper privacy adjustments after SMOTE (dotted lines) negatively impact performance compared to DP logistic regression without SMOTE (solid red line). This empirically confirms our negative result of Theorem 3, that SMOTE should not be a preferred pre-processing method for differentially private imbalanced learning.

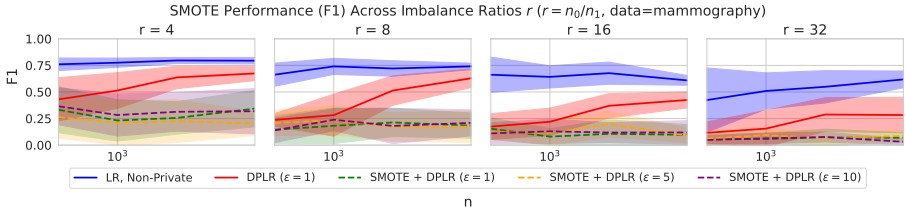

Figure 2: SMOTE pre-processing on downstream DP logistic regression (with adjusted $\epsilon$) on the *mammography* dataset. Data was subsampled (log-scale x-axis: $n \in [500, 1000, 2000, 5000, 10000]$) and evaluated across imbalance ratios $r \in [4, 8, 16, 32]$.

## B.2. Re-sampling with DP Synthetic Data

The privacy of Algorithm 3 is straightforward to see: as long as the data synthesizer in Stage 1 is $(\epsilon, \delta)$-DP, then Stage 2 will retain the same privacy guarantee by post-processing (Theorem 8).

**Proposition 10.** *Algorithm 3 is $(\epsilon, \delta)$-differentially private.*

---

**Algorithm 3** Balancing w/ Private Data Synthesizer

---

1 **Input:** $(\epsilon, \delta)$-differentially private data synthesizer $\mathcal{S}$, original dataset $D$, desired number of samples $N$, and any additional parameters for $\mathcal{S}, \mathcal{P}$.
2 **Output:** A balanced dataset $D'$ where $n_0 = n_1$.
3 **Stage 1: Parameterize a Distribution**
4 Learn/parameterize a differentially private distribution $\theta$ over the data domain i.e. $\theta \leftarrow \mathcal{S}(D, \mathcal{P})$.
5 **Stage 2: Sample a New Dataset** $D'$
   **if** $\theta$ is parametric **then**
6    Sample $N/2$ minority examples $D'_{n_1} \sim \theta \mid n_1$, then sample $N/2$ majority examples $D'_{n_0} \sim \theta \mid n_0$.
7    **return** concatenation $[D'_{n_1}, D'_{n_0}]$.
   **else if** $\theta$ is non-parametric **then**
8    Perform rejection sampling based on class label to draw balanced samples (i.e., ensure $n_0 = n_1 = N/2$ in the final dataset $D'$ by sampling from $\mathcal{S}(D)$ until target sizes are reached).
   **end if**
9 **return** $D'$ =0

---

# C. In-processing Methods and Analysis

## C.1. Bagging and Private Bagging

**Proposition 4.** *For a bagging classifier composed of non-differentially private learners to achieve $\delta = n^{-c}$, then it must also be that $\epsilon \leq \frac{1}{n}$, for all $c > 1$.*

*Proof.* From Theorem 3 in (Liu et al., 2020), given a training dataset of size $n$ and an arbitrary non-private base learner, bagging with replacement with $m$ base models and a subsample size of $k$ has privacy parameters $\epsilon = m \cdot k \cdot \log(\frac{n+1}{n})$ and $\delta = 1 - \left(\frac{n-1}{n}\right)^{m \cdot k}$. Solving for $m \cdot k$ in the $\delta$ equation and plugging in $\delta = n^{-c}$ yields,

$$m \cdot k = \frac{\log(1 - n^{-c})}{\log(n-1) - \log(n)} \ .$$

Plugging this in to the expression for $\epsilon$ gives, for $n > 1$,

$$\epsilon = \log(1 - n^{-c})\frac{\log(n+1) - \log(n)}{\log(n-1) - \log(n)}$$
$$= \log(1 - n^{-c})\frac{\log\left(1 + n^{-1}\right)}{\log\left(1 - n^{-1}\right)}$$
$$\leq \log(1 - n^{-1})\frac{\log\left(1 + n^{-1}\right)}{\log\left(1 - n^{-1}\right)}$$
$$= \log\left(1 + n^{-1}\right)$$

Thus, for $c > 1$, applying the bound of $\log(1 + x) \leq x$ yields the result. □

**Empirical exploration**   We also present some empirical results for DP Bagging to illustrate its poor performance as an in-processing method for DP imbalanced learning. Figure 3 presents the performance of DP bagging using DP logistic regression as a weak learner, compared against the two baselines of non-private logistic regression and DP logistic regression without bagging. For each DP-LR in the Bagged classifier, the privacy budget was split among the estimators using advanced composition (setting $\epsilon = 1/2$ and $\delta = 1/n^2$). The evaluation is performed on the *mammography* dataset (see Section 5) with a variety of imbalance ratios created by subsampling. As predicted, we observe that private bagging underperformed relative to a single DP logistic regression classifier across sample sizes and imbalance ratios.

**Disjoint Parallel Composition vs. Bootstrapped Bagging**   One natural approach to improving privacy in bagging is to train each weak learner on disjoint subsets of the dataset (rather than on bootstrapped or overlapping subsets). In principle, using *parallel composition* may yield a better overall privacy guarantee than the standard bootstrapped setting with advanced composition: because each individual's data can only appear in one of the weak learners, the overall privacy cost does not grow with the number of learners in the same way that it does under advanced composition.

To this end, we first evaluated a *uniformly sampled disjoint* bagging method that partitions the dataset into $k$ subsets of size $N/k$, trains one DP logistic regression (DPLR) learner per subset, and then aggregates predictions via majority vote. Table 4 (for imbalance ratio $r = 8$) and Table 5 (for $r = 32$) compare this *Disjoint* approach against the *Adv. Comp.* version, where each weak learner is trained on a bootstrapped subsample and the privacy budget is composed using advanced composition rules. We vary the total number of learners $k \in \{5, 10, 20\}$ and report $F1$ scores (mean $\pm$ std. dev. over 10 runs). Overall, performance remains modest for both methods, and one method is not clearly better than the other, indicating that simply introducing parallel composition via disjoint partitions does not substantially improve the utility of private bagging when the dataset is imbalanced. One reason is that the disjoint splits further reduce the already limited number of minority-class examples available to each learner, exacerbating data imbalance and minority class scarcity.

**Stratified Disjoint Splits**   We also examined a variant of the disjoint-split approach that leverages *stratified sampling* to preserve minority-class representation in each partition. This setting presupposes that the practitioner can publicly observe class labels when constructing subsamples; thus, this setting is *not* directly comparable with privacy assumptions we make for other methods in this paper. Nonetheless, for comparison, we report results in Table 6 ($r = 8$) and Table 7 ($r = 32$).

While stratification can slightly improve the representation of minority samples in each split, performance still remains limited. The main challenge is that each weak learner's privacy noise is amplified by the small per-learner sample size, leading to weak models that do not effectively capture minority class signal.

**Soft vs. hard voting**  A second potential avenue for mitigating the degradation from private learning within bagged ensembles is to adopt different voting schemes when aggregating predictions. While hard voting uses a simple majority of discrete class predictions, soft voting first averages predicted probabilities across learners and then applies a threshold (see, e.g., (Bauer & Kohavi, 1999) for an analysis of such voting mechanisms in the non-private setting). Tables 8 (imbalance $r = 8$) and 9 ($r = 32$) show the effect of switching from hard voting to soft voting in the standard advanced composition bagging scheme (i.e. where each weak learner is bootstrapped). In some scenarios, soft voting slightly outperforms hard voting, but in other settings the results are mixed.

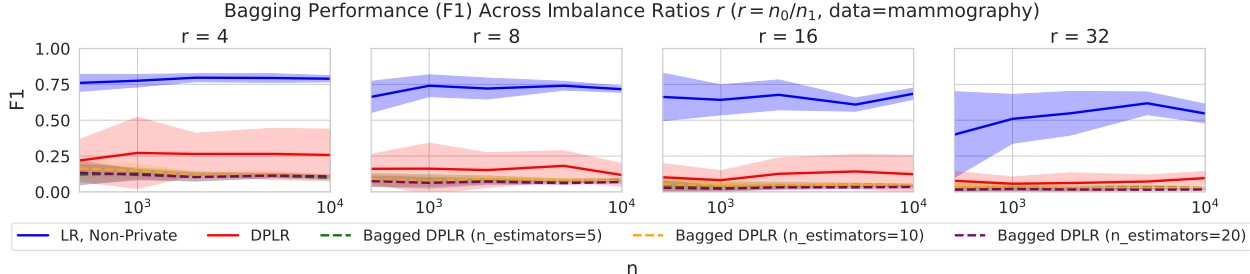

Figure 3: $F1$ score performance on subsamples of the *mammography* dataset (`imblearn`) comparing differentially private logistic regression (DPLR) and DP bagging (DPLR as weak learner). Data was subsampled (log-scale x-axis: $n \in [500, 1000, 2000, 5000, 10000]$) and evaluated across imbalance ratios $r \in [4, 8, 16, 32]$.

| Method | n=500 | n=1000 | n=2000 | n=5000 | n=10000 |
|---|---|---|---|---|---|
| LR, Non-Private | $0.662 \pm 0.109$ | $0.741 \pm 0.076$ | $0.721 \pm 0.073$ | $0.740 \pm 0.031$ | $0.716 \pm 0.022$ |
| Bagged DPLR, Adv. Comp. ($n_{\text{estimators}} = 5$) | $0.177 \pm 0.124$ | $0.178 \pm 0.115$ | $0.164 \pm 0.128$ | $0.094 \pm 0.048$ | $0.217 \pm 0.116$ |
| Bagged DPLR, Disjoint ($n_{\text{estimators}} = 5$) | $0.205 \pm 0.086$ | $0.207 \pm 0.099$ | $0.213 \pm 0.146$ | $0.170 \pm 0.087$ | $0.166 \pm 0.118$ |
| Bagged DPLR, Adv. Comp. ($n_{\text{estimators}} = 10$) | $\mathbf{0.273 \pm 0.197}$ | $0.138 \pm 0.093$ | $0.142 \pm 0.076$ | $0.201 \pm 0.066$ | $\mathbf{0.254 \pm 0.142}$ |
| Bagged DPLR, Disjoint ($n_{\text{estimators}} = 10$) | $0.113 \pm 0.078$ | $0.068 \pm 0.062$ | $\mathbf{0.249 \pm 0.190}$ | $0.121 \pm 0.076$ | $0.150 \pm 0.075$ |
| Bagged DPLR, Adv. Comp. ($n_{\text{estimators}} = 20$) | $0.150 \pm 0.153$ | $0.159 \pm 0.114$ | $0.160 \pm 0.120$ | $0.249 \pm 0.116$ | $0.186 \pm 0.193$ |
| Bagged DPLR, Disjoint ($n_{\text{estimators}} = 20$) | $0.189 \pm 0.179$ | $\mathbf{0.248 \pm 0.185}$ | $0.162 \pm 0.137$ | $\mathbf{0.263 \pm 0.116}$ | $0.253 \pm 0.119$ |

Table 4: Bagging performance with disjoint learners (uniform sampling), F1 score for imbalance ratio $r = 8$.

| Method | n=500 | n=1000 | n=2000 | n=5000 | n=10000 |
|---|---|---|---|---|---|
| LR, Non-Private | $0.400 \pm 0.299$ | $0.509 \pm 0.170$ | $0.548 \pm 0.152$ | $0.618 \pm 0.079$ | $0.547 \pm 0.065$ |
| Bagged DPLR, Adv. Comp. ($n_{\text{estimators}} = 5$) | $0.074 \pm 0.071$ | $0.069 \pm 0.065$ | $0.052 \pm 0.035$ | $0.036 \pm 0.024$ | $0.056 \pm 0.065$ |
| Bagged DPLR, Disjoint ($n_{\text{estimators}} = 5$) | $\mathbf{0.082 \pm 0.074}$ | $0.055 \pm 0.048$ | $0.068 \pm 0.045$ | $\mathbf{0.097 \pm 0.045}$ | $0.060 \pm 0.040$ |
| Bagged DPLR, Adv. Comp. ($n_{\text{estimators}} = 10$) | $0.011 \pm 0.033$ | $0.026 \pm 0.043$ | $0.048 \pm 0.029$ | $0.052 \pm 0.034$ | $0.064 \pm 0.055$ |
| Bagged DPLR, Disjoint ($n_{\text{estimators}} = 10$) | $0.034 \pm 0.081$ | $\mathbf{0.108 \pm 0.087}$ | $0.047 \pm 0.038$ | $0.096 \pm 0.059$ | $\mathbf{0.068 \pm 0.052}$ |
| Bagged DPLR, Adv. Comp. ($n_{\text{estimators}} = 20$) | $0.000 \pm 0.000$ | $0.000 \pm 0.000$ | $\mathbf{0.097 \pm 0.084}$ | $0.070 \pm 0.042$ | $0.062 \pm 0.062$ |
| Bagged DPLR, Disjoint ($n_{\text{estimators}} = 20$) | $0.000 \pm 0.000$ | $0.000 \pm 0.000$ | $0.063 \pm 0.055$ | $0.061 \pm 0.034$ | $0.065 \pm 0.049$ |

Table 5: Bagging performance with disjoint learners (uniform sampling), F1 score for imbalance ratio $r = 32$.

| Method | n=500 | n=1000 | n=2000 | n=5000 | n=10000 |
|---|---|---|---|---|---|
| LR, Non-Private | $0.662 \pm 0.109$ | $0.743 \pm 0.000$ | $0.712 \pm 0.000$ | $0.667 \pm 0.000$ | $0.714 \pm 0.000$ |
| DPLR | $0.291 \pm 0.110$ | $0.131 \pm 0.153$ | $0.254 \pm 0.168$ | $0.205 \pm 0.151$ | $0.157 \pm 0.092$ |
| Bagged DPLR, Adv. Comp. ($n_{estimators} = 5$) | $0.074 \pm 0.071$ | $0.069 \pm 0.065$ | $0.052 \pm 0.035$ | $0.036 \pm 0.024$ | $0.056 \pm 0.065$ |
| Bagged DPLR, Stratified ($n_{estimators} = 5$) | $0.154 \pm 0.084$ | $\mathbf{0.233 \pm 0.147}$ | $0.197 \pm 0.101$ | $\mathbf{0.249 \pm 0.122}$ | $0.168 \pm 0.083$ |
| Bagged DPLR, Adv. Comp. ($n_{estimators} = 10$) | $\mathbf{0.273 \pm 0.197}$ | $0.138 \pm 0.093$ | $0.142 \pm 0.076$ | $0.201 \pm 0.066$ | $\mathbf{0.254 \pm 0.142}$ |
| Bagged DPLR, Stratified ($n_{estimators} = 10$) | $0.220 \pm 0.101$ | $0.189 \pm 0.110$ | $0.167 \pm 0.098$ | $0.147 \pm 0.092$ | $0.192 \pm 0.108$ |
| Bagged DPLR, Adv. Comp. ($n_{estimators} = 20$) | $0.150 \pm 0.153$ | $0.159 \pm 0.114$ | $0.160 \pm 0.120$ | $\mathbf{0.249 \pm 0.116}$ | $0.186 \pm 0.193$ |
| Bagged DPLR, Stratified ($n_{estimators} = 20$) | $0.157 \pm 0.086$ | $0.149 \pm 0.083$ | $\mathbf{0.200 \pm 0.111}$ | $0.158 \pm 0.090$ | $0.205 \pm 0.080$ |

Table 6: Bagging performance with disjoint learners (stratified sampling), F1 score for imbalance ratio $r = 8$.

| Method | n=500 | n=1000 | n=2000 | n=5000 | n=10000 |
|---|---|---|---|---|---|
| LR, Non-Private | $0.400 \pm 0.299$ | $0.286 \pm 0.000$ | $0.375 \pm 0.000$ | $0.571 \pm 0.000$ | $0.473 \pm 0.000$ |
| DPLR | $0.086 \pm 0.055$ | $0.168 \pm 0.173$ | $0.064 \pm 0.061$ | $0.074 \pm 0.044$ | $0.070 \pm 0.074$ |
| Bagged DPLR, Adv. Comp. ($n_{estimators} = 5$) | $0.074 \pm 0.071$ | $0.069 \pm 0.065$ | $0.052 \pm 0.035$ | $0.036 \pm 0.024$ | $0.056 \pm 0.065$ |
| Bagged DPLR, Stratified ($n_{estimators} = 5$) | $\mathbf{0.058 \pm 0.049}$ | $0.084 \pm 0.069$ | $0.063 \pm 0.037$ | $0.077 \pm 0.057$ | $\mathbf{0.075 \pm 0.058}$ |
| Bagged DPLR, Adv. Comp. ($n_{estimators} = 10$) | $0.011 \pm 0.033$ | $0.026 \pm 0.043$ | $0.048 \pm 0.029$ | $0.052 \pm 0.034$ | $0.064 \pm 0.055$ |
| Bagged DPLR, Stratified ($n_{estimators} = 10$) | $0.050 \pm 0.063$ | $0.039 \pm 0.029$ | $0.067 \pm 0.044$ | $\mathbf{0.082 \pm 0.023}$ | $0.055 \pm 0.037$ |
| Bagged DPLR, Adv. Comp. ($n_{estimators} = 20$) | $0.000 \pm 0.000$ | $0.000 \pm 0.000$ | $0.097 \pm 0.084$ | $0.070 \pm 0.042$ | $0.062 \pm 0.062$ |
| Bagged DPLR, Stratified ($n_{estimators} = 20$) | $0.000 \pm 0.000$ | $\mathbf{0.087 \pm 0.050}$ | $\mathbf{0.087 \pm 0.044}$ | $0.064 \pm 0.051$ | $0.054 \pm 0.044$ |

Table 7: Bagging performance with disjoint learners (stratified sampling), F1 score for imbalance ratio $r = 32$.

| Method | n=500 | n=1000 | n=2000 | n=5000 | n=10000 |
|---|---|---|---|---|---|
| LR, Non-Private | $0.662 \pm 0.109$ | $0.741 \pm 0.076$ | $0.721 \pm 0.073$ | $0.740 \pm 0.031$ | $0.716 \pm 0.022$ |
| Bagged DPLR, Soft ($n_{estimators} = 5$) | $0.262 \pm 0.093$ | $0.220 \pm 0.127$ | $0.208 \pm 0.105$ | $0.206 \pm 0.143$ | $0.230 \pm 0.157$ |
| Bagged DPLR, Hard ($n_{estimators} = 5$) | $0.074 \pm 0.071$ | $0.069 \pm 0.065$ | $0.052 \pm 0.035$ | $0.036 \pm 0.024$ | $0.056 \pm 0.065$ |
| Bagged DPLR, Soft ($n_{estimators} = 10$) | $0.212 \pm 0.074$ | $0.226 \pm 0.098$ | $0.123 \pm 0.065$ | $0.238 \pm 0.179$ | $0.205 \pm 0.167$ |
| Bagged DPLR, Hard ($n_{estimators} = 10$) | $\mathbf{0.273 \pm 0.197}$ | $0.138 \pm 0.093$ | $0.142 \pm 0.076$ | $0.201 \pm 0.066$ | $\mathbf{0.254 \pm 0.142}$ |
| Bagged DPLR, Soft ($n_{estimators} = 20$) | $0.205 \pm 0.170$ | $\mathbf{0.265 \pm 0.177}$ | $\mathbf{0.280 \pm 0.114}$ | $0.127 \pm 0.071$ | $0.154 \pm 0.092$ |
| Bagged DPLR, Hard ($n_{estimators} = 20$) | $0.150 \pm 0.153$ | $0.159 \pm 0.114$ | $0.160 \pm 0.120$ | $\mathbf{0.249 \pm 0.116}$ | $0.186 \pm 0.193$ |

Table 8: Performance table for imbalance ratio $r = 8$, *Hard* vs. *Soft* voting.

| Method | n=500 | n=1000 | n=2000 | n=5000 | n=10000 |
|---|---|---|---|---|---|
| LR, Non-Private | $0.400 \pm 0.299$ | $0.509 \pm 0.170$ | $0.548 \pm 0.152$ | $0.618 \pm 0.079$ | $0.547 \pm 0.065$ |
| Bagged DPLR, Soft ($n_{estimators} = 5$) | $\mathbf{0.074 \pm 0.038}$ | $0.066 \pm 0.044$ | $0.060 \pm 0.046$ | $\mathbf{0.073 \pm 0.058}$ | $\mathbf{0.083 \pm 0.070}$ |
| Bagged DPLR, Hard ($n_{estimators} = 5$) | $\mathbf{0.074 \pm 0.071}$ | $0.069 \pm 0.065$ | $0.052 \pm 0.035$ | $0.036 \pm 0.024$ | $0.056 \pm 0.065$ |
| Bagged DPLR, Soft ($n_{estimators} = 10$) | $0.049 \pm 0.074$ | $0.047 \pm 0.054$ | $0.033 \pm 0.024$ | $0.064 \pm 0.038$ | $0.039 \pm 0.032$ |
| Bagged DPLR, Hard ($n_{estimators} = 10$) | $0.011 \pm 0.033$ | $0.026 \pm 0.043$ | $0.048 \pm 0.029$ | $0.052 \pm 0.034$ | $0.064 \pm 0.055$ |
| Bagged DPLR, Soft ($n_{estimators} = 20$) | $0.000 \pm 0.000$ | $\mathbf{0.085 \pm 0.156}$ | $0.019 \pm 0.028$ | $0.059 \pm 0.037$ | $\mathbf{0.083 \pm 0.036}$ |
| Bagged DPLR, Hard ($n_{estimators} = 20$) | $0.000 \pm 0.000$ | $0.000 \pm 0.000$ | $\mathbf{0.097 \pm 0.084}$ | $0.070 \pm 0.042$ | $0.062 \pm 0.062$ |

Table 9: Performance table for imbalance ratio $r = 32$, *Hard* vs. *Soft* voting.

## C.2. Warm-up: A Known Population

As a warm-up, we quantify the estimation error of the Bayes optimal classifier for a *known* Gaussian mixture.

**Example 11.** Let $\{X_i, y_i\}_{i=1}^n \in \mathbb{R}^{d=1} \times \{0, 1\}^n$ be randomly sampled such that $X$ is a mixture of Gaussians and $y$ is a binary class label. Specifically, let $\{X_i \mid y_i = 1\} \sim \mathcal{N}(\mu_1, \sigma^2)$ and $\{X_i \mid y_i = 0\} \sim \mathcal{N}(\mu_0, \sigma^2)$. The domain of $X$ here is *a priori* unbounded, but we can later bound $X$ with clipping to reduce sensitivity.

This setting was also studied in (Yang & Xu, 2020), who showed that the Bayes optimal classifier is given by $f_\theta(X) = \mathbb{I}(X \geq \theta)$ for $\theta = (\mu_0 + \mu_1)/2$ (see (Hart et al., 2000) for a textbook treatment). That is, assign the positive label if and only if $X > \theta$. We construct a private estimate of $\theta$ to build intuition for the effect of noise on imbalanced learning.

The private classification mechanism $\mathcal{M}_{BOC} : \mathbb{R} \mapsto \{0, 1\}$ makes private estimates of $\mu_0$, $\mu_1$ by first clipping each $X_i$ to lie in the range $[-R, R]$ before applying the Gaussian mechanism to the clipped data to compute the empirical mean.[4] Formally, define,

$$\hat{\mu}_b = \frac{1}{n_b} \sum_{i=1}^{n_b} \text{CLIP}(X_i, R) + \mathcal{N}\left(0, (\frac{2R}{n_b})^2 \cdot \frac{2 \log(\frac{1.25}{\delta})}{\epsilon^2}\right),$$

for $b \in \{0, 1\}$, where CLIP denotes clipping $X_i$ into the range $[-R, R]$. Then a natural mechanism for privately computing the Bayes Optimal Classifier is $\mathcal{M}_{BOC}(X) = \mathbb{I}(X \geq \hat{\theta})$ for $\hat{\theta} = \frac{\hat{\mu}_1 + \hat{\mu}_2}{2}$.

**Proposition 12.** *The mechanism $\mathcal{M}_{BOC}$ is $(2\epsilon, 2\delta)$-differentially private. Assume $\max\{|\mu_1|, |\mu_2|\} \leq B$ for some known bound $B$ and $R > B + \sigma\sqrt{2 \log(4n/\beta)}$. For any imbalance ratio $r \geq 1$, with probability at least $1 - \beta/2$, the $\hat{\theta}$ produced by $\mathcal{M}_{BOC}$ satisfies*

$$\left|\hat{\theta} - \theta\right| \leq 2\sqrt{\log(4/\beta)} \sqrt{\frac{\sigma^2}{n_0}(1 + r) + \frac{2R^2 \log(1.25/\delta)}{n_0^2 \epsilon^2} \cdot (1 + r^2)}.$$

*Furthermore, for any estimator $\tilde{\theta}$ of $\theta$, with probability at least $1 - \beta/2$,*

$$|\tilde{\theta} - \theta| \geq \sigma\sqrt{\frac{(1+r)}{n_0}} \Phi^{-1}(1 - \beta/2),$$

*where $\Phi(\cdot)$ denotes the cumulative distribution function of a standard normal distribution.*

The proof of this is relatively straightforward. As a sketch, privacy guarantees follow from the Gaussian Mechanism. For the accuracy guarantee, we first provide a high probability bound on the potential affects of clipping the data to $R$, and then provide a high-probability error bound accounting for the noise added to each of the $(\epsilon, \delta)$-differentially private estimates $\hat{\mu}_0$ and $\hat{\mu}_1$. The proof relies on known bounds for the population mean of $X$ (for example, if $X$ is a mixture of Gaussians over AGE, one could assume a minimum of 0 and a maximum of 120).

*Proof.* Recall our private mean estimates for each class are,

$$\hat{\mu}_0 = \frac{1}{n_0} \sum_{i \in \{Y_i = 0\}} \text{CLIP}(X_i, R) + G_0, \quad \hat{\mu}_1 = \frac{1}{n_1} \sum_{i \in \{Y_i = 1\}} \text{CLIP}(X_i, R) + G_1,$$

where $G_0 \sim \mathcal{N}(0, \sigma_{DP_0}^2)$, $G_1 \sim \mathcal{N}(0, \sigma_{DP_1}^2)$ and $\text{CLIP}(x, R) = \max\{-R, \min(x, R)\}$. This clipping function guarantees the sensitivity of our private mean computation function is $\Delta f = \frac{2R}{n_y}$. Thus, the Gaussian mechanism gives $(\epsilon, \delta)$-DP with variance $\sigma_{DP}^2 = \Delta f^2 \frac{2 \log(1.25/\delta)}{\epsilon^2} = \frac{8R^2 \log(1.25/\delta)}{n_y^2 \epsilon^2}$.

First, we'll argue that with our choice of $R$ with probability $1 - \beta/2$ clipping does not bias the data. To see this let $\{Z_i\}_{i=1}^n \stackrel{iid}{\sim} N(0, \sigma^2)$. We will use the following standard concentration result for the maximum of sub-Gaussian random variables to bound them.

**Lemma 13** ((Rigollet & Hütter, 2023)). *Let $Z_1, \dots, Z_n \stackrel{iid}{\sim} N(0, \sigma^2)$. Then, $\Pr(\max_{1 \leq i \leq n} |Z_i| > t) \leq 2n e^{-\frac{t^2}{2\sigma^2}}$.*

---

[4]This is the canonical private mean estimator, but we note that improved methods exist (Biswas et al., 2020; Kulesza et al., 2023).

Consecutively using the triangle inequality, Lemma 13, and $R > B + \sigma\sqrt{2\log(4n/\beta)}$ we see that,

$$\Pr[\max_{1 \le i \le n} |X_i| > R] \le \Pr[\max_{1 \le i \le n} |Z_i| + B > R] \le 2ne^{-\frac{(R-B)^2}{2\sigma^2}} \le \beta/2.$$

Therefore, with probability at least $1 - \beta/2$,

$$
\begin{aligned}
\hat{\theta} - \theta &= \frac{1}{2}\left( \frac{1}{n_0}\sum_{i \in \{Y_i=0\}} \mathrm{CLIP}(X_i, R) - \mu_0 + G_0 + \frac{1}{n_1}\sum_{i \in \{Y_i=1\}} \mathrm{CLIP}(X_i, R) - \mu_1 + G_1 \right) \\
&= \frac{1}{2}\left( \frac{1}{n_0}\sum_{i \in \{Y_i=0\}} (X_i - \mu_0) + G_0 + \frac{1}{n_1}\sum_{i \in \{Y_i=1\}} (X_i - \mu_1) + G_1 \right) \\
&\sim \frac{1}{2}\mathcal{N}\left( 0, \frac{\sigma^2}{n_0} + \frac{\sigma^2}{n_1} + \sigma_{DP_1}^2 + \sigma_{DP_0}^2 \right) \\
&= \frac{1}{2}\mathcal{N}\left( 0, \frac{\sigma^2}{n_0}(1+r) + \frac{2R^2\log(1.25/\delta)}{n_0^2\epsilon^2} \cdot (1+r^2) \right)
\end{aligned}
$$
(2)

Combining (2) with the fact that for $Z \sim \mathcal{N}(0, \nu^2)$ we have the inequality $\Pr(|Z| > t) \le 2e^{-\frac{t^2}{2\nu^2}}$ we conclude that with probability $1 - \beta$,

$$|\hat{\theta} - \theta| \le \sqrt{\frac{\sigma^2}{n_0}(1+r) + \frac{2R^2\log(1.25/\delta)}{n_0^2\epsilon^2} \cdot (1+r^2)}\sqrt{2\log(4/\beta)}.$$

This completes the utility proof of the proposed estimator.

Let's now turn to the question of the best achievable deviation. We note that in this example the MLE of $\theta$ is available in closed form, namely $\hat{\theta}_{\mathrm{MLE}} = (\bar{X}^0 + \bar{X}^1)/2$, where $\bar{X}^0 = \frac{1}{n_0}\sum_{i=1}^n (1 - y_i)X_i$ and $\bar{X}^1 = \frac{1}{n_1}\sum_{i=1}^n y_i X_i$. Furthermore $\hat{\theta}_{\mathrm{MLE}} \sim N(0, \frac{(1+r)\sigma^2}{4n_0})$ is the minimum variance unbiased estimator. It is well know that in this case the narrowest confidence interval for $\theta$ is $[\hat{\theta} - \sigma\sqrt{(1+r)/n_0}\Phi^{-1}(1 - \beta/2), \hat{\theta} + \sigma\sqrt{(1+r)/n_0}\Phi^{-1}(1 - \beta/2)]$. $\qquad\square$

Proposition 12 tells us that in Example 11, a private classifier from the ideal model class has privacy error that scales linearly in the class imbalance parameter $r$, which is minimized under no class imbalance. Unfortunately, imbalanced data often has $r \gg 1$; for example, when detecting spam on Twitter (Analytics, 2009), $r \approx 32$, and in the datasets used in Section 5 in our empirical evaluations, $r$ ranges between 8.6 and 130.

**Linking to Imbalanced Metrics**    A natural question, building on Proposition 12, is how we might weight samples when calculating $\theta$ to improve performance on imbalanced metrics, such as Recall, under this simple population model. To do this, we consider a re-weighted classifier, $f_{\theta_\gamma}$, where the weights $\gamma$ are tied to class prevalence. We can then reason about weights under this classifier and show, for example, that the true positive rate can be written as $\mathrm{TPR} = \Phi\left(\frac{(1-\gamma)(\mu_1-\mu_0)}{\sigma}\right)$. This analysis informs setting weights based on class prevalence estimates (e.g., $\gamma = 1/\Pr(y_i = 1)$) to better target imbalance-focused metrics like Recall.

**Formal Discussion**    Consider the re-weighted classifier $f_{\theta_\gamma}$ where $\theta_\gamma = \gamma\mu_1 + (1-\gamma)\mu_0$. Note that the optimal Bayes classifier is in this model class, and can be written $f_{\theta_{1/2}}(X)$. We will denote the Gaussian random variables for the majority and minority classes as $Z_0 \sim \mathcal{N}(\mu_0, \sigma_0^2)$ and $Z_1 \sim \mathcal{N}(\mu_1, \sigma_1^2)$ respectively, and the Gaussian random variable for the added, zero-centered noise for privacy as $Z \sim \mathcal{N}(0, \sigma^2)$.

Then, we can derive a population version of *true positive rate* under $f_{\theta_\gamma}$ as,

$$
\begin{aligned}
\mathrm{TPR} &= \Pr(f_{\theta_\gamma}(X) = 1 | Y = 1) \\
&= \Pr(Z_1 \ge \gamma\mu_1 + (1-\gamma)\mu_0) \\
&= \Phi(\frac{(1-\gamma)(\mu_1 - \mu_0)}{\sigma}).
\end{aligned}
$$

Our original imbalance ratio $r$ is a sample-specific estimate of the true population imbalance ratio, $r^* = \Pr(Y = 0)/\Pr(Y = 1)$. Here, we will consider $r^*$, alongside a population version of positive rate $\text{PR} = \Pr(Y = 1) = \frac{1}{1+r^*}$. This gives us insight into an exact form of the Recall metric for Example 11, which is $\frac{\text{TP}}{\text{P}} = \Phi(\frac{(1-\gamma)(\mu_1-\mu_0)}{\sigma})(1 + r^*)$. Note that Recall gets *worse* as the imbalance ratio increases. However taking $\gamma < 1/2$ *improves* Recall relative to the standard optimal Bayes classifier. One way to choose such a $\gamma$ is to take the inverse probability weight $\gamma = 1/\Pr(Y = 1) = 1 + r^*$. These population parameter considerations motivate an empirical weighted counterpart to $f_{\theta_{1/2}}(X)$. Such precise distributional knowledge is rarely known in practice and unverifiable under differential privacy (where the data cannot be accessed directly without noise mechanisms). Instead we will by default account for $r$ by setting weights inversely proportional to class prevalence in our weighted methods, as motivated by this reasoning and prior work (Chawla et al., 2004).

**Quantifying the Benefits of Re-weighting in Imbalanced Metrics**   Here we provide detailed calculations showing that the re-weighted classifier $f_{\theta_\gamma}$ outperforms the optimal Bayes classifier $f_{\theta_{1/2}}$ in various imbalanced metrics, summarized in Table 10. Recall that $f_{\theta_\gamma}$ is a threshold-based classifier defined as

$$f_{\theta_\gamma}(X) = \mathbb{I}(X \geq \theta_\gamma), \tag{3}$$

where $\theta_\gamma = \gamma\mu_1 + (1 - \gamma)\mu_0$. The parameter $\theta_\gamma$ is a weighted average of the means for each class, $\mu_1$ and $\mu_0$, with $\gamma$ controlling the weight given to each class.

| Metric | Formula |
|---|---|
| Recall (Re($\gamma$)) | $(1 + r^*) \cdot \Phi\left((1 - \gamma)\Delta\right)$ |
| Precision (Pre($\gamma$)) | $\frac{\Phi((1-\gamma)\Delta)}{\Phi((1-\gamma)\Delta)+(1+r^*)\cdot[1-\Phi(\gamma\Delta)]}$ |
| Balanced Accuracy (BA($\gamma$)) | $\frac{\Phi((1-\gamma)\Delta)+\Phi(\gamma\Delta)}{2}$ |
| F1 Score (F1($\gamma$)) | $\frac{\Phi((1-\gamma)\Delta)}{\Phi((1-\gamma)\Delta)+\frac{1}{2}[1-\Phi(\gamma\Delta)]}$ |

Table 10: Some imbalanced classification metrics, defined as functions of the imbalance weight parameter $\gamma$ for the reweighted classifier $f_{\theta_\gamma}$, where $\Delta = \frac{\mu_1-\mu_0}{\sigma}$ for ease of presentation.

**Recall Metric.**   The True Positive Rate (TPR) is defined as the probability that the classifier correctly identifies a positive instance. For a given $X$ sampled from the positive class ($Y = 1$), we have,

$$\begin{aligned}
\text{TPR} &= \Pr(f_{\theta_\gamma}(X) = 1 \mid Y = 1) \\
&= \Pr(Z_1 \geq \gamma\mu_1 + (1 - \gamma)\mu_0) \, .
\end{aligned}$$

Since $X \mid Y = 1$ is distributed as $\mathcal{N}(\mu_1, \sigma^2)$, we can standardize this normal variable as,

$$\text{TPR} = \Pr\left(\frac{X - \mu_1}{\sigma} \geq \frac{\gamma\mu_1 + (1 - \gamma)\mu_0 - \mu_1}{\sigma}\right).$$

Thus, the TPR can be written using the cumulative distribution function (CDF) of the standard normal, denoted here as $\Phi$, given

$$\text{TPR} = \Phi\left(\frac{(1 - \gamma)(\mu_1 - \mu_0)}{\sigma}\right) \, .$$

We define the population imbalance ratio $r^*$ as the ratio of the probability of the negative class to the probability of the positive class i.e.

$$r^* = \frac{\Pr(Y = 0)}{\Pr(Y = 1)}.$$

The total probability of positives (i.e. positive rate $PR$ is just

$$PR = \Pr(Y = 1) = \frac{1}{1 + r^*} \, .$$

Recall is defined as the ratio of true positives to the total actual positives, or

$$\text{Recall} = \frac{\text{TPR}}{\text{PR}} = \frac{\Phi\left(\frac{(1-\gamma)(\mu_1 - \mu_0)}{\sigma}\right)}{\frac{1}{1+r^*}} \, .$$

Simplifying gives,

$$\text{Recall} = \text{Re}(\gamma) = (1 + r^*) \cdot \Phi\left(\frac{(1 - \gamma)(\mu_1 - \mu_0)}{\sigma}\right) \, .$$

This shows that $\text{Re}(\gamma)$ decreases as the imbalance ratio $r^*$ increases, because the term $(1 + r^*)$ magnifies the effect of the Gaussian term.

This implies that $\text{Re}(\gamma)$ can be improved relative to the Bayes Optimal Classifier by choosing $\gamma < \frac{1}{2}$. This adjustment shifts the threshold $\theta_\gamma$ to be more inclusive of the positive class, thereby increasing the true positive rate.

**Precision Metric.** Precision is defined as the ratio of true positive rate to all positive predictions, or

$$\text{Precision} = \frac{\text{TPR}}{\text{TPR} + \text{FPR}},$$

where FPR denotes False Positive Rate.

False Positive Rate (FPR) are defined as the probability that the classifier incorrectly identifies a negative instance as positive. For $X$ sampled from the negative class ($Y = 0$),

$$
\begin{aligned}
\text{FPR} &= \Pr(f_{\theta_\gamma}(X) = 1 \mid Y = 0) \\
&= \Pr(Z_0 \geq \gamma\mu_1 + (1 - \gamma)\mu_0) \, .
\end{aligned}
$$

We can similarly standardize this normal variable:

$$\text{FPR} = \Pr\left(\frac{X - \mu_0}{\sigma} \geq \frac{\gamma\mu_1 + (1 - \gamma)\mu_0 - \mu_0}{\sigma}\right),$$

which simplifies to,

$$\text{FPR} = \Pr\left(Z \geq \frac{\gamma(\mu_1 - \mu_0)}{\sigma}\right),$$

where $Z \sim \mathcal{N}(0, 1)$. Thus, FPR can be written as:

$$\text{FPR} = 1 - \Phi\left(\frac{\gamma(\mu_1 - \mu_0)}{\sigma}\right) \, .$$

This yields

$$\text{Precision} = \text{Pre}(\gamma) = \frac{\text{TPR}}{\text{TPR} + \text{FPR}} = \frac{\Phi\left(\frac{(1-\gamma)(\mu_1 - \mu_0)}{\sigma}\right)}{\Phi\left(\frac{(1-\gamma)(\mu_1 - \mu_0)}{\sigma}\right) + (1 + r^*) \cdot \left[1 - \Phi\left(\frac{\gamma(\mu_1 - \mu_0)}{\sigma}\right)\right]} \, .$$

As $r^*$ increases, the FPR becomes larger, leading to a potential decrease in $\text{Pre}(\gamma)$. This result shows that in adjusting $\gamma$, better performance can be achieved on either $\text{Re}(\gamma)$ or $\text{Pre}(\gamma)$.

**Balanced Accuracy.** Balanced accuracy is defined as the average of TPR and TNR. We require the following explicit formulas for TPR and TNR, where TPR was previously defined for Recall:

$$\text{TPR} = \Phi\left(\frac{(1-\gamma)(\mu_1 - \mu_0)}{\sigma}\right),$$

and

$$\text{TNR} = \Pr\left(Z_0 \leq \theta_\gamma\right) = \Pr\left(Z \leq \gamma\left(\frac{\mu_1 - \mu_0}{\sigma}\right)\right) = \Phi\left(\frac{\gamma(\mu_1 - \mu_0)}{\sigma}\right).$$

Therefore, Balanced Accuracy is simply

$$\text{BA}(\gamma) = \frac{\Phi\left(\frac{(1-\gamma)(\mu_1-\mu_0)}{\sigma}\right) + \Phi\left(\frac{\gamma(\mu_1-\mu_0)}{\sigma}\right)}{2}.$$

**F1 Score.** F1 Score can be written as:

$$\text{F1 Score} = \frac{2 \cdot \text{TPR}}{2 \cdot \text{TPR} + \text{FPR} + \text{FNR}}.$$

The expressions for TPR, FPR, and FNR have been previously derived as follows:

$$\text{TPR} = \Phi\left(\frac{(1-\gamma)(\mu_1 - \mu_0)}{\sigma}\right),$$

$$\text{FPR} = 1 - \Phi\left(\frac{\gamma(\mu_1 - \mu_0)}{\sigma}\right),$$

$$\text{FNR} = 1 - \Phi\left(\frac{(1-\gamma)(\mu_1 - \mu_0)}{\sigma}\right).$$

Substituting these yields an expression for the F1 Score:

$$\text{F1}(\gamma) = \frac{2 \cdot \Phi\left(\frac{(1-\gamma)(\mu_1-\mu_0)}{\sigma}\right)}{2 \cdot \Phi\left(\frac{(1-\gamma)(\mu_1-\mu_0)}{\sigma}\right) + \left[1 - \Phi\left(\frac{\gamma(\mu_1-\mu_0)}{\sigma}\right)\right] + \left[1 - \Phi\left(\frac{(1-\gamma)(\mu_1-\mu_0)}{\sigma}\right)\right]}.$$

Simplifying yields:

$$\text{F1}(\gamma) = \frac{\Phi\left(\frac{(1-\gamma)(\mu_1-\mu_0)}{\sigma}\right)}{\Phi\left(\frac{(1-\gamma)(\mu_1-\mu_0)}{\sigma}\right) + \frac{1}{2}\left[1 - \Phi\left(\frac{\gamma(\mu_1-\mu_0)}{\sigma}\right)\right]}.$$

## C.3. Weighted Private ERMs

**Assumptions from (Giddens et al., 2023).** We list here (for completeness) the undesirable assumptions from (Giddens et al., 2023) that we overcome. Their privacy proof works only for loss functions that take in a single argument, which excludes standard models like logistic regression, SVM, and others. Additionally, they made the assumption that the difference of weights across neighboring datasets goes to 0 as $n \to \infty$, which is too strong for our inverse proportional weights strategy. We also note that in differential privacy, sensitivity is analyzed under worst case assumptions even if the influence of a single data point diminishes as $n$ grows large. One therefore should avoid privacy statements that rely on asymptotic assumptions.

**Notation for ERM Proof.** For parity and ease of comparison, we will use mostly overlapping notation with (Chaudhuri et al., 2011). We will denote the Euclidean norm of $\mathbf{x} \in \mathbb{R}^d$ by $\|\mathbf{x}\|_2$. For an integer $n$, the notation $[n]$ will represent the set $\{1, 2, \ldots, n\}$. Boldface will be used for vectors, and calligraphic type for sets. For a square matrix $A$, the induced $L_2$-norm will be indicated by $\|A\|_2$. Algorithms will accept as input *training data* $\mathcal{D} = (\mathbf{x}_i, \mathbf{y}_i) \in \mathcal{X} \times \mathcal{Y} : i = 1, 2, \ldots, n$, consisting of $n$ data-label pairs. In binary classification, the data space is $\mathcal{X} = \mathbb{R}^d$ and the label set is $\mathcal{Y} = 0, 1$. It will be assumed throughout that $\mathcal{X}$ is the unit ball, hence $\|\mathbf{x}_i\|_2 \leq 1$. Note that the extension of the proof to $\|\mathbf{x}_i\|_2 \leq q$ is straightforward and commonly implemented in practice. This is also how we implemented our code.

We aim to construct a *predictor* $\mathbf{f} : \mathcal{X} \to \mathcal{Y}$. The quality of our predictor on the training data is assessed using a nonnegative *loss function* $\ell : \mathcal{Y} \times \mathcal{Y} \to \mathbb{R}$. In regularized empirical risk minimization (ERM), we select a predictor $\mathbf{f}$ that minimizes the regularized empirical loss, optimizing over $\mathbf{f}$ within a hypothesis class $\mathcal{H}$. The regularizer $\lambda N(\mathbf{f})$ is used to prevent over-fitting, for some function $N$ of the predictor. Altogether, this yields the ERM loss function:

$$J(\mathbf{f}, \mathcal{D}) = \frac{1}{n} \sum_{i=1}^{n} \ell(\mathbf{y}_i, \mathbf{f}(\mathbf{x}_i)) + \lambda N(\mathbf{f}) .$$

We can slightly modify the regularized ERM by introducing a weighting scheme to correct for class imbalance. Let $\mathbf{w} = [w_1, w_2, \ldots, w_n]$ be a vector of sample weights, where each $w_i$ corresponds to a weight assigned to the $i$-th sample in the dataset $\mathcal{D}$. This yields,

$$J(\mathbf{f}, \mathcal{D}, \mathbf{w}) = \frac{1}{n} \sum_{i=1}^{n} w_i \cdot \ell(\mathbf{y}_i, \mathbf{f}(\mathbf{x}_i)) + \lambda N(\mathbf{f}).$$

We consider weights $w_i$ that do not explicitly affect the regularization term $\lambda N(\mathbf{f})$, as is standard, as regularization should penalize model complexity independent of class imbalance or weighting.

**Ridge Regression.** From here on, we will focus on *ridge regression*, so instead of a penalty of the form $\lambda N(\mathbf{f})$ we will use $\frac{\lambda}{2} \|\boldsymbol{\beta}\|_2^2$, where our predictor is $\mathbf{x}^T \boldsymbol{\beta}$ and $\boldsymbol{\beta}$ is a vector of coefficients that can be multiplied with a sample vector $\mathbf{x}$ to produce a prediction.

A common choice of weight vector $\mathbf{w} = [w_1, w_2, \ldots, w_n]$ is to compose weights such that they correspond to the inverse frequency of the class label in the training set (Provost & Fawcett, 1997). In other words, $w_i$ is inversely proportional to the prevalence of the class label $\mathbf{y}_i$ associated with each sample $(\mathbf{x}_i, \mathbf{y}_i)$. Let $n$ be the number of total samples in dataset $D$, and $Y$ be the set of unique class labels. Then let $\hat{\pi}_k = \frac{1}{n} \sum_{i=1}^{n} \mathbb{I}[y_i = k]$, $\hat{\pi}^{-1} = (\hat{\pi}_0^{-1}, \hat{\pi}_1^{-1})$ and define the weights

$$w_k = \frac{\|\hat{\pi}^{-1}\|_1^{-1}}{\pi_k} \quad \text{for } k \in \{0, 1\}. \tag{4}$$

**Considering neighboring datasets and class weights** Consider $\mathcal{D} = \{(x_i, y_i)\}_{i=1}^{n} = \{\mathbf{x}, \mathbf{y}\}$ (where $\mathbf{x}$ and $\mathbf{y}$ are the feature and label vectors, respectively), and where $y_i \in \{0, \ldots, k\}$, and a class weight $w_i$ as defined above.

Recall $n_1 \ll n_0$. We then have weights,

$$w_i = \begin{cases} \frac{n}{n_1} \left( \frac{n}{n_1} + \frac{n}{n_0} \right)^{-1} & \text{if } i \in I_1 \\ \frac{n}{n_0} \left( \frac{n}{n_1} + \frac{n}{n_0} \right)^{-1} & \text{if } i \in I_0 \end{cases}$$

where $I_0$ and $I_1$ are the set of row indexes of $\mathbf{x}$ that partitions it into $\mathbf{x}^{(0)}$ and $\mathbf{x}^{(1)}$ (denoting the majority and minority class feature vectors, respectively). Note that $\frac{n}{n_1} = \frac{1}{\hat{\pi}_1}$, and thus $(\frac{n}{n_1} + \frac{n}{n_0})^{-1} = (\frac{1}{\hat{\pi}_1} + \frac{1}{\hat{\pi}_0})^{-1}$.

Recall that for privacy, we consider neighboring dataset $\mathbf{x}'$ of size $n = n_0 + n_1$. We will denote the new weights of neighboring dataset features $\mathbf{x}'$ by $\{w_i'\}_{i=1}^n$.

We note that neighboring datasets will have weights that fall under one of two cases below. Without loss of generality we will assume they differ in the $j$th data point.

**C1:** The differing sample between the neighboring sets is such that $x_j' \neq x_j$, but keeps the same label i.e. $y_j' = y_j$. This implies that $w_i - w_i' = 0 \ \forall \, i = 1, \ldots, n$. Under this case, there is no change in weights between neighboring datasets $\mathcal{D}$ and $\mathcal{D}'$.

**C2:** The differing sample between the neighboring sets is such that $x_j' \neq x_j$ and also $y_j' \neq y_j$. In this case we need to reason about the effect this difference entails on the class weights.

In **Case 2**, it follows that,

$$\frac{1}{\hat{\pi}_1'} + \frac{1}{\hat{\pi}_1} = \begin{cases} \frac{n}{n_1-1} - \frac{n}{n_1} = \frac{n}{n_1(n_1-1)} & \text{if } I_1 \subset I_1' \\ \frac{n}{n_1+1} - \frac{n}{n_1} = \frac{n}{n_1(n_1+1)} & \text{if } I_1' \subset I_1 \end{cases} ,$$

where if $I_1' \subset I_1$ means that the minority class gets even smaller between the two neighboring datasets. The complementary scenario is,

$$\frac{1}{\hat{\pi}_0'} + \frac{1}{\hat{\pi}_0} = \begin{cases} \frac{n}{n_0+1} - \frac{n}{n_0} = \frac{-n}{n_0(n_0+1)} & \text{if } I_1 \subset I_1' \\ \frac{n}{n_0-1} - \frac{n}{n_0} = \frac{n}{n_0(n_0-1)} & \text{if } I_1' \subset I_1 \end{cases} .$$

Since by assumption $n_1 \ll n_0$, we have that $\frac{1}{\hat{\pi}_1} = \frac{n}{n_1} \gg \frac{n}{n_0} = \frac{1}{\hat{\pi}_0}$. We analyze these two scenarios independently.

**Scenario 1**: when $I_1' \subset I_1$ i.e. the minority becomes smaller. Then we have,

$$\frac{1}{\hat{\pi}_1'} + \frac{1}{\hat{\pi}_0'} = n\left(\frac{1}{n_1-1} + \frac{1}{n_0+1}\right) = \frac{n(n_1+n_0)}{(n_0+1)(n_1-1)} = \frac{n^2}{(n_0+1)(n_1-1)},$$

So,

$$w_i' = \begin{cases} \frac{n}{n_1-1} \frac{(n_1-1)(n_0+1)}{n^2}, & \text{if } i \in I_1, \\ \frac{n}{n_0+1} \frac{(n_1-1)(n_0+1)}{n^2}, & \text{if } i \notin I_1, \end{cases}$$

while $\frac{1}{\hat{\pi}_1} + \frac{1}{\hat{\pi}_0} = n\left(\frac{1}{n_1} + \frac{1}{n_0}\right) = \frac{n^2}{n_1 n_0}$, and

$$w_i = \begin{cases} \frac{n}{n_1} \cdot \frac{n_1 n_0}{n^2}, & \text{if } i \in I_1, \\ \frac{n}{n_0} \cdot \frac{n_1 n_0}{n^2}, & \text{if } i \in I_0. \end{cases}$$

Therefore,

$$\sum_{i \neq j} |w_i - w_i'| = \sum_{i \neq j, i \in I_1'} |w_i - w_i'| + \sum_{i \neq j, i \notin I_1'} |w_i - w_i'|$$

$$= (n_1 - 1)\left|\frac{n_0}{n} - \frac{n_0+1}{n}\right| + n_0 \left|\frac{n_1}{n} - \frac{n_1-1}{n}\right|$$

$$= \frac{n_1 - 1}{n} + \frac{n_0}{n} = 1 - \frac{1}{n}$$

In **Scenario 2**, we have $I_1 \subset I_1'$ i.e. (minority becomes larger). Thus,

$$\frac{1}{\hat{\pi}_1'} + \frac{1}{\hat{\pi}_0'} = \frac{n}{n_1+1} + \frac{n}{n_0-1} = \frac{n^2}{(n_1+1)(n_0-1)},$$

, and

$$w'_i = \begin{cases} \frac{n}{n_1+1} \frac{(n_1+1)(n_0-1)}{n^2} = \frac{n_0-1}{n}, & \text{if } i \in I'_1, \\[2mm] \frac{n}{n_0-1} \frac{(n_0-1)(n_1+1)}{n^2} = \frac{n_1+1}{n}, & \text{if } i \notin I'_1. \end{cases}$$

Therefore,

$$\sum_{i \neq j} |w_i - w'_i| = n_1 \cdot \left| \frac{n_0-1}{n} - \frac{n_0}{n} \right| + (n_0 - 1) \cdot \left| \frac{n_1+1}{n} - \frac{n_1}{n} \right|$$

$$= \frac{n_1 + n_0 - 1}{n} = 1 - \frac{1}{n}$$

We have thus shown that the total change in weights, if the minority class size and label proportions change by one, is bounded as $\sum_{i \neq j}^{n} |w_i - w'_i| \leq 1 - \frac{1}{n}$. We give this intermediate result in Lemma 14.

**Lemma 14.** *Consider $\mathcal{D} = \{(x_i, y_i)\}_{i=1}^{n} = \{\mathbf{x}, \mathbf{y}\}$, where $y_i \in \{0, 1\}$ and a neighboring data set $\mathcal{D}'$ differing in the jth data point. Then, for the weights defined in* (4) *we have that $\sum_{i \neq j} |w_i - w'_i| \leq 1 - \frac{1}{n}$.*

Now, for completeness, we reproduce standard definitions in the form they appear in (Chaudhuri et al., 2011), including a slightly stronger variation of Definition 1 then what is described in Section 2.

**Assumptions on loss.** We make almost the same loss assumptions as (Chaudhuri et al., 2011). Here, we restate definitions of *strictly convex* and *$\tau$-strongly convex* from their paper for convenience. We also require the convex loss function $\ell(\cdot, \cdot)$ to be **twice** differentiable functions with respect to the second argument, and such that $|\frac{\partial}{\partial \eta} \ell(y, \eta)| \leq 1$ and $|\frac{\partial^2}{\partial \eta^2} \ell(y, \eta)| \leq c$ for some fixed $c$.

**Definition 15.** A function $H(\boldsymbol{\beta})$ over $\boldsymbol{\beta} \in \mathbb{R}^d$ is *strictly convex* if for all $\alpha \in (0, 1)$, $\boldsymbol{\beta}$, and $\boldsymbol{\beta}'$,

$$H\left(\alpha \boldsymbol{\beta} + (1 - \alpha)\boldsymbol{\beta}'\right) < \alpha H(\boldsymbol{\beta}) + (1 - \alpha)H(\boldsymbol{\beta}').$$

It is *$\tau$-strongly convex* if for all $\alpha \in (0, 1)$, $\boldsymbol{\beta}$, and $\boldsymbol{\beta}'$,

$$H\left(\alpha \boldsymbol{\beta} + (1 - \alpha)\boldsymbol{\beta}'\right) \leq \alpha H(\boldsymbol{\beta}) + (1 - \alpha)H(\boldsymbol{\beta}') - \frac{1}{2}\tau\alpha(1 - \alpha) \left\|\boldsymbol{\beta} - \boldsymbol{\beta}'\right\|_2^2.$$

**Privacy model.** Assume $\mathcal{A}(\mathcal{D})$ generates a classifier, and let $\mathcal{D}'$ be a dataset that differs from $\mathcal{D}$ in one entry (assumed to be the private value of one individual). They are neighboring datasets in the standard sense, e.g. $\mathcal{D}'$ and $\mathcal{D}$ share $n - 1$ points $(\mathbf{x}_i, y_i)$. The algorithm $\mathcal{A}$ ensures DP if, for any set $\mathcal{S}$, the probability that $\mathcal{A}(\mathcal{D}) \in \mathcal{S}$ is close to the probability that $\mathcal{A}(\mathcal{D}') \in \mathcal{S}$, with the probability taken over the randomness in the algorithm.

**Definition 16.** An algorithm $\mathcal{A}(\mathcal{B})$ taking values in a set $\mathcal{T}$ provides $\epsilon$-DP if

$$\sup_{\mathcal{S} \subseteq \mathcal{T}} \sup_{\mathcal{D}, \mathcal{D}'} \frac{\mu\left(\mathcal{S} \mid \mathcal{B} = \mathcal{D}\right)}{\mu\left(\mathcal{S} \mid \mathcal{B} = \mathcal{D}'\right)} \leq e^{\epsilon},$$

where the first supremum is over all measurable $\mathcal{S} \subseteq \mathcal{T}$, the second is over all datasets $\mathcal{D}$ and $\mathcal{D}'$ differing in a single entry, and $\mu(\cdot | \mathcal{B})$ is the conditional distribution (measure) on $\mathcal{T}$ induced by the output $\mathcal{A}(\mathcal{B})$ given a dataset $\mathcal{B}$. The ratio is interpreted to be 1 whenever the numerator and denominator are both 0.

We also restate sensitivity, as it appears in (Chaudhuri et al., 2011). Consider $g : (\mathbb{R}^m)^n \to \mathbb{R}$, a scalar function of $z_1, \ldots, z_n$, where each $z_i \in \mathbb{R}^m$ represents the private value of individual $i$; the sensitivity of $g$ is defined as follows.

**Definition 17.** The sensitivity of a function $g : (\mathbb{R}^m)^n \to \mathbb{R}$ is the maximum change in the value of $g$ when one entry of the input database changes. More formally, the sensitivity $S(g)$ of $g$ is defined as:

$$S(g) = \max_{i \in [n]} \max_{z_1, \ldots, z_n, z'_i} |g(z_1, \ldots, z_{i-1}, z_i, z_{i+1}, \ldots, z_n) - g(z_1, \ldots, z_{i-1}, z'_i, z_{i+1}, \ldots, z_n)|.$$

For the function $A(\mathcal{D}) = \arg\min J(\boldsymbol{\beta}, \mathcal{D})$, the output is a vector $A(\mathcal{D}) + \mathbf{b}$, where $\mathbf{b}$ is random noise with a density of $\nu(\mathbf{b}) = \frac{1}{\alpha} e^{-\gamma \|\mathbf{b}\|}$, where $\alpha$ is the normalizing constant. The parameter $\gamma$ depends on $\epsilon$ and the $L_2$-*sensitivity* of $A(\cdot)$.

**Definition 18.** The $L_2$-sensitivity of a vector-valued function is defined as the maximum change in the $L_2$ norm of the value of $g$ when one entry of the input database changes. More formally,

$$S(A) = \max_i \max_{z_1,\ldots,z_n,z_i'} \|A(z_1,\ldots,z_i,\ldots) - A(z_1,\ldots,z_i',\ldots)\|_2.$$

**Objective perturbation.** The approach to private ERM first proposed by (Chaudhuri et al., 2011) adds noise to the objective function itself and then produces the minimizer of the perturbed objective. The perturbed objective is:

$$J_{\text{priv}}(\boldsymbol{\beta}, \mathcal{D}) = J(\boldsymbol{\beta}, \mathcal{D}) + \frac{1}{n}\mathbf{b}^T\boldsymbol{\beta},$$

Note that the privacy parameter here does not depend on the sensitivity of the of the classification algorithm. That is, the privacy parameter $\epsilon$ is determined by the amount of noise added to the objective function through $\frac{1}{n}\mathbf{b}^T\boldsymbol{\beta}$, and it depends on the properties of the loss function and the regularizer rather than on the sensitivity of the classification algorithm's output. With the addition of a weight vector $\mathbf{w}$, this is perturbed objective becomes:

$$J_{\text{priv}}(\boldsymbol{\beta}, \mathcal{D}, \mathbf{w}) = J(\boldsymbol{\beta}, \mathcal{D}, \mathbf{w}) + \frac{1}{n}\mathbf{b}^T\boldsymbol{\beta},$$

### C.3.1. PRIVACY OF ALGORITHM 1

In this section, we show that Algorithm 1 using the weighted ERM objective function $J_{\text{priv}}(\boldsymbol{\beta}, \mathcal{D}, \mathbf{w})$ is $\epsilon$-differentially private. e.g. the output of the weighted $J_{\text{priv}}(\boldsymbol{\beta}, \mathcal{D}, \mathbf{w})$ is $(\epsilon, 0)$-differentially private. We assume for each $w_i \in \mathbf{w}$, $|w_i| \leq 1$. Note in particular that our analysis covers the case of *logistic regression*, which as stated, (Chaudhuri et al., 2011) does not. Still, much of what follows is a adapted directly from the proof given by (Chaudhuri et al., 2011), with careful accounting for the weights vector $\mathbf{w}$; for sake of completeness and ease of comparison, all steps are stated as closely as possible to what appears in the prior work.

**Theorem 5.** *Algorithm 1 instantiated with a loss function $\ell(y, \eta)$ that is convex and twice differentiable with respect to $\eta$, with $|\frac{\partial}{\partial\eta}\ell(y,\eta)| \leq 1$ and $|\frac{\partial^2}{\partial\eta^2}\ell(y,\eta)| \leq c$ for all $y$, is $\epsilon$-differentially private.*

*Proof.* Consider $\boldsymbol{\beta}_{priv}$ output by Algorithm 1. Note,

$$\beta_{\text{priv}} = \arg\min_{\beta} \left\{ \underbrace{\frac{1}{n}\sum_{i=1}^{n} w_i\ell\left(y_i, \mathbf{x}_i^T\boldsymbol{\beta}\right) + \frac{1}{2}(\lambda + \Delta)\|\boldsymbol{\beta}\|_2^2 +}_{\text{weighted-ERM objective}} \underbrace{\frac{1}{n}\mathbf{b}^T\boldsymbol{\beta}}_{\text{objective perturbation}} \right\} \tag{5}$$

We observe that given *any* fixed $\boldsymbol{\beta}_{priv}$ and a fixed dataset $\mathcal{D}$, there always exists a $\mathbf{b}$ such that Algorithm 1 outputs $\boldsymbol{\beta}_{priv}$ on input $\mathcal{D}$. Because $\ell$ is differentiable and convex, and $N(\cdot)$ is differentiable, we can take the gradient of the objective function and set it to $\mathbf{0}$ at $\boldsymbol{\beta}_{priv}$. Therefore, we set

$$\begin{aligned} 0 &= \nabla J_{\text{priv}}(\boldsymbol{\beta}_{priv}, \mathcal{D}, \mathbf{w}) \\ &= \nabla J(\boldsymbol{\beta}_{priv}, \mathcal{D}, \mathbf{w}) + \frac{1}{n}\mathbf{b} + \Delta\boldsymbol{\beta}_{priv} \\ &= \frac{1}{n}\sum_{i=1}^{n} w_i \cdot \nabla\ell(y_i, \mathbf{x}_i^T\boldsymbol{\beta}_{priv}) + (\lambda + \Delta)\boldsymbol{\beta}_{priv} + \frac{1}{n}\mathbf{b}, \end{aligned}$$

and therefore

$$\mathbf{b} = -\sum_{i=1}^{n} w_i \cdot \ell'(y_i, \mathbf{x}_i^T\boldsymbol{\beta}_{priv})\mathbf{x}_i - n(\lambda + \Delta)\boldsymbol{\beta}_{priv}. \tag{6}$$

We claim that as $\ell$ is *twice* differentiable and $J(\boldsymbol{\beta}, \mathcal{D}) + \frac{\Delta}{2}\|\boldsymbol{\beta}\|_2^2$ is strongly convex, given a dataset $\mathcal{D} = (\mathbf{x}_1, y_1), \ldots, (\mathbf{x}_n, y_n)$, there is a bijection between $\mathbf{b}$ and $\boldsymbol{\beta}_{priv}$. Equation (6) shows that two different $\mathbf{b}$ values cannot result in the same $\boldsymbol{\beta}_{priv}$. Furthermore, since the objective is strictly convex, for a fixed $\mathbf{b}$ and $\mathcal{D}$, there is a unique $\boldsymbol{\beta}_{priv}$;

therefore the map from $\mathbf{b}$ to $\boldsymbol{\beta}_{priv}$ is injective. The relation Equation (6) also shows that for any $\boldsymbol{\beta}_{priv}$, there exists a $\mathbf{b}$ for which $\boldsymbol{\beta}_{priv}$ is the minimizer, so the map from $\mathbf{b}$ to $\boldsymbol{\beta}_{priv}$ is surjective.

To show $\epsilon$-DP, we need to compute the ratio $g(\boldsymbol{\beta}_{priv}|\mathcal{D})/g(\boldsymbol{\beta}_{priv}|\mathcal{D}')$ of the densities of $\boldsymbol{\beta}_{priv}$ under the two datasets $\mathcal{D}$ and $\mathcal{D}'$. This ratio can be written as:

$$\frac{g(\boldsymbol{\beta}_{priv}|\mathcal{D})}{g(\boldsymbol{\beta}_{priv}|\mathcal{D}')} = \frac{\mu(\mathbf{b}|\mathcal{D})}{\mu(\mathbf{b}'|\mathcal{D}')} \cdot \frac{|\det(\mathbf{J}(\boldsymbol{\beta}_{priv} \to \mathbf{b}|\mathcal{D}))|^{-1}}{|\det(\mathbf{J}(\boldsymbol{\beta}_{priv} \to \mathbf{b}'|\mathcal{D}'))|^{-1}},$$

where $\mathbf{J}(\boldsymbol{\beta}_{priv} \to \mathbf{b}|\mathcal{D})$, $\mathbf{J}(\boldsymbol{\beta}_{priv} \to \mathbf{b}|\mathcal{D}')$ are the Jacobian matrices of the mappings from $\boldsymbol{\beta}_{priv}$ to $\mathbf{b}$, and $\mu(\mathbf{b}|\mathcal{D})$ and $\mu(\mathbf{b}|\mathcal{D}')$ are the densities of $\mathbf{b}$ given the output $\boldsymbol{\beta}_{priv}$, when the datasets are $\mathcal{D}$ and $\mathcal{D}'$ respectively.

First, we bound the ratio of the Jacobian determinants. Let $\mathbf{b}^{(j)}$ denote the $j$-th coordinate of $\mathbf{b}$. From Equation (6) we have,

$$\mathbf{b}^{(j)} = -\sum_{i=1}^{n} w_i \cdot \ell'(y_i, \boldsymbol{\beta}_{priv}^T \mathbf{x}_i)\mathbf{x}_i^{(j)} - n(\lambda + \Delta)\boldsymbol{\beta}_{priv}^{(j)} .$$

Given a dataset $\mathcal{D}$, the $(j, k)$-th entry of the Jacobian matrix $\mathbf{J}(\mathbf{f} \to \mathbf{b}|\mathcal{D})$ is

$$\frac{\partial \mathbf{b}^{(j)}}{\partial \boldsymbol{\beta}_{priv}^{(k)}} = -\sum_i w_i \cdot \ell''(y_i, \boldsymbol{\beta}_{priv}^T \mathbf{x}_i)\mathbf{x}_i^{(j)}\mathbf{x}_i^{(k)} - n(\lambda + \Delta)\mathbb{I}(j = k),$$

where $\mathbb{I}(\cdot)$ is the indicator function. We note that the Jacobian is defined for all $\boldsymbol{\beta}_{priv}$ because $\|\boldsymbol{\beta}\|_2^2$ and $\ell$ are globally twice differentiable.

Let $\mathcal{D}$ and $\mathcal{D}'$ be two datasets which differ in the value of the $n$-th item such that $\mathcal{D} = \{(\mathbf{x}_1, y_1), \ldots, (\mathbf{x}_{n-1}, y_{n-1}), (\mathbf{x}_n, y_n)\}$ and $\mathcal{D}' = \{(\mathbf{x}_1, y_1), \ldots, (\mathbf{x}_{n-1}, y_{n-1}), (\mathbf{x}'_n, y'_n)\}$. Moreover, we define matrices $A$ and $E$ as follows

$$A = n\lambda I + \sum_{i=1}^{n} w_i \cdot \ell''(y_i, \boldsymbol{\beta}_{priv}^T \mathbf{x}_i)\mathbf{x}_i\mathbf{x}_i^T + n\Delta I_d$$

$$E = \sum_{i=1}^{n} \left[ -w_i \, \ell''(y_i, \boldsymbol{\beta}_{priv}^T \mathbf{x}_i)\mathbf{x}_i\mathbf{x}_i^T + w'_i \, \ell''(y'_i, \boldsymbol{\beta}_{priv}^T \mathbf{x}'_i)\mathbf{x}'_i\mathbf{x}'^T_i \right]$$

Then, $\mathbf{J}(\boldsymbol{\beta}_{priv} \to \mathbf{b}|\mathcal{D}) = -A$, and $\mathbf{J}(\boldsymbol{\beta}_{priv} \to \mathbf{b}|\mathcal{D}') = -(A + E)$.

We now account for the fact that some $w_i$ and $y_i$ may change across neighboring datasets. Consider each summand inside $E$. If $i$ is not the one that changed (i.e., $i \neq n$), then $y'_i = y_i$ and $\mathbf{x}'_i = \mathbf{x}_i$. The change in that term is purely from the difference in weights $w_i$ vs. $w'_i$. Since $\|\mathbf{x}_i\|_2 \leq 1$ and $\ell''(\cdot) \leq c$, one gets $\|(w'_i)^2\ell''(\ldots)\mathbf{x}_i\mathbf{x}_i^T - w_i\ell''(\ldots)\mathbf{x}_i\mathbf{x}_i^T\|_2 \leq 2c|w'_i - w_i|$. By Lemma 14, summing over all such $i \neq n$ yields a total $\leq 2c\sum_{i\neq n}|w'_i - w_i| \leq 2c(1 - \frac{1}{n})$. For $i = n$ (the changed point), the difference can alter $\mathbf{x}_n$ vs. $\mathbf{x}'_n$ and $\ell''$, but still, the norm is at most $2c$. Adding them up gives

$$\|E\|_2 \leq 2c\left(1 - \frac{1}{n}\right) + 2c \leq 4c.$$

Consider,

$$\frac{|\det(\mathbf{J}(\boldsymbol{\beta}_{priv} \to \mathbf{b}|\mathcal{D}'))|}{|\det(\mathbf{J}(\boldsymbol{\beta}_{priv} \to \mathbf{b}|\mathcal{D}))|} = \frac{|\det(A + E)|}{|\det A|}$$

By sub-multiplicativity of the spectral norm, and recalling $\det(A + E) = \exp\left(\mathbf{tr}\log(I + A^{-1}E)\right)$, we can get,

$$\left|\frac{\det(A+E)}{\det A}\right| = \left|\det(I + A^{-1}E)\right| \leq \exp\left(\mathrm{tr}(A^{-1}E)\right) \leq \exp\left(\frac{4cd}{n(\lambda + \Delta)}\right).$$

Thus, we will define $\epsilon'' = \frac{4cd}{n(\lambda+\Delta)}$, and set $\epsilon' = \epsilon - \epsilon''$. Hence $\frac{|\det(A+E)|}{|\det(A)|} \leq e^{\epsilon''}$.

Next, we bound the ratio of the densities of $\mathbf{b}$. Recall that $|\ell'(z)| \leq 1$, for any $z$ and $|w_i|, |y_i|, \|\mathbf{x}_i\| \leq 1$, and Lemma 14 shows that $\sum_{i \neq j} |w_i - w_i'| \leq 1 - \frac{1}{n}$, so we have that,

$$\|\mathbf{b}' - \mathbf{b}\|_2 = \left\| w_j \ell'(y_j, x_j^T \beta_{\text{priv}}) x_j - w_j' \ell'(y_j, x_j'^T \beta_{\text{priv}}) x_j' + \sum_{i \neq j} (w_i - w_i') \ell'(y_i, x_i^T \beta_{\text{priv}}) x_i \right\|_2$$

$$\leq 2 + \sum_{i \neq j} |w_i - w_i'| = 2 + 1 - \frac{1}{n}.$$

This implies that,

$$\|\mathbf{b}\| - \|\mathbf{b}'\| \leq \|\mathbf{b} - \mathbf{b}'\| \leq 3.$$

which differs slightly from the original (Chaudhuri et al., 2011) work, which bounded $\|\mathbf{b} - \mathbf{b}'\| \leq 2$. With this bound adjustment, we can write:

$$\frac{\mu(\mathbf{b}|\mathcal{D})}{\mu(\mathbf{b}'|\mathcal{D}')} = \frac{\|\mathbf{b}\|^{d-1} e^{-\epsilon' \|\mathbf{b}\|/3} \cdot \frac{1}{\text{surf}(\|\mathbf{b}\|)}}{\|\mathbf{b}'\|^{d-1} e^{-\epsilon' \|\mathbf{b}'\|/3} \cdot \frac{1}{\text{surf}(\|\mathbf{b}'\|)}} \leq e^{\epsilon'(\|\mathbf{b}\| - \|\mathbf{b}'\|)/3} \leq e^{\epsilon'},$$

where $\text{surf}(x)$ denotes the surface area of the sphere in $d$ dimensions with radius $x$. Here the last step follows from the fact that $\text{surf}(x) = s(1)x^{d-1}$, where $s(1)$ is the surface area of the unit sphere in $\mathbb{R}^d$.

Finally, we are ready to bound the ratio of densities:

$$\frac{g(\boldsymbol{\beta}_{priv}|\mathcal{D})}{g(\boldsymbol{\beta}_{priv}|\mathcal{D}')} = \frac{\mu(\mathbf{b}|\mathcal{D})}{\mu(\mathbf{b}'|\mathcal{D}')} \cdot \frac{|\det(\mathbf{J}(\boldsymbol{\beta}_{priv} \to \mathbf{b}|\mathcal{D}'))|}{|\det(\mathbf{J}(\boldsymbol{\beta}_{priv} \to \mathbf{b}'|\mathcal{D}))|}$$

$$= \frac{\mu(\mathbf{b}|\mathcal{D})}{\mu(\mathbf{b}'|\mathcal{D}')} \cdot \frac{|\det(A + E)|}{|\det A|}$$

$$\leq e^{\epsilon'} \cdot e^{\epsilon - \epsilon'}$$

$$\leq e^{\epsilon}.$$

Thus, Algorithm 1 satisfies Definition 16. $\qquad\qquad\qquad\qquad\qquad\qquad\qquad\qquad\qquad\qquad\qquad\qquad\qquad\qquad\qquad$ $\square$

# D. DP-SGD FTTransformer

For completeness, we give the DP-SGD algorithm with the *weighted* cross-entropy loss explicitly embedded in Algorithm 4. For more details on the DP-SGD algorithm, see (Abadi et al., 2016) and see (Yousefpour et al., 2021) for details on the empirical implementation.

---

**Algorithm 4** Differentially Private SGD (with weighted Cross-Entropy Loss)

---

1 **Inputs:** Database $\mathcal{D} = \{x_i, y_i\}$ with $n$ entries where each $y_i \in \{0, 1\}$, privacy parameters $(\epsilon, \delta)$, learning rate $\eta$, clipping norm $C$, minibatch size $B$, batch sampling probability $q = L/n$, number of iterations $T$, initial random model parameters $\theta$.

2 **Output:** Model parameters $\theta_{\text{priv}}$.

   **for** iteration $t = 1$ to $T$ **do**

3     Construct a batch of expected size $L$ by sampling each point into the batch with probability $q$

4     Partition the batch into minibatches of size $B$

    **for** each minibatch $b$ **do**

5       Compute model predictions $\hat{y}_i = f(x_i; \theta)$ for each $i \in b$.

6       Compute binary weighted cross-entropy loss as
$$\mathcal{L}(y, \hat{y}; \mathbf{w}) = -\frac{1}{B} \sum_{i=1}^{B} w_i \left[ y_i \log(\hat{y}_i) + (1 - y_i) \log(1 - \hat{y}_i) \right]$$

7       Compute per-sample gradients $\nabla \mathcal{L}_i = w_i (\hat{y}_i - y_i) \mathbf{x}_i$

8       Clip gradients $\tilde{\nabla} \mathcal{L}_i = \nabla \mathcal{L}_i \cdot \min \left( 1, \frac{C}{\|\nabla \mathcal{L}_i\|_2} \right)$

9       Parameterize $\sigma^2$ for $(\epsilon', \delta')$-DP, where $\epsilon' = O\left( \epsilon / \sqrt{T \log\left(\frac{1}{\delta}\right)} \right)$, for $(\epsilon, \delta)$-DP overall (Abadi et al., 2016).

10       Add noise: $\tilde{\nabla} \mathcal{L}_i = \tilde{\nabla} \mathcal{L}_i + \mathcal{N}(0, \sigma^2 C^2 \mathbf{I})$

11       Update model parameters $\theta = \theta - \eta \cdot \frac{1}{B} \sum_{i=1}^{B} \tilde{\nabla} \mathcal{L}_i$

   **end for**

   **end for**

12 Return differentially private model parameters: $\theta_{\text{priv}} = \theta$. =0

---

**Proposition 6.** *Algorithm 4, a standard DP-SGD procedure with weighted cross-entropy loss given by $\mathcal{L}(y, \hat{y}; \mathbf{w}) = -\frac{1}{n} \sum_{i=1}^{n} w_i \left[ y_i \log(\hat{y}_i) + (1 - y_i) \log(1 - \hat{y}_i) \right]$, is $(\epsilon, \delta)$-differentially private.*

*Proof.* $\mathcal{L}(y, \hat{y}; \mathbf{w})$ does not effect the sensitivity of the gradient $\nabla \mathcal{L}_i$ with respect to each sample; the gradient is bounded by the norm bound $C$ due to clipping. When each per-sample gradient $\nabla \mathcal{L}_i$ is clipped to $\tilde{\nabla} \mathcal{L}_i = \nabla \mathcal{L}_i \cdot \min \left( 1, \frac{C}{\|\nabla \mathcal{L}_i\|_2} \right)$, the sensitivity of the gradient is limited to $2C$. Adding Gaussian noise calibrated to this sensitivity ensures that the overall training procedure satisfies $(\epsilon, \delta)$-DP. Re-weighting of samples in the loss function *pre-clipping* does not affect these privacy guarantees. $\square$

## D.1. Note on Weighted DP-SGD Sensitivity

We note that in the standard unweighted case where $w_i = 1$ for all $i = 1, \ldots, n$ clipping guarantees that the minibatch average gradient $\frac{1}{B} \sum_{i \in b} \tilde{\nabla} \mathcal{L}_i$ has sensitivity $2C/B$. This is used in the noise calibration of standard implementations of noisy DP-SGD. We note however that in the weighted case all the $\tilde{\nabla} \mathcal{L}_i$ could change due to the weights $w_i$, making the sensitivity in fact $2C$. We show below that in fact the sensitivity of the average $\frac{1}{B} \sum_{i \in b} \nabla \mathcal{L}_i$ is bounded by $2C/B$ if $\|\nabla \mathcal{L}_i\| \leq C$.

**Lemma 19** (Sensitivity of Weighted DP-SGD under Class Weights). *For each mini-batch $b$ of size $B$, let $\tilde{\nabla} L_i = w_i(\hat{y}_i - y_i)x_i$ denote the weighted per-sample gradient. Suppose that $\|(\hat{y}_i - y_i)x_i\| \leq C$ Then,*

$$\left\| \sum_{i \in b} (\tilde{\nabla} L_i - \tilde{\nabla} L_i') \right\|_2 \leq \frac{2Cn_0}{Bn} - \frac{C}{n} < \frac{2C}{B}.$$

*Proof.* By construction, $w_i$ depends only on the ratio of $n_1$ and $n_0$. In the refined analysis given by Lemma 14, we showed that the total change in weights satisfies $\sum_{i=1}^{n} |w_i - w_i'| \leq 1 - \frac{1}{n}$. We will show something analogous here for the gradients; consider two neighboring datasets $\mathcal{D}$ and $\mathcal{D}'$ differing in one sample.

Weights are $w_i$ (recall $n_1 \ll n_0$). As before, let,

$$w_i = \begin{cases} \frac{n_0}{n}, & \text{if } i \in I_1, \\[2mm] \frac{n_1}{n}, & \text{if } i \in I_0, \end{cases} \tag{7}$$

, where $I_0$ and $I_1$ are the set of row indexes of $X$ that partitions it into $D_0$ and $D_1$. We can denote $\frac{n}{n_1} = \frac{1}{\hat{\pi}_1}$, and thus $(\frac{n}{n_1} + \frac{n}{n_0})^{-1} = (\frac{1}{\hat{\pi}_1} + \frac{1}{\hat{\pi}_0})^{-1}$ Keeping the minibatches equal for $D$, $D'$, we have, $\tilde{\nabla} L_i = w_i(\hat{y}_i - y_i)x_i$. so

$$\tilde{\nabla} L_i - \tilde{\nabla} L_i' = \begin{cases} (w_i - w_i')(\hat{y}_i - y_i)x_i, & i \neq j, \\[2mm] w_j(\hat{y}_j - y_j)x_j - w_j'(\hat{y}_j' - y_j')x_j', & i = j. \end{cases}$$

We'll first consider the case $I_1' \subset I_1$ i.e. the minority becomes smaller. Then we have,

$$\frac{1}{\hat{\pi}_1'} + \frac{1}{\hat{\pi}_0'} = n\left(\frac{1}{n_1 - 1} + \frac{1}{n_0 + 1}\right) = \frac{n(n_1 + n_0)}{(n_0 + 1)(n_1 - 1)} = \frac{n^2}{(n_0 + 1)(n_1 - 1)}.$$

So,

$$w_i' = \begin{cases} \frac{n}{n_1 - 1} \frac{(n_1 - 1)(n_0 + 1)}{n^2}, & \text{if } i \in I_1, \\[2mm] \frac{n}{n_0 + 1} \frac{(n_1 - 1)(n_0 + 1)}{n^2}, & \text{if } i \notin I_1, \end{cases}$$

while $\frac{1}{\hat{\pi}_1} + \frac{1}{\hat{\pi}_0} = n\left(\frac{1}{n_1} + \frac{1}{n_0}\right) = \frac{n^2}{n_1 n_0}$, and

$$w_i = \begin{cases} \frac{n}{n_1} \cdot \frac{n_1 n_0}{n^2}, & \text{if } i \in I_1, \\[2mm] \frac{n}{n_0} \cdot \frac{n_1 n_0}{n^2}, & \text{if } i \in I_0. \end{cases}$$

We'll consider this now with the difference in the loss between the two sets,

$$\tilde{\nabla} L_i - \tilde{\nabla} L_i' = \begin{cases} \left(\frac{n_0}{n} - \frac{n_0 + 1}{n}\right)(\hat{y}_i - y_i)x_i, & i \in I_1, i \neq j, \\[2mm] \left(\frac{n_1}{n} - \frac{n_1 - 1}{n}\right)(\hat{y}_i - y_i)x_i, & i \in I_0, \\[2mm] \frac{n_0}{n}(\hat{y}_j - y_j)x_j - \frac{n_0 + 1}{n}(\hat{y}_j' - y_j')x_j', & i = j. \end{cases}$$

After simplifying, we get,

$$\tilde{\nabla} L_i - \tilde{\nabla} L_i' = \begin{cases} -\frac{1}{n}(\hat{y}_i - y_i)x_i, & i \in I_1, i \neq j, \\[2mm] \frac{1}{n}(\hat{y}_i - y_i)x_i, & i \in I_0, \\[2mm] \frac{n_0}{n}(\hat{y}_j - y_j)x_j - \frac{n_0 + 1}{n}(\hat{y}_j' - y_j')x_j', & i = j. \end{cases}$$

Conversely, when $I_1 \subset I_1'$ (i.e. the minority gets larger), we have

$$\tilde{\nabla} L_i - \tilde{\nabla} L_i' = \begin{cases} \frac{1}{n}(\hat{y}_i - y_i)x_i, & i \in I_1, \\[2mm] -\frac{1}{n}(\hat{y}_i - y_i)x_i, & i \in I_0, i \neq j, \\[2mm] \frac{n_1}{n}(\hat{y}_j - y_j)x_j - \frac{n_1 + 1}{n}(\hat{y}_j' - y_j')x_j, & i = j. \end{cases}$$

Therefore,

$$\frac{1}{B}\left\|\sum_{i \in b_k}\left(\tilde{\nabla} L_i - \tilde{\nabla} L_i'\right)\right\|_2 \leq \frac{B - 1}{B} \cdot \frac{C}{n} + \frac{C}{B} \cdot \frac{2n_0 + 1}{n} = \frac{2Cn_0}{Bn} - \frac{C}{n}.$$

$\square$

## D.2. Visualizing Decision Boundaries

Next we explore the effect of differential privacy on decision boundaries by presenting visualizations on 2-dimensional synthetic data. These visualizations of decision boundaries help develop intuition for how private noise impacts model predictions, particularly in class-imbalanced settings.

We generate a small ($n = 1000$) synthetic 2-dimensional mixture of Gaussians, where majority (negative) and minority (positive) classes are separable in the feature space. Specifically, the random vector $[X_1, X_2]$ is sampled from the following process: with probability 0.9, $[X_1, X_2] \sim \mathcal{N}([0,0], \begin{bmatrix} 4 & 0 \\ 0 & 4 \end{bmatrix})$, and with probability 0.1, $[X_1, X_2] \sim \mathcal{N}([4,4], \begin{bmatrix} 4 & 0 \\ 0 & 4 \end{bmatrix})$. Thus, the mixture has two components: one centered at $[0, 0]$ and the other at $[4, 4]$, both independent and with variance 4.

Figure 4 compares the decision boundaries of non-differentially private and differentially private classifiers on this data, allowing us to directly observe the impact of the privacy preserving methods on how the model makes decisions. The blue points represent majority (negative) class examples, while the red points represent minority (positive) class examples. The blue region denotes where the model will predict a negative label, and the red region denotes where the model will predict a positive label. The underlying data distributions are also visible in these figures, represented as an mean-centered ellipse capturing 2 standard deviations of the 2d-Gaussian.

Inspecting Figure 4 helps build intuition for the effect of DP on decision boundaries. We observe that *Priv. Weighted FTT* fails to learn a meaningful decision boundary (labeling everthing negative), while *Priv. LogReg* is catastrophically noisy (flipping the decision boundary). *GEM + NonPriv. XGBoost* (Algorithm 3) is lossy relative to *SMOTE + NonPriv. XGBoost*, but maintains a class separating boundary.

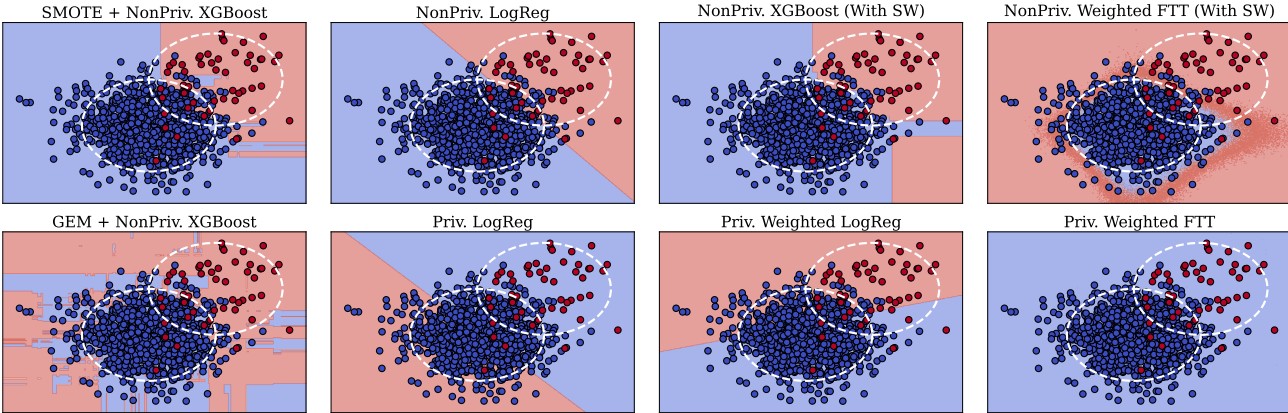

Figure 4: Top row shows decision boundaries of non-DP classifiers (high performance on the task, $AUC \in [0.94, 0.97]$). Bottom row illustrates the decision boundaries of DP classifiers ($\epsilon = 1.0$, $\delta = 1e{-}5$ where applicable), which perform worse. The underlying true data generating function for each class is represented as an ellipse (dotted white line), where the center of the ellipse is the mean and each point on the dotted line represents 2 standard deviations from the mean.

# E. Additional Experimental Results and Details

**Methods Evaluated**  Here, we restate exhaustively the range of methods we evaluate under different privacy and class imbalance conditions. We categorize methods as pre-processing or in-processing methods.

We evaluate: (1) a private synthetic data method (GEM) as a pre-processing step, generating a class-balanced sample for a downstream, non-private XGBoost model (*GEM + NonPriv. XGBoost*, Section 3.3), (2) a private ERM logistic regression model as an in-processing step *without* class weights (*Priv. LogReg*, exact method from (Chaudhuri et al., 2011), see Section 4.2), (3) a private ERM logistic regression model as an in-processing step *with* sample weights (*Priv. Weighted LogReg*, our modified algorithm under class weighting, Algorithm 1 in Section 4.2), and (4) a DP-SGD trained FTTransformer model as an in-processing step *with* sample weights in the cross-entropy loss (*Priv. Weighted FTT*, Section 4.3).

We also compare the performance of these methods against the following non-private baselines: (1) a vanilla XGBoost model with in-processing sample weights (*NonPriv. Weighted XGBoost*), (2) an XGBoost model *without* class sample weights, using SMOTE as a pre-processing step (*SMOTE + NonPriv. XGBoost*), (3) a logistic regression model with and without sample weights (as in-processing) (*NonPriv. Weighted LogReg / NonPriv. LogReg*), and (4) a non-private FTTransformer model with and without sample weights in the cross-entropy loss (as in-processing) (*NonPriv. Weighted FTT / NonPriv. FTT*). These methods serve as baselines for comparison to measure the effects of adding differential privacy, and the role of weighting in model performance.

**Experimental Details**  All datasets we chose were purposefully low-dimensional enough to be run with GEM. Neural models (GEM and FTTransformer) were trained using an NVIDIA T4 GPU, with $\epsilon \in \{0.05, 0.1, 0.5, 1.0, 5.0\}$ (privacy budget range following guidance from (McKenna et al., 2022)). Private models were trained for 20 epochs, while non-private models were trained for 100 epochs with early stopping. FTTransformer was initialized with default architecture hyper-parameters (dimension=32, depth=6, 8 heads, dropout of 0.1). DP-SGD was performed with the Opacus pytorch library using recommended parameters (Yousefpour et al., 2021). No hyperparameter tuning was performed for the private models to ensure "honest" comparisons (Papernot & Steinke, 2021); hyperparameters were lightly tuned for non-private models using randomized cross-validation. Results are given with standard deviations over 10 randomly seeded data splits and parameter initializations. GEM models are computationally expensive (Liu et al., 2021; Rosenblatt et al., 2024a); they were trained in parallel on the same NVIDIA T4 and took over 50 compute hours. XGBoost and LogReg models trained within seconds, while FTTransformer models required minutes.

| ID | Name | Repository & Target | $r = \frac{n_0}{n_1}$ | Size $n$ | # Features |
|---|---|---|---|---|---|
| 1 | ecoli | UCI, target: imU | 8.6 | 336 | 7 |
| 2 | yeast_me2 | UCI, target: ME2 | 28 | 1,484 | 8 |
| 3 | solar_flare_m0 | UCI, target: M-0 | 19 | 1,389 | 32 |
| 4 | abalone | UCI, target: 7 | 9.7 | 4,177 | 10 |
| 5 | car_eval_34 | UCI, target: good, v good | 12 | 1,728 | 21 |
| 6 | car_eval_4 | UCI, target: vgood | 26 | 1,728 | 21 |
| 7 | mammography | UCI, target: minority | 42 | 11,183 | 6 |
| 8 | abalone_19 | UCI, target: 19 | 130 | 4,177 | 10 |

Table 11: Imbalanced learning datasets used from the `imblearn` package.

**GEM Summary.**  Much of tabular private synthetic data generation has focused on matching distributions based on the broad class of linear statistical queries, often referred to as counting queries. The objective is generally set up as follows: you are given a finite set of queries $Q$, and the objective is to construct a synthetic dataset $D$ such that the maximum error across all queries in $Q$, defined as $\max_{q \in Q} |q(D)|$, is minimized.

GEM is an $(\epsilon, \delta)$-DP neural method that fits a private, parameterized weight distribution $G_\theta$, where $\theta$ represents the learnable parameters of the model. It follows the Select-Measure-Project paradigm, and its main novelty lies in the *project* step: the method fits a neural network, denoted as $G_\theta$, to approximate a distribution over the data domain in a differentially private manner. This network generates a product distribution $P_\theta$, where $P_\theta$ represents the output distribution over a discretized version of the data domain.

| Hyperparameter | Value | Description |
|---|---|---|
| $k$ | 3 | Dimension of marginal considered in random query workload. |
| $T$ | 100 | Number of iterations for the GEM algorithm. |
| $\alpha$ | 0.67 | Weighting parameter for the GEM algorithm. |
| loss_p | 2 | The $p$-norm used in the loss function. |
| lr | 1e−4 | Learning rate for the optimizer in the GEM algorithm. |
| max_idxs | 100 | Maximum number of indices considered during each iteration. |
| max_iters | 100 | Maximum number of iterations for optimization. |
| ema_weights_beta | 0.9 | Smoothing parameter for exponential moving average weights. |
| embedding_dim | 512 | Dimensionality of the embedding space for the neural network generator. |

Table 12: Hyperparameters we used when running GEM and their respective descriptions.

The process works by sampling random Gaussian noise vectors $z$, which are passed through the neural network $G_\theta$ to output a distribution $P_\theta(z)$ in the same domain as the target data. This product distribution is normalized to ensure it behaves as a valid marginal probability vector. Once fit, arbitrarily many samples can be generated from the fully specified distribution $P_\theta$.

Any statistical query $q$ can be described as a function mapping $P_\theta$ to a value in $[0,1]$, i.e., $q(P_\theta) = \sum_{x \in X} \phi(x) P_\theta(x)$, where $\phi(x)$ is the predicate function defining the query. Any query $q$ is differentiable with respect to the parameters $\theta$ of the model. Given a set of queries $\tilde{q}_i \in \tilde{Q}_{1:T}$, which are privately selected using the Exponential Mechanism, and answers $\tilde{a}_i \in \tilde{A}_{1:T}$ privately computed using an additive noise mechanism, a natural loss function for the parameterization $\theta$ is given by:

$$\mathcal{L}_{GEM}\left(\theta, \tilde{Q}_{1:T}, \tilde{A}_{1:T}\right) = \sum_{i \in [T]} |\tilde{q}_i(P_\theta) - \tilde{a}_i|.$$

GEM iteratively updates $\theta$ to minimize this loss function, incorporating the observed queries and answers. The most common linear query class used are $k$-way marginal queries (Hardt et al., 2012; Vietri et al., 2020; Aydore et al., 2021). In our experiments, we parameterized GEM to use 3-way marginal queries. We detail other hyperparameter settings for GEM in Table 12.

**PrivBayes Summary.** PrivBayes builds a Bayesian network to approximate the joint distribution of the data by factorizing it into a sequence of conditional probabilities, which it can then sample from to create differentially private synthetic data. To ensure DP, it first selects an attribute ordering using mutual information (privatized by an additive noise mechanism) to determine parent-child relationships. Then, for each attribute, it estimates the attribute's conditional probability distribution given its parent attributes using a DP noise-perturbed frequency table. Once the Bayesian network is constructed, synthetic data points are generated by sampling from the learned network.

**FTTransformer Summary.** We adapt a recently proposed transformer-based model, FTTransformer (Gorishniy et al., 2021), to the DP setting, which involves minor adjustments to the model architecture for compatibility with Opacus (Yousefpour et al., 2021). FTTransformer is a neural tabular data classifier that is competitive with well-known gradient boosting tree-based methods like XGBoost (Chen & Guestrin, 2016); its novelty lies in data transformations for attenuation by the attention layers in a transformer architecture (Wolf et al., 2020; Tay et al., 2022; Khan et al., 2022). Our empirical results rely on modifications to implementations for DP-SGD from the Opacus library (Yousefpour et al., 2021) and the base implementation for FTTransformer from (Huang et al., 2020).

After experimenting with different neural architectures in the non-private setting, we found that the FTTransformer architecture was significantly better than other methods on tabular data tasks, even for imbalanced classification. However, when transitioning to the private setting, we found that all of the neural methods using DP-SGD had trouble under class imbalance. FTTransformer still performed best among these (albeit poorly relative to other model classes), so we included the Private FTTransformer implementation to represent the class of neural models trained with DP-SGD (using a weighted cross-entropy loss, which helped a little on imbalanced classification metrics).

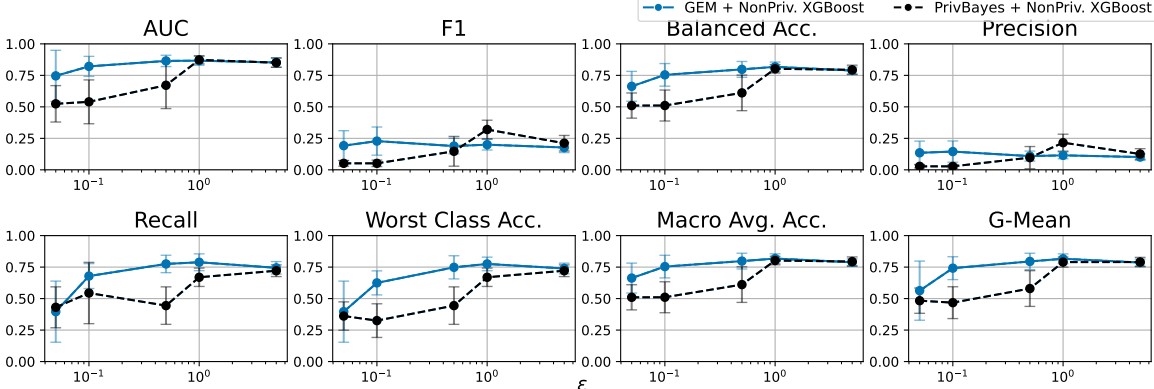

Figure 5: Comparison of PrivBayes ([Zhang et al., 2017](#)) and GEM ([Liu et al., 2021](#)) as private preprocessing steps on the *mammography* dataset, with XGBoost as the downstream non-private classifier. PrivBayes, while generally weaker, shows similar performance trends to GEM as $\epsilon$ increases and is a strong private pre-processing step for imbalanced classification.

### E.1. Performance of models on all `imblearn` datasets

This section presents figures that detail exhaustive performance across privacy parameter (i.e., varying $\epsilon$ from 0.01 to 5.0) for all the datasets listed in Table 11.

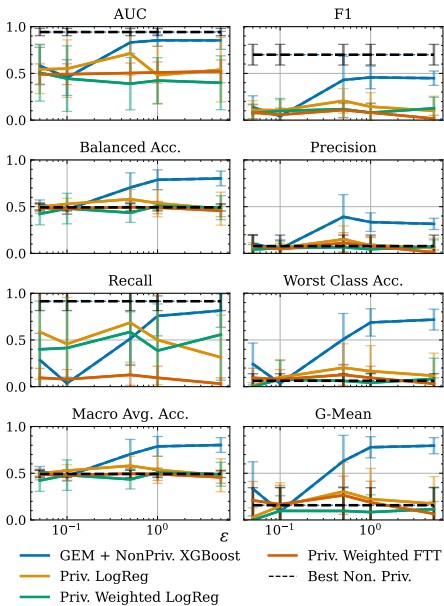

Figure 6: Privacy-preserving predictors across $\epsilon$ settings for *ecoli* dataset.

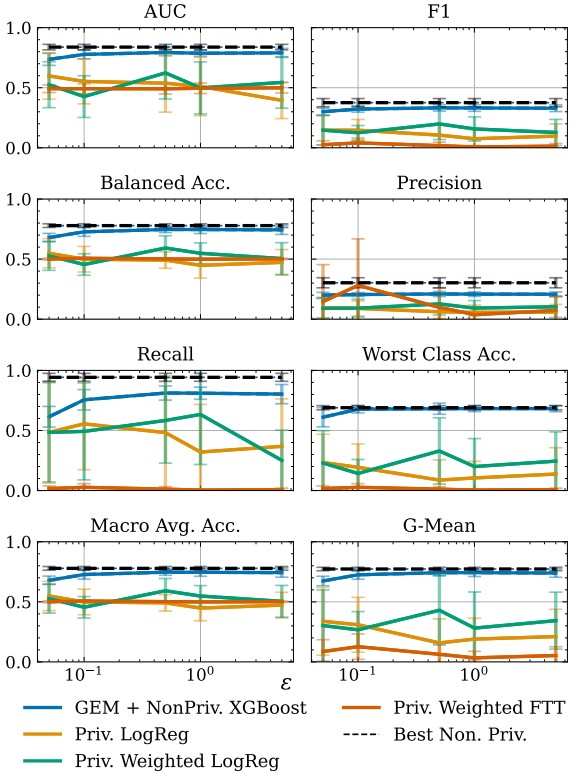

Figure 7: Privacy-preserving predictors across $\epsilon$ settings for *abalone* dataset.

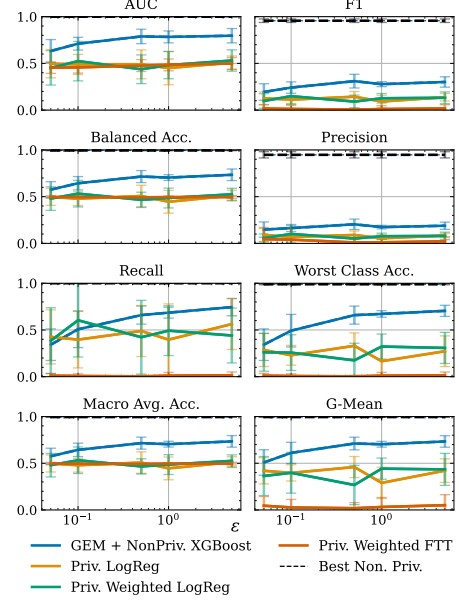

Figure 8: Privacy-preserving predictors across $\epsilon$ settings for $car\_eval\_34$ dataset.

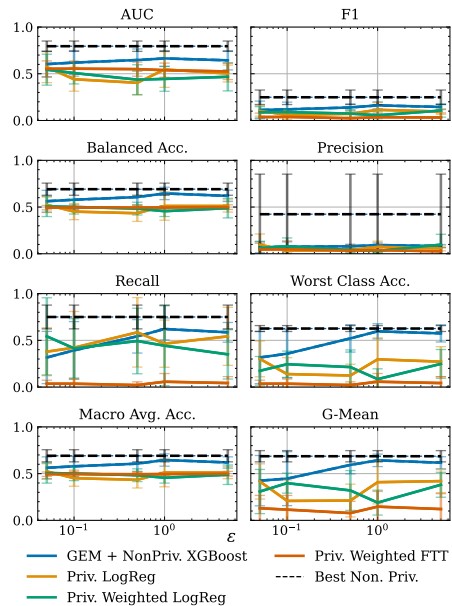

Figure 9: Privacy-preserving predictors across $\epsilon$ settings for $solar\_flare\_m0$ dataset.

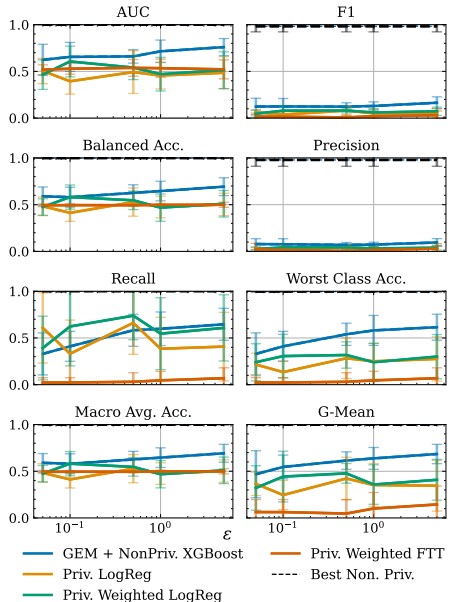

Figure 10: Privacy-preserving predictors across $\epsilon$ settings for $car\_eval\_4$ dataset.

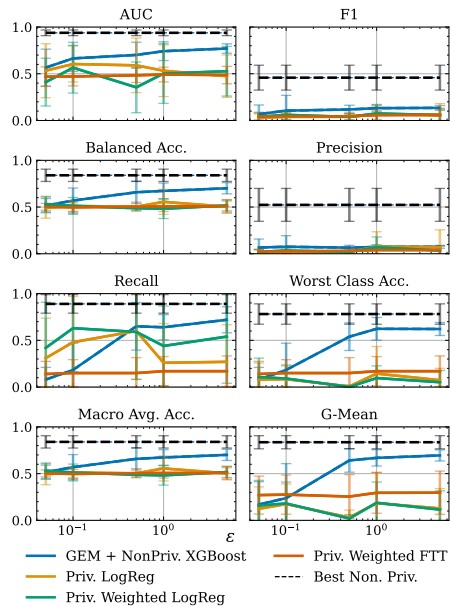

Figure 11: Privacy-preserving predictors across $\epsilon$ settings for $yeast\_me2$ dataset.

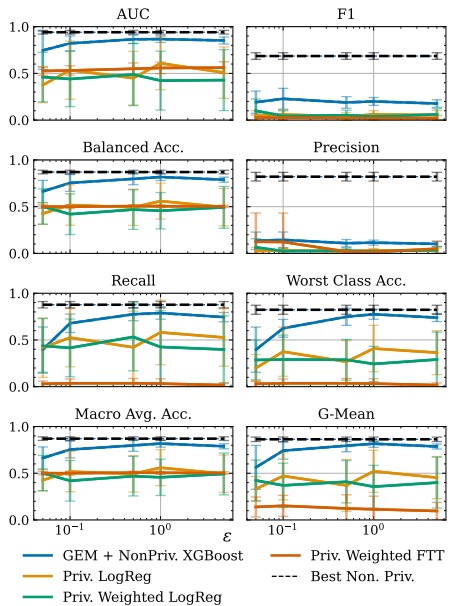

Figure 12: Privacy-preserving predictors across $\epsilon$ settings for $mammography$ dataset.

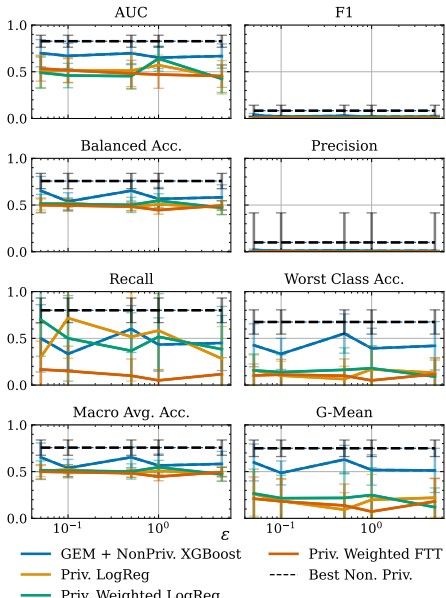

Figure 13: Privacy-preserving predictors across $\epsilon$ settings for $abalone\_19$ dataset.

## E.2. Complete non-private results

Table 13: Ecoli Dataset

| Metrics | Standard ↑ | | | | Imbalanced ↑ | | | |
|---|---|---|---|---|---|---|---|---|
| Approach | AUC | F1 | Bal-ACC | Prec./Recall | Worst-ACC | Avg-ACC | G-Mean | MCC |
| Non-Private ↓ | | | | | | | | |
| Identity + NonPriv. LogReg | 0.94 ± 0.04 | 0.09 ± 0.12 | 0.53 ± 0.04 | 0.3 ± 0.42 / 0.06 ± 0.07 | 0.06 ± 0.07 | 0.53 ± 0.04 | 0.15 ± 0.19 | 0.11 ± 0.16 |
| Identity + NonPriv. Weighted LogReg | 0.93 ± 0.04 | 0.55 ± 0.11 | 0.77 ± 0.08 | 0.52 ± 0.12 / 0.61 ± 0.17 | 0.61 ± 0.17 | 0.77 ± 0.08 | 0.75 ± 0.1 | 0.51 ± 0.13 |
| Identity + NonPriv. XGBoost | 0.91 ± 0.07 | 0.63 ± 0.15 | 0.77 ± 0.09 | 0.77 ± 0.19 / 0.56 ± 0.17 | 0.56 ± 0.17 | 0.77 ± 0.09 | 0.73 ± 0.12 | 0.61 ± 0.16 |
| Identity + NonPriv. Weighted XGBoost | 0.91 ± 0.08 | 0.65 ± 0.16 | 0.8 ± 0.1 | 0.72 ± 0.18 / 0.63 ± 0.2 | 0.63 ± 0.2 | 0.8 ± 0.1 | 0.77 ± 0.13 | 0.63 ± 0.17 |
| SMOTE + NonPriv. LogReg | 0.94 ± 0.04 | 0.61 ± 0.09 | 0.88 ± 0.06 | 0.47 ± 0.1 / 0.89 ± 0.11 | 0.83 ± 0.07 | 0.88 ± 0.06 | 0.88 ± 0.06 | 0.59 ± 0.1 |
| SMOTE + NonPriv. Weighted LogReg | 0.94 ± 0.04 | 0.51 ± 0.07 | 0.86 ± 0.05 | 0.36 ± 0.06 / 0.91 ± 0.1 | 0.79 ± 0.06 | 0.86 ± 0.05 | 0.85 ± 0.05 | 0.5 ± 0.08 |
| SMOTE + NonPriv. Weighted XGB | 0.94 ± 0.04 | 0.68 ± 0.1 | 0.85 ± 0.07 | 0.65 ± 0.12 / 0.74 ± 0.15 | 0.74 ± 0.14 | 0.85 ± 0.07 | 0.84 ± 0.08 | 0.65 ± 0.11 |
| SMOTE + NonPriv. XGBoost | 0.94 ± 0.04 | 0.7 ± 0.11 | 0.85 ± 0.08 | 0.7 ± 0.16 / 0.74 ± 0.18 | 0.74 ± 0.17 | 0.85 ± 0.08 | 0.84 ± 0.1 | 0.68 ± 0.12 |
| Identity + NonPriv. Weighted FTTransformer | 0.51 ± 0.12 | 0.12 ± 0.09 | 0.51 ± 0.04 | 0.12 ± 0.10 / 0.13 ± 0.11 | 0.13 ± 0.11 | 0.51 ± 0.04 | 0.21 ± 0.21 | 0.09 ± 0.09 |

Table 14: Abolone Dataset

| Metrics | Standard ↑ | | | | Imbalanced ↑ | | | |
|---|---|---|---|---|---|---|---|---|
| Approach | AUC | F1 | Bal-ACC | Prec./Recall | Worst-ACC | Avg-ACC | G-Mean | MCC |
| Non-Private ↓ | | | | | | | | |
| Identity + NonPriv. LogReg | 0.81 ± 0.02 | 0.0 ± 0.0 | 0.5 ± 0.0 | 0.0 ± 0.0 / 0.0 ± 0.0 | 0.0 ± 0.0 | 0.5 ± 0.0 | 0.0 ± 0.0 | 0.0 ± 0.0 |
| Identity + NonPriv. Weighted LogReg | 0.81 ± 0.02 | 0.38 ± 0.03 | 0.73 ± 0.03 | 0.27 ± 0.03 / 0.63 ± 0.04 | 0.63 ± 0.04 | 0.73 ± 0.03 | 0.72 ± 0.03 | 0.32 ± 0.04 |
| Identity + NonPriv. XGBoost | 0.84 ± 0.02 | 0.19 ± 0.05 | 0.55 ± 0.02 | 0.29 ± 0.08 / 0.14 ± 0.04 | 0.14 ± 0.04 | 0.55 ± 0.02 | 0.36 ± 0.05 | 0.15 ± 0.06 |
| Identity + NonPriv. Weighted XGBoost | 0.84 ± 0.02 | 0.35 ± 0.04 | 0.66 ± 0.03 | 0.3 ± 0.04 / 0.43 ± 0.05 | 0.43 ± 0.05 | 0.66 ± 0.03 | 0.62 ± 0.04 | 0.28 ± 0.05 |
| SMOTE + NonPriv. LogReg | 0.83 ± 0.02 | 0.36 ± 0.01 | 0.78 ± 0.02 | 0.22 ± 0.01 / 0.87 ± 0.03 | 0.69 ± 0.02 | 0.78 ± 0.02 | 0.77 ± 0.01 | 0.34 ± 0.02 |
| SMOTE + NonPriv. Weighted LogReg | 0.82 ± 0.02 | 0.31 ± 0.01 | 0.76 ± 0.02 | 0.19 ± 0.01 / 0.94 ± 0.03 | 0.58 ± 0.02 | 0.76 ± 0.02 | 0.74 ± 0.01 | 0.31 ± 0.02 |
| SMOTE + NonPriv. Weighted XGB | 0.84 ± 0.02 | 0.37 ± 0.04 | 0.69 ± 0.03 | 0.3 ± 0.04 / 0.49 ± 0.07 | 0.49 ± 0.07 | 0.69 ± 0.03 | 0.66 ± 0.04 | 0.3 ± 0.05 |
| SMOTE + NonPriv. XGBoost | 0.84 ± 0.02 | 0.32 ± 0.04 | 0.64 ± 0.03 | 0.29 ± 0.03 / 0.36 ± 0.06 | 0.36 ± 0.06 | 0.64 ± 0.03 | 0.57 ± 0.04 | 0.25 ± 0.04 |
| Identity + NonPriv. Weighted FTTransformer | 0.70 ± 0.03 | 0.06 ± 0.09 | 0.52 ± 0.04 | 0.19 ± 0.20 / 0.07 ± 0.14 | 0.07 ± 0.14 | 0.52 ± 0.04 | 0.19 ± 0.19 | 0.08 ± 0.08 |

Table 15: Car_eval_34 Dataset

| Metrics | Standard ↑ | | | | Imbalanced ↑ | | | |
|---|---|---|---|---|---|---|---|---|
| Approach | AUC | F1 | Bal-ACC | Prec./Recall | Worst-ACC | Avg-ACC | G-Mean | MCC |
| Non-Private ↓ | | | | | | | | |
| Identity + NonPriv. LogReg | 1.0 ± 0.0 | 0.86 ± 0.05 | 0.89 ± 0.04 | 0.95 ± 0.03 / 0.79 ± 0.08 | 0.79 ± 0.08 | 0.89 ± 0.04 | 0.89 ± 0.05 | 0.85 ± 0.05 |
| Identity + NonPriv. Weighted LogReg | 1.0 ± 0.0 | 0.85 ± 0.03 | 0.98 ± 0.0 | 0.74 ± 0.05 / 1.0 ± 0.0 | 0.97 ± 0.01 | 0.98 ± 0.0 | 0.98 ± 0.0 | 0.84 ± 0.03 |
| Identity + NonPriv. XGBoost | 1.0 ± 0.0 | 0.96 ± 0.02 | 0.98 ± 0.02 | 0.94 ± 0.03 / 0.97 ± 0.03 | 0.97 ± 0.03 | 0.98 ± 0.02 | 0.98 ± 0.02 | 0.95 ± 0.02 |
| Identity + NonPriv. Weighted XGBoost | 1.0 ± 0.0 | 0.94 ± 0.03 | 0.99 ± 0.0 | 0.89 ± 0.05 / 1.0 ± 0.0 | 0.99 ± 0.01 | 0.99 ± 0.0 | 0.99 ± 0.0 | 0.94 ± 0.03 |
| SMOTE + NonPriv. LogReg | 1.0 ± 0.0 | 0.85 ± 0.04 | 0.98 ± 0.0 | 0.74 ± 0.04 / 1.0 ± 0.0 | 0.97 ± 0.01 | 0.98 ± 0.0 | 0.98 ± 0.0 | 0.84 ± 0.03 |
| SMOTE + NonPriv. Weighted LogReg | 1.0 ± 0.0 | 0.73 ± 0.03 | 0.97 ± 0.0 | 0.58 ± 0.03 / 1.0 ± 0.0 | 0.94 ± 0.01 | 0.97 ± 0.0 | 0.97 ± 0.0 | 0.74 ± 0.02 |
| SMOTE + NonPriv. Weighted XGB | 1.0 ± 0.0 | 0.95 ± 0.02 | 0.99 ± 0.01 | 0.92 ± 0.05 / 0.99 ± 0.02 | 0.98 ± 0.01 | 0.99 ± 0.01 | 0.99 ± 0.01 | 0.95 ± 0.02 |
| SMOTE + NonPriv. XGBoost | 1.0 ± 0.0 | 0.96 ± 0.02 | 0.98 ± 0.01 | 0.94 ± 0.04 / 0.97 ± 0.03 | 0.97 ± 0.03 | 0.98 ± 0.01 | 0.98 ± 0.01 | 0.95 ± 0.02 |
| Identity + NonPriv. Weighted FTTransformer | 1.00 ± 0.00 | 0.92 ± 0.04 | 0.96 ± 0.03 | 0.91 ± 0.04 / 0.94 ± 0.06 | 0.93 ± 0.06 | 0.96 ± 0.03 | 0.03 ± 0.03 | 0.04 ± 0.04 |

Table 16: Solar_flare_m0 Dataset

| Metrics | Standard ↑ | | | | Imbalanced ↑ | | | |
|---|---|---|---|---|---|---|---|---|
| Approach | AUC | F1 | Bal-ACC | Prec./Recall | Worst-ACC | Avg-ACC | G-Mean | MCC |
| Non-Private ↓ | | | | | | | | |
| Identity + NonPriv. LogReg | 0.79 ± 0.06 | 0.03 ± 0.05 | 0.51 ± 0.02 | 0.15 ± 0.34 / 0.01 ± 0.03 | 0.01 ± 0.03 | 0.51 ± 0.02 | 0.05 ± 0.11 | 0.04 ± 0.1 |
| Identity + NonPriv. Weighted LogReg | 0.79 ± 0.06 | 0.25 ± 0.08 | 0.67 ± 0.08 | 0.17 ± 0.05 / 0.44 ± 0.16 | 0.44 ± 0.16 | 0.67 ± 0.08 | 0.62 ± 0.11 | 0.22 ± 0.1 |
| Identity + NonPriv. XGBoost | 0.73 ± 0.04 | 0.09 ± 0.09 | 0.53 ± 0.03 | 0.17 ± 0.16 / 0.06 ± 0.06 | 0.06 ± 0.06 | 0.53 ± 0.03 | 0.19 ± 0.17 | 0.08 ± 0.09 |
| Identity + NonPriv. Weighted XGBoost | 0.74 ± 0.04 | 0.2 ± 0.05 | 0.61 ± 0.04 | 0.15 ± 0.04 / 0.31 ± 0.09 | 0.31 ± 0.09 | 0.61 ± 0.04 | 0.53 ± 0.07 | 0.15 ± 0.06 |
| SMOTE + NonPriv. LogReg | 0.76 ± 0.07 | 0.19 ± 0.04 | 0.67 ± 0.06 | 0.11 ± 0.02 / 0.58 ± 0.12 | 0.57 ± 0.1 | 0.67 ± 0.06 | 0.66 ± 0.07 | 0.17 ± 0.06 |
| SMOTE + NonPriv. Weighted LogReg | 0.75 ± 0.07 | 0.17 ± 0.03 | 0.69 ± 0.06 | 0.1 ± 0.02 / 0.75 ± 0.13 | 0.63 ± 0.03 | 0.69 ± 0.06 | 0.69 ± 0.06 | 0.17 ± 0.06 |
| SMOTE + NonPriv. Weighted XGB | 0.7 ± 0.04 | 0.09 ± 0.06 | 0.52 ± 0.04 | 0.09 ± 0.06 / 0.1 ± 0.08 | 0.1 ± 0.08 | 0.52 ± 0.04 | 0.27 ± 0.16 | 0.04 ± 0.07 |
| SMOTE + NonPriv. XGBoost | 0.7 ± 0.05 | 0.07 ± 0.06 | 0.52 ± 0.02 | 0.08 ± 0.07 / 0.06 ± 0.05 | 0.06 ± 0.05 | 0.52 ± 0.02 | 0.2 ± 0.15 | 0.03 ± 0.05 |
| Identity + NonPriv. Weighted FTTransformer | 0.76 ± 0.06 | 0.09 ± 0.12 | 0.53 ± 0.04 | 0.20 ± 0.27 / 0.06 ± 0.08 | 0.06 ± 0.08 | 0.53 ± 0.04 | 0.20 ± 0.20 | 0.14 ± 0.14 |

Table 17: Car_eval_4 Dataset

| Metrics | Standard ↑ | | | | Imbalanced ↑ | | | |
|---|---|---|---|---|---|---|---|---|
| Approach | AUC | F1 | Bal-ACC | Prec./Recall | Worst-ACC | Avg-ACC | G-Mean | MCC |
| **Non-Private ↓** | | | | | | | | |
| Identity + NonPriv. LogReg | 1.0 ± 0.0 | 0.75 ± 0.1 | 0.82 ± 0.07 | 0.93 ± 0.08 / 0.64 ± 0.13 | 0.64 ± 0.13 | 0.82 ± 0.07 | 0.79 ± 0.08 | 0.76 ± 0.1 |
| Identity + NonPriv. Weighted LogReg | 1.0 ± 0.0 | 0.75 ± 0.06 | 0.99 ± 0.0 | 0.6 ± 0.08 / 1.0 ± 0.0 | 0.97 ± 0.01 | 0.99 ± 0.0 | 0.99 ± 0.0 | 0.76 ± 0.05 |
| Identity + NonPriv. XGBoost | 1.0 ± 0.0 | 0.98 ± 0.06 | 0.99 ± 0.03 | 0.98 ± 0.07 / 0.98 ± 0.05 | 0.98 ± 0.05 | 0.99 ± 0.03 | 0.99 ± 0.03 | 0.98 ± 0.06 |
| Identity + NonPriv. Weighted XGBoost | 1.0 ± 0.0 | 0.85 ± 0.06 | 0.99 ± 0.0 | 0.74 ± 0.09 / 1.0 ± 0.0 | 0.99 ± 0.01 | 0.99 ± 0.0 | 0.99 ± 0.0 | 0.85 ± 0.06 |
| SMOTE + NonPriv. LogReg | 1.0 ± 0.0 | 0.81 ± 0.06 | 0.99 ± 0.0 | 0.68 ± 0.08 / 1.0 ± 0.0 | 0.98 ± 0.01 | 0.99 ± 0.0 | 0.99 ± 0.0 | 0.82 ± 0.05 |
| SMOTE + NonPriv. Weighted LogReg | 1.0 ± 0.0 | 0.77 ± 0.06 | 0.99 ± 0.0 | 0.63 ± 0.08 / 1.0 ± 0.0 | 0.98 ± 0.01 | 0.99 ± 0.0 | 0.99 ± 0.0 | 0.78 ± 0.05 |
| SMOTE + NonPriv. Weighted XGB | 1.0 ± 0.0 | 0.96 ± 0.06 | 1.0 ± 0.0 | 0.92 ± 0.1 / 1.0 ± 0.0 | 1.0 ± 0.01 | 1.0 ± 0.0 | 1.0 ± 0.0 | 0.96 ± 0.06 |
| SMOTE + NonPriv. XGBoost | 1.0 ± 0.0 | 0.97 ± 0.05 | 0.99 ± 0.01 | 0.95 ± 0.09 / 0.99 ± 0.02 | 0.99 ± 0.02 | 0.99 ± 0.01 | 0.99 ± 0.01 | 0.97 ± 0.05 |
| Identity + NonPriv. Weighted FTTransformer | 0.99 ± 0.01 | 0.82 ± 0.10 | 0.94 ± 0.07 | 0.78 ± 0.14 / 0.90 ± 0.14 | 0.89 ± 0.13 | 0.94 ± 0.07 | 0.07 ± 0.07 | 0.10 ± 0.10 |

Table 18: Yeast_me2 Dataset

| Metrics | Standard ↑ | | | | Imbalanced ↑ | | | |
|---|---|---|---|---|---|---|---|---|
| Approach | AUC | F1 | Bal-ACC | Prec./Recall | Worst-ACC | Avg-ACC | G-Mean | MCC |
| **Non-Private ↓** | | | | | | | | |
| Identity + NonPriv. LogReg | 0.88 ± 0.06 | 0.0 ± 0.0 | 0.5 ± 0.0 | 0.0 ± 0.0 / 0.0 ± 0.0 | 0.0 ± 0.0 | 0.5 ± 0.0 | 0.0 ± 0.0 | 0.0 ± 0.0 |
| Identity + NonPriv. Weighted LogReg | 0.88 ± 0.06 | 0.27 ± 0.07 | 0.66 ± 0.04 | 0.22 ± 0.07 / 0.37 ± 0.08 | 0.37 ± 0.08 | 0.66 ± 0.04 | 0.59 ± 0.07 | 0.25 ± 0.07 |
| Identity + NonPriv. XGBoost | 0.93 ± 0.03 | 0.33 ± 0.13 | 0.62 ± 0.05 | 0.52 ± 0.18 / 0.25 ± 0.11 | 0.25 ± 0.11 | 0.62 ± 0.05 | 0.49 ± 0.11 | 0.34 ± 0.13 |
| Identity + NonPriv. Weighted XGBoost | 0.94 ± 0.03 | 0.46 ± 0.13 | 0.77 ± 0.08 | 0.38 ± 0.13 / 0.58 ± 0.15 | 0.58 ± 0.15 | 0.77 ± 0.08 | 0.74 ± 0.1 | 0.45 ± 0.14 |
| SMOTE + NonPriv. LogReg | 0.9 ± 0.04 | 0.29 ± 0.04 | 0.84 ± 0.07 | 0.18 ± 0.03 / 0.81 ± 0.14 | 0.78 ± 0.11 | 0.84 ± 0.07 | 0.84 ± 0.07 | 0.34 ± 0.06 |
| SMOTE + NonPriv. Weighted LogReg | 0.9 ± 0.04 | 0.2 ± 0.01 | 0.82 ± 0.04 | 0.11 ± 0.01 / 0.89 ± 0.1 | 0.74 ± 0.03 | 0.82 ± 0.04 | 0.82 ± 0.04 | 0.26 ± 0.03 |
| SMOTE + NonPriv. Weighted XGB | 0.92 ± 0.04 | 0.4 ± 0.09 | 0.71 ± 0.04 | 0.39 ± 0.12 / 0.45 ± 0.08 | 0.45 ± 0.08 | 0.71 ± 0.04 | 0.66 ± 0.07 | 0.39 ± 0.09 |
| SMOTE + NonPriv. XGBoost | 0.92 ± 0.04 | 0.42 ± 0.12 | 0.71 ± 0.06 | 0.41 ± 0.15 / 0.45 ± 0.12 | 0.45 ± 0.12 | 0.71 ± 0.06 | 0.66 ± 0.1 | 0.41 ± 0.12 |
| Identity + NonPriv. Weighted FTTransformer | 0.51 ± 0.10 | 0.03 ± 0.04 | 0.50 ± 0.05 | 0.02 ± 0.03 / 0.09 ± 0.16 | 0.09 ± 0.16 | 0.50 ± 0.05 | 0.22 ± 0.22 | 0.05 ± 0.05 |

Table 19: Mammography Dataset

| Metrics | Standard ↑ | | | | Imbalanced ↑ | | | |
|---|---|---|---|---|---|---|---|---|
| Approach | AUC | F1 | Bal-ACC | Prec./Recall | Worst-ACC | Avg-ACC | G-Mean | MCC |
| **Non-Private ↓** | | | | | | | | |
| Identity + NonPriv. LogReg | 0.9 ± 0.02 | 0.53 ± 0.05 | 0.7 ± 0.03 | 0.8 ± 0.07 / 0.4 ± 0.06 | 0.4 ± 0.06 | 0.7 ± 0.03 | 0.63 ± 0.05 | 0.55 ± 0.05 |
| Identity + NonPriv. Weighted LogReg | 0.91 ± 0.02 | 0.4 ± 0.02 | 0.84 ± 0.02 | 0.27 ± 0.02 / 0.73 ± 0.05 | 0.73 ± 0.05 | 0.84 ± 0.02 | 0.84 ± 0.03 | 0.43 ± 0.03 |
| Identity + NonPriv. XGBoost | 0.94 ± 0.02 | 0.69 ± 0.04 | 0.79 ± 0.03 | 0.82 ± 0.05 / 0.59 ± 0.06 | 0.59 ± 0.06 | 0.79 ± 0.03 | 0.77 ± 0.04 | 0.69 ± 0.03 |
| Identity + NonPriv. Weighted XGBoost | 0.94 ± 0.02 | 0.65 ± 0.04 | 0.87 ± 0.03 | 0.57 ± 0.04 / 0.75 ± 0.06 | 0.75 ± 0.06 | 0.87 ± 0.03 | 0.86 ± 0.04 | 0.65 ± 0.04 |
| SMOTE + NonPriv. LogReg | 0.91 ± 0.02 | 0.29 ± 0.01 | 0.86 ± 0.02 | 0.17 ± 0.01 / 0.82 ± 0.05 | 0.82 ± 0.05 | 0.86 ± 0.02 | 0.86 ± 0.02 | 0.35 ± 0.02 |
| SMOTE + NonPriv. Weighted LogReg | 0.92 ± 0.02 | 0.19 ± 0.01 | 0.85 ± 0.02 | 0.11 ± 0.01 / 0.88 ± 0.03 | 0.82 ± 0.01 | 0.85 ± 0.02 | 0.85 ± 0.02 | 0.27 ± 0.01 |
| SMOTE + NonPriv. Weighted XGB | 0.92 ± 0.02 | 0.66 ± 0.04 | 0.87 ± 0.02 | 0.58 ± 0.06 / 0.75 ± 0.04 | 0.75 ± 0.04 | 0.87 ± 0.02 | 0.86 ± 0.02 | 0.65 ± 0.04 |
| SMOTE + NonPriv. XGBoost | 0.93 ± 0.02 | 0.65 ± 0.04 | 0.86 ± 0.03 | 0.59 ± 0.05 / 0.73 ± 0.05 | 0.73 ± 0.05 | 0.86 ± 0.03 | 0.85 ± 0.03 | 0.65 ± 0.04 |
| Identity + NonPriv. Weighted FTTransformer | 0.88 ± 0.03 | 0.21 ± 0.16 | 0.59 ± 0.08 | 0.40 ± 0.34 / 0.19 ± 0.17 | 0.19 ± 0.17 | 0.59 ± 0.08 | 0.23 ± 0.23 | 0.16 ± 0.16 |

Table 20: Abolone_19 Dataset

| Metrics | Standard ↑ | | | | Imbalanced ↑ | | | |
|---|---|---|---|---|---|---|---|---|
| Approach | AUC | F1 | Bal-ACC | Prec./Recall | Worst-ACC | Avg-ACC | G-Mean | MCC |
| **Non-Private ↓** | | | | | | | | |
| Identity + NonPriv. LogReg | 0.75 ± 0.11 | 0.0 ± 0.0 | 0.5 ± 0.0 | 0.0 ± 0.0 / 0.0 ± 0.0 | 0.0 ± 0.0 | 0.5 ± 0.0 | 0.0 ± 0.0 | 0.0 ± 0.0 |
| Identity + NonPriv. Weighted LogReg | 0.72 ± 0.11 | 0.06 ± 0.05 | 0.59 ± 0.1 | 0.03 ± 0.03 / 0.25 ± 0.18 | 0.25 ± 0.18 | 0.59 ± 0.1 | 0.42 ± 0.25 | 0.07 ± 0.07 |
| Identity + NonPriv. XGBoost | 0.72 ± 0.12 | 0.03 ± 0.09 | 0.51 ± 0.03 | 0.1 ± 0.32 / 0.02 ± 0.05 | 0.02 ± 0.05 | 0.51 ± 0.03 | 0.04 ± 0.13 | 0.04 ± 0.13 |
| Identity + NonPriv. Weighted XGBoost | 0.76 ± 0.11 | 0.04 ± 0.04 | 0.55 ± 0.07 | 0.02 ± 0.02 / 0.15 ± 0.15 | 0.15 ± 0.15 | 0.55 ± 0.07 | 0.29 ± 0.26 | 0.04 ± 0.06 |
| SMOTE + NonPriv. LogReg | 0.82 ± 0.06 | 0.05 ± 0.01 | 0.76 ± 0.08 | 0.02 ± 0.0 / 0.72 ± 0.18 | 0.68 ± 0.13 | 0.76 ± 0.08 | 0.75 ± 0.09 | 0.11 ± 0.03 |
| SMOTE + NonPriv. Weighted LogReg | 0.83 ± 0.06 | 0.03 ± 0.0 | 0.73 ± 0.06 | 0.02 ± 0.0 / 0.8 ± 0.13 | 0.66 ± 0.02 | 0.73 ± 0.06 | 0.73 ± 0.05 | 0.08 ± 0.02 |
| SMOTE + NonPriv. Weighted XGB | 0.79 ± 0.1 | 0.08 ± 0.06 | 0.57 ± 0.06 | 0.06 ± 0.04 / 0.17 ± 0.11 | 0.17 ± 0.11 | 0.57 ± 0.06 | 0.36 ± 0.2 | 0.09 ± 0.07 |
| SMOTE + NonPriv. XGBoost | 0.78 ± 0.09 | 0.08 ± 0.07 | 0.57 ± 0.07 | 0.06 ± 0.05 / 0.15 ± 0.15 | 0.15 ± 0.15 | 0.57 ± 0.07 | 0.31 ± 0.23 | 0.08 ± 0.08 |
| Identity + NonPriv. Weighted FTTransformer | 0.56 ± 0.09 | 0.01 ± 0.01 | 0.53 ± 0.10 | 0.00 ± 0.01 / 0.20 ± 0.38 | 0.12 ± 0.22 | 0.53 ± 0.10 | 0.29 ± 0.29 | 0.04 ± 0.04 |

