# OpenReview forum: "Differential Privacy Under Class Imbalance: Methods and Empirical Insights"
_ICML.cc/2025/Conference — ICML 2025 poster_

### Official Review · Reviewer_PW3A · 2025-03-09

**Overall Recommendation:** 4

**Summary:**

This work looks at training classifiers with differential privacy (DP) guarantees in the presence of data imbalance (in the binary classification case) while enforcing fairness guarantees. They look at data augmentation methods and in-processing methods where fairness is attempted to be imposed by changing the model training process (viz. by using weights for different classes). It is seen that sophisticated private data augmentation methods (viz. GEM) tend to outdo in-processing based methods and certainly outperform non-private upsampling/oversampling methods. In addition, data augmentation needs to be done carefully and privately, as methods like oversampling or SMOTE can increase the sensitivity of the data due to dependence on existing minority samples.

**Claims And Evidence:**

* The claims are supported by strong empirical evidence over binary classification tasks for different models for 8 datasets from imbalanced-learn and across multiple methods.
* Some claims about privacy are backed up by theoretical DP guarantees.

**Essential References Not Discussed:**

No missing references come to mind.

**Experimental Designs Or Analyses:**

* I think there's only one issue, that is with the heterogeneity of model types/architectures used for comparing different methods, as discussed in "Methods and Evaluation Criteria". It might be that this is a solid evaluation, but I need to understand the authors' choice of using different models for different methods better: 1) Why did they decide to use different models (XGBoost, logistic regression, FTTransformer)? What influenced their choices? Is this really an apples-to-apples comparison? 2) What may change if they use the exact same classifier across privacy-addition methods, so to speak?

**Methods And Evaluation Criteria:**

* Yes they do.
* However, the authors do not appear to use the same classifier architecture across all different methods, which concerns me as regards to the fairness of comparison.

**Other Comments Or Suggestions:**

None.

**Other Strengths And Weaknesses:**

* Please refer to aforementioned comments. I think this paper has many strengths and provides a nice overview of using fairness-aware DPML methods to practitioners.
* However, one big weakness (potentially) is the heterogeneity of the classifier types used in the evaluation, which potentially is not an apples-to-apples comparison between the methods discussed in the paper.
* One minor limitation, which the authors acknowledge (and should not be held too strongly against them) is that their discussion and evaluation is limited to binary classification. However, it will be appreciated if the authors can provide a few more insights on generalizing it to the multiclass setting, beyond what is discussed in the paper.

**Questions For Authors:**

* Can the authors please run experiments over all the methods that they discuss but compare over the same architecture for a better apples-to-apples comparison? This alone, if addressed, will be sufficient for me to raise my score.
* Or can you please justify, very concretely, why it is valid and fair to have a comparison using different models for each method? I know that there might be some limitations (viz. bagging only being possible with certain architectures viz. XGBoost). However, could it be that there exists another compatible architecture which, if used, may lead to a different trend?

As such, with the heterogeneity between models, I cannot fully certify if the empirical takeaways are actually valid, because this does not seem like an apples-to-apples comparison. What would help is if these trends are seen while using the **same** model across all these methods.

**Relation To Broader Scientific Literature:**

* This supplements previous work on fairness in terms of utility and privacy (viz. Tran et al.) and works on alleviating data imbalance (viz. SMOTE), including by generating synthetic data with DP (viz. GEM, PrivBayes), in the context of machine learning, and contributes in terms of studying the efficacy of data processing or in-processing as a way of imposing fairness requirements. This investigation is timely and important as practitioners may benefit from studying the efficacy of these methods in terms of balancing fairness and privacy guarantees while maintaining utility.

**Theoretical Claims:**

* The proofs seem correct, at least for the one on naive oversampling. I have not been through other proofs in detail.

---

> ### Author Rebuttal · Authors · 2025-03-31
>
> > **…limited to binary classification…insights on generalizing to the multiclass setting?**
>
> Thank you for this interesting direction for extending our work; many of our results do extend naturally to multi-class settings. We’ll update our revised paper with an expanded version of the following:
>
> **For SMOTE Result:** In both Proposition 4 and Theorem 5, the term $\left\lceil\frac{N}{n_1}\right\rceil$ reflects the number of iterations needed over the minority samples (with $N$ additional samples generated and $n_1$ original minority instances). In a multiclass setting with \(c\) classes -- each with $n_1, \dots, n_c$ samples -- we can simply apply the procedure independently for each class. For class $i$, one generates $N_i$ synthetic samples so that the iteration term becomes $\left\lceil\frac{N_i}{n_i}\right\rceil.$ Taking the worst-case over classes, i.e., $\max_{i \in [c]}\left\lceil\frac{N_i}{n_i}\right\rceil,$ ensures that the overall privacy analysis carries over directly.
>
> **For ERM / DP-SGD:** Multinomial logistic regression and categorical cross-entropy loss (soft-max loss) are natural extensions of the binary setting to multi-class with these models. The loss (softmax with cross-entropy) is convex and differentiable with respect to the model parameters, so the sensitivity bounds used in our weighted ERM analysis hold -- with appropriate adjustments to account for the gradient computed over **all** classes.
>
> **For GEM / Other Synthetic Data:**
> Extending the DP synthetic data generation methods like GEM to multiclass settings is also straightforward! In the multiclass case, the generator is trained to learn the joint distribution over all variables, including a target variable that is potentially categorical.
>
> >  **...please justify why it is valid/fair to compare using different models for each method?**
>
> Thank you for raising this point. In the revised paper, we will clarify the driving question of our work, and add a version of the following discussion to address your concerns.
>
> Our work could be framed as a first step to answering the question: **Given that you want to make predictions using sensitive, class-imbalanced data, for a fixed privacy budget, which differentially private approach will give you the best performance?** We are not aware (as noted by Reviewer uyZY) of any other work that directly attacks this question.
>
> These comparisons were inherently limited by the availability of DP algorithms with explicit adaptations under class imbalance (for example, we went to great lengths to construct a weighted the DP-ERM algorithm, see Theorem 5, Lemma 14 and the proofs, etc. in Appendix C.3.1).
>
> Then, within each approach *variant* (disparate approaches both in pre-processing and in-processing), we selected candidates for highlighted comparison:
>
> 1. **Most performant pre-processing methods (GEM, DP Synthetic Data)**:
> We use DP synthetic data approaches (GEM and PrivBayes) to generate a class balanced, synthetic dataset. This decoupling allowed us to use any strong non-private classifier downstream. We chose XGBoost precisely because it is known for its robust performance on imbalanced data, and when tested in the non-private setting (see Appendix E.2) we found it generally outperformed a logistic regression baseline (as expected). We made a point to highlight that (strong private data synthesizer)+(strong non-private downstream classifier) is a promising framework; evaluating just that class of models could be its own follow-up work.
>
> 2. **Most performant in-processing methods (Weighted ERM and DP-SGD with FTTransformer)**:
> These methods are representative of, respectively, a private convex optimizer (for ERM) and a DP-SGD-trained neural architecture. Again, we needed to adapt an ERM algorithm under class imbalance. We believe adapting other private in-processing approaches under class imbalance is a very interesting direction for future work; however, it is not often straightforward due to particular care required in the privacy analysis for the sensitivity adjustment under weights.
>
> We did not highlight the empirical comparisons for SMOTE and for bagging (both private and non-private), as they performed poorly and reduced the clarity of our takeaways for the most performant methods we evaluated.
>
> In summary, we acknowledge that its worth carefully considering the appropriateness of assessing methods *across* model families, and appreciate your concern here. Our intent with this paper was to highlight the trade-offs that arise when addressing class imbalance under DP, and provide a study that covered a lot of ground. We’d further like to highlight that the heterogeneity in approach mirrors trends in prior work, where methods are often compared at fixed privacy budgets, in their most effective form, rather than compared within a uniform model class [e.g., Jayaraman et al. 2019, https://arxiv.org/abs/1902.08874 or Suriyakumar et al., 2021 https://arxiv.org/pdf/2010.06667 ).

---

> > ### Comment · Reviewer_PW3A · 2025-04-06
> >
> > Dear authors, thank you so much for your rebuttal and your detailed answers.
> >
> > I stand convinced of your first response, and I appreciate the thoroughness of it. As for the second, and I had to spend some time thinking about this because I appreciate your thorough response but struggled with how convinced I was with it: I understand the limitations, but to answer precisely the question you have framed up there, I am still not convinced that saying "using DP method X with model A outperforms DP method Y with model B, therefore, DP method A is better" is correct. It may be, but unless the authors use either the same methods or a well-justified metric that brings all of them on a level playing field, I am afraid I am not convinced by this argument, for a comparative study. It does, to an extent, answer, what is the "best" we can do with DP method "X" as compared to "Y", but that is not exactly the same question in my mind.
> >
> > Therefore, I'll have to retain my score. I am, however, open to discussion.

---

> > > ### Author Response · Authors · 2025-04-06
> > >
> > > **We’re glad that you appreciated our response on multi-class generalization;** we were pleased to have an opportunity to consider it and add a discussion in the revised paper.
> > >
> > >
> > > We understand your hesitation around our central comparative evaluation strategy; as you put it, you’re still not sure that "Using DP method X with model A outperforms DP method Y with model B, therefore, DP method X is better" is the right way to compare methods.
> > >
> > >
> > > *We believe it would be helpful to reframe the structure of our comparison, to motivate why the comparative structure we chose was natural, meaningful and practically motivated.* We do this in with following 3 points:
> > >
> > >
> > > **1. What Are We Comparing? Unit of Comparison Is a Pipeline, Not a Model:**
> > >
> > >
> > > In the presence of both class imbalance and privacy constraints, privacy-preserving methods are rarely deployed in isolation from downstream modeling choices. That is, differential privacy is not a plug-in property -- it can interact with data characteristics, model class/architecture and optimization choices. Therefore, we argue that the natural unit of evaluation is the full learning pipeline -- from DP mechanism to model class to optimization procedure.
> > >
> > >
> > > To be clear about what we mean by a “learning pipeline,” we’ll define the triple $(A, M, f)$, where:
> > >
> > >
> > > 1. $A$: the DP algorithm (e.g., GEM, DP-SGD, weighted DP-ERM, pre-processing steps that incur privacy loss, etc.)
> > > 2. $M$: the intermediate representation or data output (e.g., synthetic dataset, private gradients)
> > > 3. $f$: the final prediction function (e.g., XGBoost, logistic regression)
> > >
> > >
> > > Our comparisons then ask: **Given a fixed privacy budget $\epsilon$, and a practical goal of maximizing predictive performance on imbalanced data, which pipeline $(A, M, f)$ yields the best results?**
> > >
> > >
> > > We argue that this question is natural and important, because it reflects how DP methods are actually deployed: with tailored architectures and loss designs best suited to a level of privacy, data context and type of privacy mechanism used.
> > >
> > >
> > > **2. Conversely, A Uniform Architecture May Undermine the Validity of the Comparison:**
> > >
> > >
> > > We fully agree that "all else equal" comparisons (same architecture across multiple approaches to ensuring DP) are valuable for isolating specific effects. But here, enforcing architectural uniformity across fundamentally different DP techniques would introduce distortion:
> > > 1. Synthetic data methods (e.g., GEM) produce tabular data that can be passed to **any model.** GEM tends to perform best when paired with tree-based learners (which are known to perform well on tabular data), so it’s natural to use them downstream. However, we would be happy to add results comparing GEM+XGBoost with GEM+NonPrivLogReg and GEM+NonPrivFTTransformer if this would help address your concerns.
> > > 2. (Weighted) DP-SGD is designed and used with deep neural models in mind. Running (weighted) DP-SGD to update a model like logistic regression would likely not perform well at all, but we could try it if the comparison would alleviate some of the problems you see.
> > > 3. Similarly, the weighted ERM-based method targets convex losses under stronger (linear) model class assumptions, but this means we can conduct a more in depth privacy analysis.
> > >
> > >
> > > In summary, we argue that using the same architecture across all methods risks favoring some methods and punishing others. By contrast, our approach chooses the most appropriate and representative model for each method family, at **pipeline** scale of comparison
> > >
> > >
> > > **3. Standards from Prior Work**
> > > Our evaluation framework aligns with some precedents set by prior work in the DP literature. For example, as we noted previously, [Jayaraman et al. 2019] and [Suriyakumar et al. 2021] both compare multiple DP methods under their most effective training pipelines, not just under a uniform model architecture. Our goal is not to isolate architecture as a variable, but to ask: Which end-to-end strategy works best for private imbalanced learning? We believe this is a natural question to ask, especially as we conduct this initial study that tries to cover a lot of ground on DP and imbalance learning.
> > >
> > >
> > > **Your point is well taken though:** in our revised paper, we will clarify this evaluation philosophy. **We will also explicitly communicate that our empirical takeaways are on the scale of method+architecture pipelines** -- not about variations/nuances on the model classes in isolation (e.g. which architecture is best for weighted DP-SGD? etc.) which will require further exploration in future work.
> > >
> > >
> > > We hope that this can address your concerns about the fairness of comparison. Thanks for engaging in the rebuttal phase, we appreciate your thoughts and feedback.

---

### Official Review · Reviewer_EyJH · 2025-03-14

**Overall Recommendation:** 3

**Summary:**

This paper studies the problem of privacy when the dataset is imbalanced such that there is one class that has significantly less data points than the other class. Specifically, the paper tackles the problem of training a binary classifier on imbalanced data. Known techniques for up-sampling the minority class data samples are studied and compared, while discussing the shortcomings and advantages of each. This was also shown experimentally.
## update after rebuttal
The authors have shed more light to their contribution; therefore, I raised my score to a 3.

**Claims And Evidence:**

The paper supports the claims well with experiments.

**Essential References Not Discussed:**

N/A

**Experimental Designs Or Analyses:**

The experiments were designed well to support the claims of the paper. The comparisons done between the different private up-sampling techniques and their effect on the accuracy of the model supported the claims made in the paper.

**Methods And Evaluation Criteria:**

The methods to evaluate the results are valid and they use real datasets, which is very good to show the real implications of this problem and different techniques.

**Other Comments Or Suggestions:**

N/A

**Other Strengths And Weaknesses:**

The paper is very well written and easy to read and follow. However, my main issue with the paper is the originality. The paper uses different known techniques and compares them while giving some insights about them. I think this work is incremental despite offering a nice overview on the topic. I would recommend another venue for this work.

**Questions For Authors:**

N/A

**Relation To Broader Scientific Literature:**

This work offers a good overview and comparison of the different techniques that could be used when the data is imbalanced. This would be helpful in some areas, such as medicine, where data could be very skewed.

**Theoretical Claims:**

Although the work is mostly empirical, there are some propositions, lemmas, and theorems. Mainly, the proofs are in the appendix. I checked the correctness to the best of my ability, and I did not find any issues.

---

> ### Author Rebuttal · Authors · 2025-03-31
>
> > **The paper is very well written and easy to read and follow.**
>
> Thank you for your positive feedback regarding the clarity and readability of our paper. We spent a lot of time considering how best to present the nuances of this particular classification setting, and so appreciate this recognition.
>
> > **However, my main issue with the paper is the originality. The paper uses different known techniques and compares them while giving some insights about them. I think this work is incremental despite offering a nice overview on the topic.**
>
> We respectfully disagree; we believe ICML is the right venue for this work. Our paper builds on established imbalanced learning techniques and standard DP mechanisms, and it is, to the best of our knowledge, the first comprehensive study that systematically investigates differential privacy under class imbalance. We acknowledge that we evaluate class-imbalanced adaptations of pre-existing methods; however, this work is quite involved, and crucial to answering the questions our paper poses. We highlight several contributions, including:
>
> **(1)** We rigorously show that well-known methods such as SMOTE and non-private bagging -- techniques that have been successfully applied in non-private imbalanced classification settings -- can dramatically inflate the sensitivity / render privacy guarantees meaningless when directly applied under DP.
>
> **(2)** We introduce a weighted variant of the canonical private ERM approach, filling a notable gap in prior work (Theorem 5, see Lemma 14 and Section C.3.1 for the extensive necessary adjustments to show privacy).
>
> **(3)** Our extensive empirical study -- across multiple imbalanced datasets and a range of privacy budgets -- is the “first work to extensively study the class-imbalance setting under DP” as noted by Reviewer uyZY. Our results not only demonstrate the limitations of certain approaches but highlight promising methods like DP synthetic data (via GEM) as (pre-processing) and the weighted ERM approach (as in-processing).

---

### Official Review · Reviewer_KHpD · 2025-03-14

**Overall Recommendation:** 3

**Summary:**

This paper deals with the (in)consistency of differential privacy and imbalanced class learning, especially binary classification problem where the minority class is very small. The non-private learning algorithms for the imbalanced classes usually increase the weights of minority classes through oversampling, argumentation, Bagging, reweighting methods etc. However, these methods are shown to be inconsistent with differential privacy because DP would increase the privacy risk, bias and unfairness. So this paper evaluated the private versions of these methods and provide some suggestions.

**Claims And Evidence:**

yes

**Essential References Not Discussed:**

no as far as I know

**Experimental Designs Or Analyses:**

yes. The experiments are well-designed

**Methods And Evaluation Criteria:**

yes

**Other Comments Or Suggestions:**

(1) Line 85: undersampling -> subsampling
(2) Line 152: L2->L_2

**Other Strengths And Weaknesses:**

Strengths:

(1) The paper provided a systemetic evaluation of different differential private algorithms used in the standard machine learning for imbalanced classes.  The observations are interesting and convincing.
(2)The paper tackles an interesting and important problem in practice.

Weakness:
(1) The abstract says this work"formalizes these challenges". But I did not find such a formalization.  Also the abstract says that this paper "provides a number of algorithmic solutions".  I could not find those solutions in the paper.   My impression is that this paper mainly focus on the evaluation of the DP version of those learning algorithms for imbalanced learning not on the methods.
(2) The presentation is a little sloppy. Moreover, different parts seems to be isolated from each other.  There are many propisitions in this paper. It seems that they are just flattened.  I don't know which one is the central and main proposition.
(3) Proposition 2 is trivial.  It is an easy observation.
(4) Theorem 5 is very confusing.  There are no explicit intuition before this proposition.
(5)  Algorithm deals with multi-label classification inAlgorithm 1 instead of binary classification problem as specified in Introduction.

**Questions For Authors:**

Q1:  In the last senetence of the first paragraph in Introduction, "these methods ...  assume that false positives and false negatives have equal misclassification costs". What does this mean?
Q2:  Could you explain more about the privacy loss scale in Lines 78-80?
Q3:  Could you explain Line 6 inAlgorithm 1?
Q4: Could you explain the reason why "Re-weighting of samples in the loss function pre-clipping does not affect these privacy
guarantees" (Line 1795)?

**Relation To Broader Scientific Literature:**

This paper helps better understand the relationship between differential privacy and those standard learning algorithms for imbalance classes.    It seems that it is related to invidualized differential privacy in the literature, which is not discussed in the paper.

**Theoretical Claims:**

not all of them. I checked some essential ones and those in Section 3. They are correct.

---

> ### Author Rebuttal · Authors · 2025-03-31
>
> > **...related to invidualized differential privacy, not discussed...**
>
> We thank the reviewer for raising this point. While our sensitivity analysis -- specifically, how certain samples in imbalanced datasets incur higher privacy loss -- echoes themes from individualized/personalized DP, our work is rooted in standard, global worst-case DP. We will add a brief discussion in the revised version to clarify the connection.
>
> > **(W1) ...did not find a formalization / could not find solutions...**
>
> A fully general formalization of the challenges in DP imbalanced classification is difficult without strong distributional assumptions. Instead, we formalize key sub-problems. For example, Theorem 3 and Proposition 4 characterize the SMOTE and bagging approaches under class imbalance, and Example 11 with Proposition 12 (deferred to the Appendix) does so under a Gaussian mixture assumption. We would be happy to revise from "formalizes these challenges" to more precisely stating that we formalize "approach-specific challenges."
>
> Our solutions include both theoretical insights and concrete algorithmic adaptations. For example, we introduce a private weighted DP-ERM algorithm (Algorithm 1, Theorem 5) and analyze DP-SGD with weighted cross-entropy (Proposition 6, Lemma 19). We also propose a class-conditional sampling pre-processing approach with private synthetic data (Algorithm 3) and provide extensive empirical evaluations. We’d be happy to also adjust the language to clarify these claims.
>
> > **(W2)...didn't know what was central and main proposition...**
>
> Our work does not have a single central theorem; instead, we compare a variety of methodologies for imbalanced classification under differential privacy constraints.
>
> > **(W3)...Proposition 2 trivial...**
>
> We agree, Proposition 2 (regarding the sensitivity amplification via oversampling) is straightforward. We wanted to formalize this intuitive observation to set the stage for the more complex sensitivity analysis in the SMOTE setting (Theorem 5, see Appendix B.1 for proof). We’d be happy to present it informally, inline, so that the central contributions remain emphasized.
>
> > **(W4)...Theorem 5 is confusing...**
>
> Due to space constraints, we deferred much of the discussion and setup for Theorem 5 to Appendix C.3 (in particular, see the paragraph titled “Notation for ERM Proof” and the subsequent Lemma 14, which accounts for the main sensitivity adjustment for weights for privacy). In short, Theorem 5 shows that Algorithm 5 is differentially private. The two paragraphs preceding Theorem 5 respectively provide **(1)** the notation with discussion of weights, and **(2)** intuition for how the weights play a role in the proof. We'd be happy to move anything from Appendix C.3 back into the body to make this result more clear.
>
> > **(W5) (typos)**
>
> Thank you for catching the typos - we will correct them in the revised version (e.g., adjust Line 1 in the Algorithm so that $y_i \in \\{0,~1\\}$)
>
> > **(Q1) [what do we mean by assuming that FP and FN have equal misclassification costs]?**
>
> This refers to the common design assumption that the cost of a false positive is identical to that of a false negative. For many applications where imbalanced learning commonly arises, this assumption doesn’t hold. For example, in detecting rare cancers or financial fraud, missing a rare but critical positive event (false negative) might be far more consequential than a false alarm.
>
> > **(Q2)...privacy loss scale in Lines 78-80?**
>
> Certainly. In those lines, we briefly summarize how SMOTE can drastically increase the sensitivity of a downstream DP algorithm. Specifically, by generating multiple synthetic points from a single minority example, the effective privacy parameter $\epsilon$ is scaled by a factor that is exponential in the data dimension and linear in the number of synthetic points. This means that even if the base algorithm is $\epsilon$-DP, applying SMOTE without proper adjustments as a pre-processing step may lead to an effective privacy loss ($\epsilon’$) that is substantially higher. See e.g., the scaling and reverse-scaling we give in Table 2.
>
> > **(Q3) Line 6 in Algorithm 1**
>
> Line 6 describes the core optimization of the weighted DP-ERM algorithm: minimizing the weighted empirical risk, which combines the per-sample loss (weighted by class frequency), an objective perturbation noise term, and a regularizer. We would be happy to elaborate on this step (and provide references for its standard use) in the final version for added clarity.
>
> > **(Q4) (explain Line 1795)?**
>
> The privacy guarantees in our algorithm are based on the sensitivity of the per-sample gradient, which is controlled by a clipping step that bounds the norm of each gradient to a fixed constant C. Although re-weighting changes the magnitude of the computed gradients, the subsequent clipping ensures that no individual sample's contribution exceeds C. Thus, the overall sensitivity remains unchanged.

---

### Official Review · Reviewer_uyZY · 2025-03-14

**Overall Recommendation:** 4

**Summary:**

The paper explores class imbalance in differentially private ML settings. The authors consider common pre-processing and in-processing methods for dealing with class imbalance, and look at extending them to the DP setting. They show that some commonly used non-private methods like SMOTE are not well suited to the DP setting and that alternatives like DP synthetic data or DP-weighted ERM perform better in a private class imbalance setting. Experiments are performed over 8 benchmark imbalanced binary classification datasets.

**Claims And Evidence:**

The paper generally make two main claims:
1. The first claim is that oversampling methods are poor for DP settings as they increase the sensitivity of DP algorithms that are trained on them. This is supported by clear theorems that show the sensitivity is amplified and these approaches are not well-suited.
2. The second is that the use of DP synthetic data is the strongest approach for dealing with class-imbalance. This is fairly well-supported by an extensive set of experiments, although some results I feel are misleading or unclear (see weaknesses + questions below).

**Essential References Not Discussed:**

There are no related works that I feel are missing or not discussed.

**Experimental Designs Or Analyses:**

The experimental design is generally sound. The methods are ranked across 8 benchmark datasets for imbalanced data and a specific dataset is chosen to highlight multiple evaluation metrics across the methods.  However, I have a few issues with the setup used for GEM (see questions below).

**Methods And Evaluation Criteria:**

The methods and evaluation criteria used are suitably chosen for the problem and are fairly exhaustive. The paper considers a good number of methods for dealing with class imbalance (as highlighted in Table 1) and how to extend them to a DP setting. Experiments are performed over 8 commonly used benchmark datasets in imbalanced binary classification problems and multiple evaluation metrics are presented for the classifiers.

**Other Comments Or Suggestions:**

N/A

**Other Strengths And Weaknesses:**

**Strengths:**

- Class imbalance in a DP setting is an important practical problem that has many uses yet is not well-studied in the literature.  This paper addresses and fills this research gap.
- The paper covers a wide-range of class imbalance methods for both pre-processing and in-processing and shows how to extend them to the DP setting. The experiments are also extensive and show a clear conclusion that DP synthetic data is the most robust method.
- The paper is well-written and generally clear.

**Weaknesses:**

- The results are limited to binary classification settings and it is not immediately clear how findings or some methods can be extended to a multi-class setting.
- The leading conclusion is that DP synthetic data is a strong option to handle class imbalance, however the evaluation against other baselines seems unfair (see below).
- I think SMOTE presented as is, is also an unfair baseline (see questions below).

**Questions For Authors:**

1. The GEM+XGBoost method is misleading because it is compared against (mostly) logistic regression baselines. It is unclear from Table 3 how much of the benefit comes from the DP synthetic data or if the benefit is from the actual XGBoost model vs. the simpler logistic regression models. Why was XGBoost chosen instead of using GEM+LogReg? Indeed, some of the tables in the Appendix for the non-private setting highlight that there is often a big accuracy discrepancy between (non-private) LogReg and XGBoost and this would also be reflected in the DP experiments.
2. The comparison for SMOTE seems somewhat unfair. As stated, using SMOTE as-is increases the DP sensitivity which in turn amplifies the DP epsilon needed to maintain the same level of DP which is a nice result. However, the actual SMOTE algorithm has not been adapted for privacy compared to other methods specifically changed to provide DP guarantees. Do you see natural extensions to SMOTE that are privacy-friendly, i.e., by perturbing points that are sampled?
3. How can the leading methods be extended to multiclass settings? It’s briefly discussed for the oversampling methods like SMOTE, but do the methods like synthetic data and weighted ERM easily extend?
4. For GEM, it is unclear to me if rejection sampling was used or conditional sampling? If conditional sampling, was the generator network structure changed over the standard GEM approach?
5. Was any of the GEM training procedure changed to adapt to a class-imbalanced setting? More specifically, for the workload of queries given to GEM, do you have suggestions for achieving best accuracy in imbalanced settings?
6. To clarify in L340, I presume $B$ in the $2C/B$ refers to the DP-SGD minibatch size? I am not sure this is clear from the main paper.

**Relation To Broader Scientific Literature:**

The key contributions fit well into the broader literature on DP-ERM and private synthetic data generation and as far as I am aware, these methods have not been studied extensively in class-imbalance settings. To the best of my knowledge, this is also the first work to extensively study the class-imbalance setting under DP.

**Theoretical Claims:**

I did not review the full technical proofs (contained in the appendix) but the stated results in the main paper seem to logically follow.

---

> ### Author Rebuttal · Authors · 2025-03-31
>
> > **(Q1) Why XGBoost instead of using GEM+LogReg?**
>
> Thank you for your comment, hopefully we can clarify our choices here. Our primary goal was to compare general approaches to handling imbalanced classification under differential privacy -- not necessarily to benchmark model families (e.g., logistic regression vs. XGBoost, etc.). In our setup, we tested GEM and PrivBayes as differentially private pre‐processing steps to generate a balanced, synthetic dataset. This approach decouples the rebalancing from the classifier so that we can use any strong non-private model downstream, as we know that private ERM (regression) methods have performance limitations [e.g., Jayaraman et al. 2019, https://arxiv.org/abs/1902.08874]. Then, we chose XGBoost because this is the choice that a practitioner would likely make due to its well known overall performance, and in particular its robustness on imbalanced datasets.
>
> However, you are correct that, although a non-private XGBoost model will generally outperform non-private logistic regression, in some of our non-private experiments in Appendix E.2 the LogReg method slightly outperformed XGBoost. However, in our private experiments, we found that the GEM+XGBoost approach was more robust to noise introduced in the dataset for privacy. We’d be happy to add these experiments on the performance of the GEM+LogReg approach into the final version of our paper. However, we stress that our aim generally was to illustrate that leveraging DP synthetic data enables the use of state-of-the-art non-private classifiers, and that (strong private data synthesizer)+(strong non-private downstream classifier) is a promising approach.
>
> > **(Q2) Do you see natural extensions to SMOTE that are privacy-friendly...?**
>
> Thank you for bringing this up; as part of this work, we did spend time considering how one might develop a differentially private SMOTE algorithm. However, in our analysis for Theorem 3, we showed that SMOTE’s linear interpolation approach makes it inherently very sensitive (i.e., its sensitivity grows exponentially with the data dimension and linearly with the number of synthetic samples). In other words, even if one were to add noise directly to the interpolated points (with a differentially private additive noise mechanism), the resulting privacy loss would be unacceptably high. We view this as a key challenge: the reliance on the precise locations of pairs of minority points renders a direct privatization of SMOTE impractical.
>
> Instead, our analysis of SMOTE motivated us to leverage established DP synthetic data methods based on the ``Select–Measure–Project’’ paradigm. These methods learn a DP approximation of the underlying data distribution and then generates synthetic samples in a way that avoids the high sensitivity pitfalls of linear interpolation (by relying on less sensitive $k$-way marginal measurements). So, in summary, our results suggest that using DP synthetic data generation methods circumvents linear interpolation between minority examples in the data (which we showed was highly sensitive, making a direct differentially private adaptation of SMOTE impractical).
>
> > **(Q3)...How can the leading methods be extended to multiclass settings?**
>
> Thank you for this interesting direction for extending our work! **Please see our response to *Reviewer PW3A*, who also asked for a discussion of this extension.**
>
> > **(Q4)...rejection sampling or conditional sampling (GEM)**
>
> In our experiments we adopted a conditional sampling strategy -- this is sample efficient and directly addresses class imbalance. Importantly, the underlying generator network structure remains unchanged relative to the standard GEM approach. We will make this more clear in the final version of the paper.
>
> > **(Q5)...was any of the GEM training procedure changed?**
>
> No substantial changes were made to the core training procedure of GEM itself. Our approach leverages the standard GEM training protocol; the adaptation to handle imbalance is performed at the sampling stage. We view this as a feature, since practitioners would not have to change their training procedures to accommodate imbalanced data. Regarding the workload of queries (i.e., the set of measurements used during synthetic data generation), our experiments relied on the default $k$-way marginal selection. That said, in general we believe that selecting queries that capture the most informative low-dimensional marginals -- especially those relevant to the class imbalance itself -- could potentially improve performance in imbalanced settings. This is an interesting direction for future work, and we will add a discussion on it in the final version, thank you.
>
> > **(Q6)...DP-SGD minibatch size clarification.**
>
> Apologies, that did not make it up from the statement of Lemma 19 in the Appendix, but yes, that is correct, B denotes the DP-SGD minibatch size. We will make sure to state this in the final version, thank you!

---

> > ### Comment · Reviewer_uyZY · 2025-04-08
> >
> > I thank the authors for their response. I feel most of my questions have been adequately addressed and I have raised my score to a 4.
> >
> > However, I strongly encourage the authors to include results on GEM+LogReg in a revised version (whilst still keeping GEM+XGBoost) to give a data point for a more uniform comparison. I have read the discussion with Reviewer PW3A and feel the argument about studying methods as a combined pipeline is valid (i.e., an advantage of DP-SDG is it can be used with any model), but still feel strongly there should at least be some initial experiments where GEM+LogReg is compared.

---

> > > ### Author Response · Authors · 2025-04-08
> > >
> > > Dear Reviewer uyZY,
> > >
> > > We're pleased that our response addressed many of your concerns, and that you took the time to review our discussion with Reviewer PW3A and found it convincing. We will add results on GEM+LogReg in the revised version of the paper, as requested.
> > >
> > > Thank you again for taking the time to engage with us during this rebuttal phase!

---

### Decision · Program_Chairs · 2025-05-01

**Decision:**

Accept (poster)

**Comment:**

This paper initiates an examination of the differential privacy properties of representative pre-processing and in-processing methods for class imbalance learning, with new insights backed up by theoretical and empirical justifications.

The work is well-presented and paves the way for further investigations on private class imbalance learning. All reviewers are on the positive side. In the final paper, authors are encouraged to clarify the issues raised in reviewers, especially regarding the choice of baselines in Table 3 and its implications as suggested by reviewers uyZY/PW3A.